# The nematode worm *C. elegans* chooses between bacterial foods as if maximizing economic utility

Abraham Katzen[1], Hui-Kuan Chung[2,3†], William T Harbaugh[4†],
Christina Della Iacono[1], Nicholas Jackson[1], Elizabeth E Glater[5], Charles J Taylor[6],
Stephanie K Yu[7], Steven W Flavell[7], Paul W Glimcher[2,3], James Andreoni[8],
Shawn R Lockery[1]*

[1]Institute of Neuroscience, University of Oregon, Eugene, United States; [2]Center for Neural Science, New York University, New York, United States; [3]Neuroscience Institute, New York University School of Medicine, New York, United States; [4]Department of Economics, University of Oregon, Eugene, United States; [5]Department of Neuroscience, Pomona College, Claremont, United States; [6]Department of Chemistry, Pomona College, Claremont, United States; [7]Picower Institute for Learning and Memory, Department of Brain & Cognitive Sciences, Massachusetts Institute of Technology, Cambridge, United States; [8]Department of Economics, University of California, San Diego, La Jolla, United States

*For correspondence:
shawn.lockery@nemametrix.com

†These authors contributed equally to this work

**Abstract** In value-based decision making, options are selected according to subjective values assigned by the individual to available goods and actions. Despite the importance of this faculty of the mind, the neural mechanisms of value assignments, and how choices are directed by them, remain obscure. To investigate this problem, we used a classic measure of utility maximization, the Generalized Axiom of Revealed Preference, to quantify internal consistency of food preferences in *Caenorhabditis elegans*, a nematode worm with a nervous system of only 302 neurons. Using a novel combination of microfluidics and electrophysiology, we found that *C. elegans* food choices fulfill the necessary and sufficient conditions for utility maximization, indicating that nematodes behave as if they maintain, and attempt to maximize, an underlying representation of subjective value. Food choices are well-fit by a utility function widely used to model human consumers. Moreover, as in many other animals, subjective values in *C. elegans* are learned, a process we find requires intact dopamine signaling. Differential responses of identified chemosensory neurons to foods with distinct growth potentials are amplified by prior consumption of these foods, suggesting that these neurons may be part of a value-assignment system. The demonstration of utility maximization in an organism with a very small nervous system sets a new lower bound on the computational requirements for utility maximization and offers the prospect of an essentially complete explanation of value-based decision making at single neuron resolution in this organism.

## Editor's evaluation

In this thought-provoking study, the authors adopt a framework from economic decision-making theory to food choice behaviours in the nematode *C. elegans*. Based on their findings, they propose that worms behave consistently with the Generalized Axiom of Revealed Preference, a classic measure of utility maximization.

## Introduction

One of the primary functions of the brain is to make decisions that maximize a person's *welfare*. Welfare, meaning satisfaction of needs and desires, is subjective, based on values the individual assigns idiosyncratically to goods and outcomes. Can welfare maximization nevertheless be investigated in objective terms? One solution to this problem is revealed preference theory (*Samuelson, 1938*) which identifies the patterns of behavior, observable as such, that are necessary and sufficient evidence that subjects are choosing in ways consistent with welfare maximization or, in economic terminology, *utility maximization*. These patterns have been defined mathematically by the Generalized Axiom of Revealed Preference (GARP; *Houthakker, 1950*; *Afriat, 1967*; *Varian, 1982*). Previous studies have utilized GARP to quantify utility maximization in children and adults under a variety of economic and physiological conditions (*Harbaugh et al., 2001*; *Andreoni and Miller, 2002*; *Burghart et al., 2013*; *Lazzaro et al., 2016*).

GARP is significant for decision neuroscience because it provides a definitive behavioral test for utility maximization, or its absence. This test can be applied to almost any organism that makes choices between desirable goods that incur costs. The basic concept underlying this axiom is that a maximizing agent's choices must be internally consistent. If a person is observed to prefer X over Y when both are available then, other things being equal, this person should not also prefer Y over X, a pattern that is obviously inconsistent. Furthermore, internal consistency must extend to preferences revealed indirectly, through transitivity. For example, a persons observed to choose X over Y and Y over Z, have indirectly revealed that they would choose X over Z. If instead Z is chosen over X, then their decision making cannot be an instance of goal-directed maximization in any significant sense of the term. The number and severity of GARP violations (assuming the chooser is motivated to maintain or improve his or her welfare), has been taken as a measure of cognitive function (*Camille et al., 2011*). In humans and non-human primates, it can also be correlated with physical variables such as neuroanatomy and neuronal activity (*Chung et al., 2017*; *Pastor-Bernier et al., 2019*). These studies reveal the range of insights that can be gained by combining revealed preference theory and neuroscience.

To our knowledge, however, tests for utility maximization using GARP have yet to be applied to organisms more amenable to mechanistic studies such as mice, zebrafish, fruit flies, and nematodes. A major goal of this study was to determine whether food choices of the nematode *C. elegans*, a microscopic round worm with a nervous system of only 302 neurons, are consistent with GARP and thus exhibit a form of utility maximization. A positive result would establish a simple experimental system in which neuronal activity correlated with utility could be manipulated both physiologically and genetically to establish behavioral causality. Such a finding would also be interesting from a comparative perspective, extending the domain of utility-based decision making far beyond the boundaries of organisms that are generally considered to have cognition.

A worm might seem a surprising choice for investigating utility maximization. However, *C. elegans* possesses a sophisticated behavioral repertoire that can be organized into three broad functional categories (*Faumont et al., 2012*; *Yapici et al., 2014*): (1) maintenance behaviors, such as feeding, defecation, mating, and egg laying; (2) escape reflexes, for avoiding life threatening conditions such as noxious heat, ultraviolet light, high oxygen or $CO_2$, toxins, desiccation, and predation by fungi, mites, and other nematodes; and (3) habitat and resource-localization behaviors, including a variety of spatial orientation strategies that enable *C. elegans* to obtain goods such as hospitable living conditions and resources (e.g. food and mating partners), and to avoid inhospitable conditions and the lack of resources.

*C. elegans* exhibits a considerable range of decision making behaviors (*Faumont et al., 2012*; *Yapici et al., 2014*): (1) action versus inaction, such as probabilistic withdrawal responses *Culotti and Russell, 1978*; *Chalfie et al., 1985*; *Shinkai et al., 2011*; (2) approach versus avoidance, such as when an initially attractive odor or taste is made aversive by pairing it with the absence of food *Colbert and Bargmann, 1995*; *Saeki et al., 2001*; *Torayama et al., 2007*; (3) appetitive responses, such as when worms are presented with a choice between benign or pathogenic food *Zhang et al., 2005*; and (4) choice under risk, such as when worms must decide whether to risk crossing a potentially lethal chemical barrier to obtain food (*Shinkai et al., 2011*; *Ghosh et al., 2016*). Other examples include the choice to remain in a food patch rather than to leave to find a mate (*Barrios et al., 2008*) or the choice to remain in a dwindling patch of food rather than leave for a possibly better patch

*Bendesky et al., 2011*; *Milward et al., 2011*; the latter has strong parallels with optimal foraging theory (*Busch and Olofsson, 2012*). *C. elegans* has also been shown to exhibit bounded rationality (*Simon, 1957*), a property it shares with humans and most other animals. Its pairwise preferences for attractive odors generally obey transitivity but with a considerable number of exceptions (*Iwanir et al., 2019*). Similarly, its pairwise preferences are not generally influenced by introduction of a third option in the choice set, another classical mark of rationality, but again with a considerable number of exceptions (*Cohen et al., 2019*). However, none of these preference tests provide necessary and sufficient evidence for utility maximization.

We selected food choice for our investigation of utility maximization. *C. elegans* is an omnivorous bacterivore that mainly inhabits rotting plant material such as decaying fruits and fleshy stems (*Frézal and Félix, 2015*). Its natural habitat contains thousands of different species of bacteria (*Samuel et al., 2016*), including both beneficial and pathogenic varieties. Each beneficial species has a characteristic nutritional quality defined in terms of the growth rate of individual worms cultured on that species (*Avery and Shtonda, 2003*; *Samuel et al., 2016*). In contrast, pathogenic species can be lethal (*Tan et al., 1999*). Therefore, food choice has immediate fitness consequences for the worm, and it is reasonable to expect that *C. elegans* food-choice mechanisms have been shaped by evolution to be efficient and therefore internally consistent.

There is considerable evidence that, with the exception of some species of pathogenic bacteria (*Zhang et al., 2005*), food preference in *C. elegans* is not innate. Over the time course of a standard chemotaxis assay (1 hr), previously unfed L1 larvae accumulate equally in high (H) and medium (M) quality food patches located on the same test plate. Similar results were obtained for eight of nine other pairs of beneficial bacteria that differed in quality (*Shtonda and Avery, 2006*). This observation suggests that innate preferences based on chemotaxis to odors are mostly absent. Differential accumulation in larvae develops gradually, reaching a maximum at 24 hr. During this period which worms leave and re-enter the patches many times, providing the basis for a comparative mechanism. Mutants with compromised feeding ability (*eat-2*, *eat-5*), accumulate more strongly than wild type worms in the better of two strains of *E. coli* that differ in quality but are unlikely to differ in smell. The most plausible explanation for this observation is that food consumption provides feedback that drives food choice. Preferences acquired in one food environment (a bacteria laden agar plate) are retained when worms are transferred to a second environment containing different foods. For example, worms pre-exposed to high-quality food on the first plate become fussy eaters, exhibiting a bias against medium-quality food when transferred to a second plate containing novel high- and medium-quality foods (*Shtonda and Avery, 2006*). Worms also exhibit a preference for previously encountered food relative to novel food. Food preferences can be reversed by pairing the less preferred food with methamphetamine or cocaine, drugs of abuse that are associated with reward in humans (*Musselman et al., 2012*). Together, these observations show that worms behave as if they form memories of previously encountered food in a process can be described as *food quality learning*.

In this study, we utilized GARP to test for utility maximization and investigated its behavioral and neuronal mechanisms. This was accomplished by means of a microfluidic device that enabled us to offer single, semi-restrained worms high- and medium-quality bacteria at a range of different relative abundances while monitoring consumption electrophysiologically. We found food choices of naïve and trained animals are consistent with utility maximization. Worms behave as if they employ an underlying representation of utility that they were acting to maximize. Preference data were well fit by a utility function widely used to model the behavior of human consumers. At the behavioral level, utility maximization relies on a chemotaxis strategy known as *klinotaxis*. In this strategy, head bends during sinusoidal locomotion are biased by chemosensory input such that bends are deeper on the side where attractant concentration is increasing. At the neuronal level, we found that chemosensory neurons known to modulate head position are able to discriminate between high- and medium-quality food, and that food quality training increases this ability. These findings establish a new model system in which to investigate the neuronal and genetic basis of subjective value and its behavioral expression.

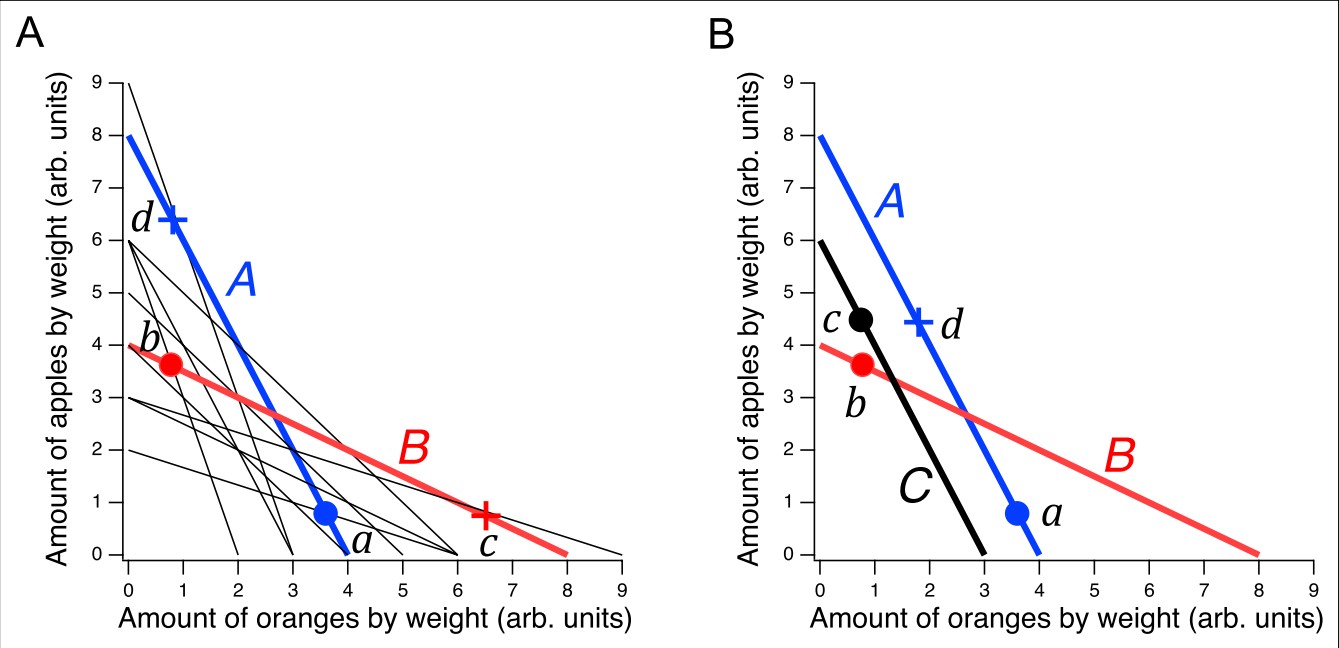

**Figure 1.** Design of a GARP experiment and tests for utility maximization. (**A**) Direct violation of utility maximization. Diagonal lines indicate choice sets (*n*=11). Choice sets are distinguished by having different values of the overall budget and/or different prices for at least one of the goods. Within a choice set, the expenditure implied by each bundle of goods is constant and equals the budget. In *A* (*blue*), the budget is $8, oranges are $2 per unit, apples are $1 per unit. In *B* (*red*): the budget is $8, oranges are $1 per unit, apples are $2 per unit. *Filled circles*, chosen amounts; *plus signs*, available amounts not chosen. Given the choices shown and the more is better rule, $a \succcurlyeq d \succ b \implies a \succ b$ and $b \succcurlyeq c \succ a \implies b \succ a$. Therefore, choices *a* and *b* directly violate utility maximization. (**B**). Indirect violation of utility maximization. Symbols as in A. The choices *a* and *c* constitute an indirect violation of utility maximization as described in the text.

## Results

### Revealed preference theory

In a typical experiment testing utility maximization with human participants (e.g. *Harbaugh et al., 2001*), each person is given a series of *choice sets*. A choice set comprises a list of consumption options, called *bundles*, from which participants are asked to pick the bundle they most prefer. Choice sets are constructed so that the available bundles are made from varying quantities of goods (e.g. apples and oranges; *Figure 1A*). Some experiments use more than two goods, but for exposition will we focus the simple case of two goods. Each choice set is defined by a unique combination of a budget and prices, and the cost of each bundle equals the budget. Budget and prices can be in units of money or of time. As a result of the budget constraint, a bundle with more of one good necessarily has less of the other, yielding an inherent *trade-off* between goods.

*Figure 1A* shows an ensemble of such choice sets, each of which can be conceptualized as offering two goods at different prices and under different budgets. The lines in the figure, called *budget lines*, depict the pricing constraints and trade-offs particular to each choice set. In *A*, for example, oranges are twice the price of apples so the chooser must forgo two units of apples for each additional unit of oranges. Choosing is construed as selecting the most preferred option from those available in the choice set. While in theory a person might choose a bundle inside the budget line and leave money on the table, in practice we assume more is preferred to less, and so these choices are typically not offered.

The constellation of choices a person makes across the many different choice sets in the ensemble is analyzed for the presence of combinations that violate the Generalized Axiom of Reveled Preference (GARP) which has been shown to be a necessary and sufficient condition for observed choices to be consistent with utility maximization. Here, we provide a non-technical explanation of the underlying theory; technical treatments are available elsewhere (*Varian, 1982*; *Harbaugh et al., 2001*; *Burghart et al., 2013*).

Revealed preference violations can be direct or indirect. Both types of violations depend on the assumption that more of a good is better than less of it (*strong monotonicity* of utility). The filled circles in *Figure 1A* are one possible pair of choices, $a$ and $b$, selected to exemplify a *direct* violation. In choice set $A$, $a$ was selected over $d$, which was also in the choice set but was not chosen. We infer from this choice that $a$ is at least as good as $d$, which we write as $a \succcurlyeq d$. (We cannot conclude that $a$ is better than $d$ because the person could have been indifferent between them, with $a$ being chosen randomly.) Noting that $d$ has the same number of oranges but more apples than $b$, we infer (by strong monotonicity) that $d$ is *strictly* preferred to $b$, written as $d \succ b$. Combining the inferences $a \succcurlyeq d$ and $d \succ b$, we conclude that $a$ should be strictly preferred to $b$, that is, it should also be true that $a \succ b$. This preference is said to be revealed *directly*, because when people choose $a$, they do so over another option on the same budget line, $d$, that has more of at least one of the goods than $b$. Similar logic applies to choice set $B$ such that $b$ is directly revealed preferred to $a$, that is, $b \succ a$. These two preferences constitute a violation because they are inconsistent; there is no underlying maximization process of any kind that could allow for this combination of choices.

A person *indirectly* reveals a preference for one bundle over another when there is a *sequence* of directly revealed preferences that link the two by transitivity. *Figure 1B* illustrates an indirect violation. In addition to the original choices $a$ and $b$, the person picked $c$ in choice set $C$. We observe that $c$ is preferred to $b$ because it has more of at least one good than the latter, so $c \succ b$. And, as before, $b \succ a$, from which we conclude by transitivity that $c$ is preferred to $a$. We write this as $c \succ *a$, where the asterisk indicates indirectness. However, at the same time the person reveals $a \succcurlyeq d$ while $d \succ c$, from which we conclude $a \succ *c$. These two preferences constitute a violation because they are inconsistent; again, no maximization process could allow for this combination of preferences. Further, it can be shown that when only two goods are available, the presence of direct violations is a necessary condition for indirect violations (*Rose, 1958*; *Heufer, 2009*). This fact is the foundation for the neuronal mechanism of utility maximization proposed in the Discussion.

Adherence to GARP demonstrates rationality in the technical economic sense (henceforth *technical rationality*). As discussed above, this form of rationality is based on choice consistency. Technical rationality stands in contrast to other forms of rationality (*Kacelnik, 2006*). These include *psychological rationality* which emphasizes the process by which decisions are made, namely logically consistent reasoning, rather than decision outcomes. There is also *biological rationality* which refers to decisions that are consistent with inclusive fitness. None of these forms of rationality are necessary prerequisites for any other. Of particular importance to our study is that demonstration of technical rationality neither presupposes nor establishes psychological rationality. A technically rational agent may or may not use reasoning to make choices.

Finally, tests of adherence to GARP have been utilized to quantify technical rationality. Although strict adherence to GARP is binary, *Afriat, 1972* proposed a measure, called Afriat's efficiency index (AEI), to quantify the severity of GARP violations. This measure has been utilized to quantify the extent of deficits in technical rationality in patients with damage to their ventromedial frontal lobe (*Camille et al., 2011*). It was also utilized to correlate mathematical ability with the degree of technical rationality in children (*Harbaugh et al., 2001*).

## GARP for worms

The foregoing examples illustrate that testing adherence to GARP entails the following prerequisites:

1. Two different goods to choose between, with more preferred to less for each good.
2. Measurement of the consumption choices.
3. Consumption trade-offs, such that consuming more of one good always means consuming less of the other.
4. Observation of decision outcomes on budget lines that intersect.

The first step in this study was to develop and validate the means to fulfill these prerequisites in ecologically realistic ways for *C. elegans*.

### Prerequisite (i): Two different goods

As goods, we used bacteria species having high (H, *Comamonas*) and medium (M, *Bacillus simplex*) quality as a food source, defined in terms of the growth rate of individual worms (*Avery and Shtonda, 2003*; *Shtonda, 2004*; *Shtonda and Avery, 2006*). Chemical analysis of volatile organic compounds

**Table 1.** Volatile organic compounds released by H and M food.

| Designation | CAS# | Compound | H food (DA1877) | M food (DA1885) |
|---|---|---|---|---|
| *a* | 1534-08-3 | Methylthioacetate* | 0.0 | 1.0 |
| *b* | 624-92-0 | Dimethyl disulfide* | 19.4 | 38.3 |
| *c* | 2432-51-1 | S-Methyl butanethioate† | 0.0 | 4.7 |
| *d* | 23747-45-7 | S-Methyl 3-methylbutanethioate | 0.0 | 2.4 |
| *e* | 3658-80-8 | Dimethyltrisulfide* | 1.5 | 6.4 |

Compounds were identified by gas chromatography-mass spectrometry of headspace of H and M food and confirmed with known standards. Amounts were inferred from the area under elution peaks, averaged across two replicates, and normalized to the amount of methylthioacetate in DA1885.

*Identified chemoattractant; other compounds are uncharacterized in chemotaxis assays.

†Vapor is nematicidal.

released by these bacteria revealed that each species emits a different blend of compounds (*Table 1*). These include three compounds that are unique to M food (*a*, *c*, *d*). One (*a*) is a chemoattractant *Hsueh et al., 2017*; two are uncharacterized in *C. elegans* chemotaxis assays (*c*, *d*). The two remaining compounds (*b*, *e*) are common to both species. Compound *b* was shown to be attractive in two out of three studies, otherwise neutral (*Hsueh et al., 2017*; *Worthy et al., 2018b*; *Choi et al., 2022*) and *e* is an attractant except at high concentrations (*Choi et al., 2022*). Compound *c* is nematicidal (*Xu et al., 2015*). Our chemical analysis establishes the olfactory substrate for qualitative differences in the worm's perception of H and M bacteria, analogous to the qualitative differences by which human choosers distinguish between goods (e.g. colors and flavors of fruits in *Figure 1*). We cannot, of course, conclude on this basis that the worm's nervous system actually distinguishes H from M; formally, the neurosensory representations of H and M could be similar. Further evidence on this point is presented below. It is notable that M food releases greater amounts of chemoattractants than H food does, yet is less attractive (*Figure 2C*, azide+). A possible explanation is that the nematicidal compound released by M food is a repellant that mitigates its potential to attract.

We chose to work with adult worms as they are easier to handle and count than the L1 larvae used in the original food-choice experiments (*Shtonda and Avery, 2006*). We were initially uncertain whether older worms could learn new food preferences, so we began by investigating the magnitude, neuronal dependence, and mechanisms of food quality learning in the developmental period spanning late L3 to young adulthood. Synchronized, late L3 worms (N2) were transferred to a training plate which contained an equal number of similar sized patches of H and M foods (*Figure 2A*, Trained). Preference for H versus M food was assessed the following day at the young adult stage on a test plate having a single pair of H and M food patches (henceforth, the *open-field accumulation assay*). Control worms were transferred to a mock training plate and tested in parallel with trained worms (*Figure 2A*, Untrained). Preference index *I* was quantified on a scale such that +1 and −1 represent absolute preference for H and M food, respectively; 0 represents indifference. Note that accumulation assays are not the same as the revealed preference assays used later in this study to investigate utility maximization. In particular, accumulation assays do not allow worms to consume mixtures of the goods (bundles) in the same feeding bout, nor do they challenge worms with different relative food densities (prices).

Trained N2 worms preferred H food to M food more strongly than untrained worms (*Figure 2B and N2*, Trained vs. Untrained, *Table 2*[1];) indicating a significant effect of food quality training. We conclude that worms in the developmental period under study can learn new food preferences; we refer to these as *trained preferences*. However, untrained N2 worms also preferred H to M food even though they were encountering these foods for the first time (*Figure 2B and N2* Untrained, $I > 0$, *Table 2*[2]). The preference for H food in untrained worms is interesting because hatchlings, which are eating food for the first time, show equal preference for these two foods (*Shtonda and Avery, 2006*). One possibility is that preferences in untrained worms are the result of learning over the 60 min experiment. Another possibility is that these preferences arise from events that occurred between hatching and testing; we refer to such preferences as *latent preferences*. To distinguish between these possibilities, we ran accumulation assays with and without the paralytic agent sodium

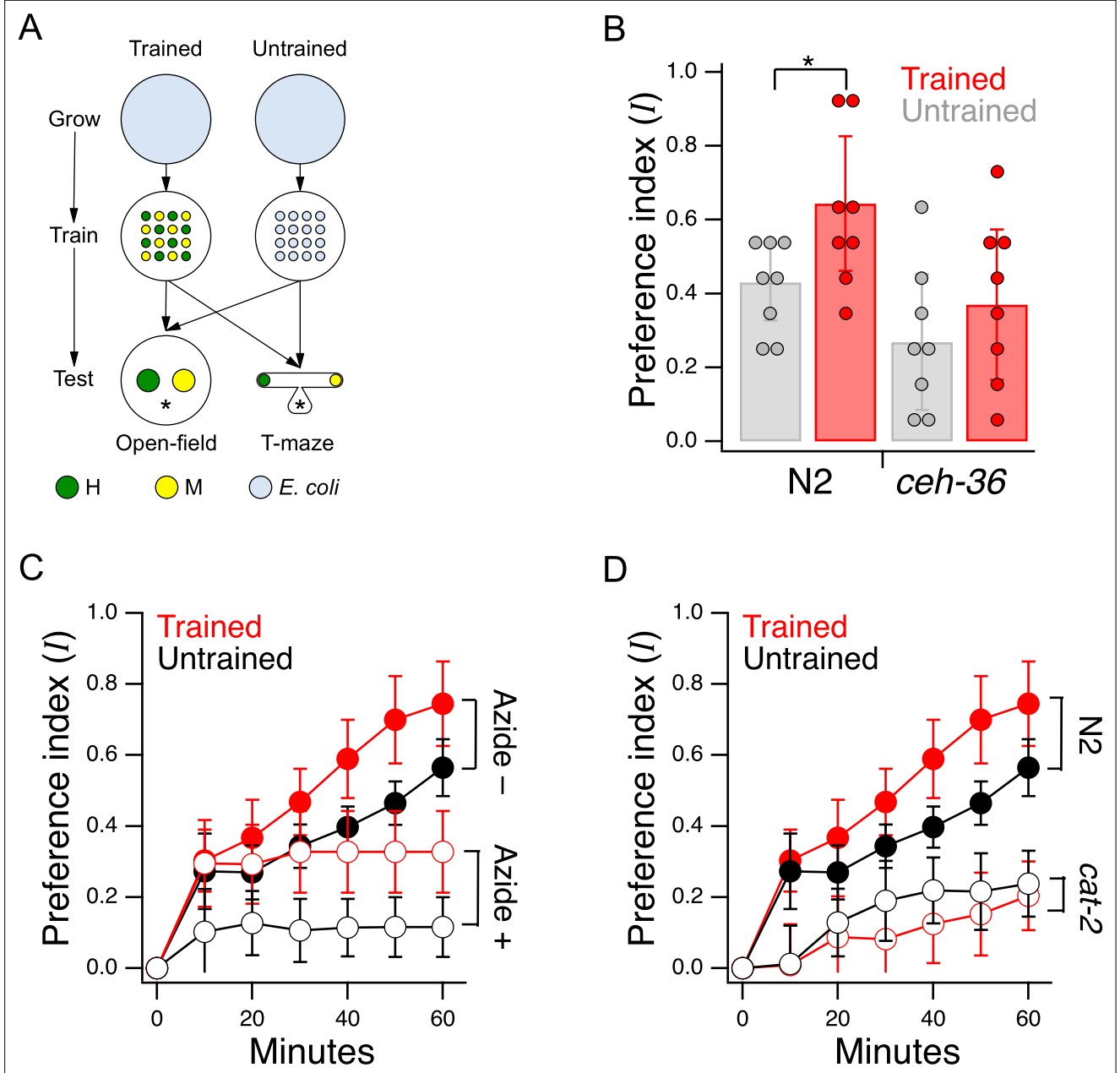

**Figure 2.** Edible bacteria act as goods over which worms form preferences through experience. (**A**) Food quality training and preference assays. *Filled circles* represent patches of bacteria as indicated in the key. *Stars* indicate worm starting locations. (**B**). Mean preference at $t = 60$ min. in the open-field accumulation assay for trained and untrained N2 and *ceh-36* mutants. *Asterisk*, see *Table 2*[1]. Replicates, Trained, strain($N$): N2(8), *ceh-36*(9). Replicates, Untrained, strain($N$): N2(8), *ceh-36*(8). (**C**). Mean preference vs. time for trained and untrained N2 worms in T-maze accumulation assays, with and without sodium azide in the food patches. (**D**). Mean preference index vs. time for trained and untrained *cat-2(tm2261)* mutants and N2 controls in T-maze accumulation assays. N2 data are from C. (**B–D**). Error bars, 95% CI. For sample size (N), statistical methods used, and significance level, see *Table 2*, rows 1-15.

The online version of this article includes the following source data and figure supplement(s) for figure 2:

**Source data 1.** Edible bacteria act as goods over which worms form preferences through experience.

**Source data 2.** Loss of dopamine signaling does not reduce proportion of time on food.

**Figure supplement 1.** T-maze diagram.

**Figure supplement 2.** Loss of dopamine signaling does not reduce proportion of time on food.

**Table 2.** Statistics.

Horizontal location of cell entries varies by row.

| Row | Figure | Test | Effect or comparison tested | Units of replication or sampling | Number of replicates or samples | Statistic | | DF 1 or combined DF | DF 2 | p | Effect size |
|---|---|---|---|---|---|---|---|---|---|---|---|
| 1 | 2B* | t-test | Trained N2 vs. Untrained N2 | Assay plates | N=8 | t | 2.50 | 14 | – | **2.56E-02** | 1.248 |
| 2 | 2B | t-test | N2 Untrained, 60 min mean I>0 | Assay plates | N=8 | t | 10.22 | 7 | – | **1.86E-05** | – |
| 3 | 2B | t-test | ceh-36 Untrained, 60 min mean I>0 | Assay plates | N=8 | t | 3.58 | 7 | – | **9.01E-03** | – |
| 4 | 2B | t-test | Trained ceh-36 vs. Untrained N2 | Assay plates | N=8 | t | 0.64 | 14 | – | 2.50E-01 | – |
| 5 | 2B | t-test | Untrained ceh-36 vs. Untrained N2 | Assay plates | N=8 | t | 1.87 | 14 | – | 8.22E-02 | – |
| 6 | 2B | t-test | Trained ceh-36 vs. Untrained ceh-36 | Assay plates | N=8 | t | 0.90 | 14 | – | 3.82E-01 | – |
| 7 | 2C | Two-factor ANOVA, repeated measures, main effect | Azide- Trained vs. Azide- Untrained | Assay plates | N≥9 / treatment | F | 11.28 | 1 | 21 | **2.98E-03** | 0.349 |
| 8 | 2C | t-test | Azide +Trained, 60 min mean I>0 | Assay plates | N=10 | t | 6.35 | 10 | – | **8.36E-05** | – |
| 9 | 2C | t-test | Azide+, Untrained, 60 min mean I>0 | Assay plates | N=10 | t | 3.09 | 10 | – | **1.15E-02** | – |
| 10 | 2C | Two-factor ANOVA, repeated measures, main effect | Azide +Trained vs. Azide +Untrained | Assay plates | N=10 / treatment | F | 10.32 | 1 | 20 | **4.37E-03** | 0.340 |
| 11 | 2C | Two-factor ANOVA, repeated measures, main effect | Azide +Trained, vs. Azide– Trained | Assay plates | N≥9 / treatment | F | 11.17 | 1 | 19 | **3.43E-03** | 0.370 |
| 12 | 2C | Two-factor ANOVA, repeated measures, main effect | Azide +Untrained vs. Azide– Untrained | Assay plates | N≥10 / treatment | F | 38.28 | 1 | 22 | **3.16E-06** | 0.635 |
| 13 | 2D | Two-factor ANOVA, repeated measures, main effect | Untrained N2 vs. Untrained cat-2 | Assay plates | N≥9 / strain | F | 23.25 | 1 | 21 | **9.14E-05** | 0.493 |
| 14 | 2D | Two-factor ANOVA, repeated measures, main effect | Trained N2 vs. Trained cat-2 | Assay plates | N=9 / strain | F | 52.50 | 1 | 18 | **9.74E-07** | 0.207 |
| 15 | 2D | Two-factor ANOVA, repeated measures, main effect | Trained cat-2 vs. Untrained cat-2 | Assay plates | N=9 / treatment | F | 0.90 | 1 | 18 | 6.43E-01 | – |
| 16 | 4A | Two-factor ANOVA, main effect | Familiar vs. Unfamiliar | Worms | N≥19 / treatment | F | 10.46 | 1 | 54 | **2.10E-03** | 0.162 |
| 17 | 4 A* | t-test | Unfamiliar, grown in H vs. grown in M | Worms | N≥6 | t | 2.65 | 17 | – | **2.10E-02** | 0.876 |
| 18 | 4B | t-test | Trained, f_H>0.5 | Worms | N=19 | t | 8.60 | 18 | – | **8.60E-08** | – |
| 19 | 4B | t-test | Untrained, f_H>0.5 | Worms | N=28 | t | 4.35 | 27 | – | **1.70E-04** | – |
| 20 | 4B* | t-test | Trained vs. Untrained | Worms | N≥19 | t | 2.95 | 44 | – | **5.11E-03** | 0.850 |
| 21 | 4E | Two-factor ANOVA | Main effect of optical density | Worms | N≥22 / density | F | 31.58 | 3 | 108 | **9.80E-15** | 0.467 |
| 22 | 4F | Two-factor ANOVA | Main effect of optical density | Worms | N≥22 / density | F | 3.10 | 3 | 106 | **3.00E-02** | 0.081 |

*Table 2 continued on next page*

*Table 2 continued*

| Row | Figure | Test | Effect or comparison tested | Units of replication or sampling | Number of replicates or samples | Statistic | | DF 1 or combined DF | DF 2 | p | Effect size |
|---|---|---|---|---|---|---|---|---|---|---|---|
| 23 | 5B | Two-factor ANOVA | Main effect of price ratio | Worms | Avg N=15 / ratio | F | 44.13 | 6 | 195 | **7.89E-34** | 0.576 |
| 24 | 5B | Two-factor ANOVA | Main effect of training | Worms | Avg N=15 / ratio | F | 36.16 | 1 | 195 | **8.82E-09** | 0.156 |
| 25 | 5B | t-test | Trained, point *a*, mean $f\_H<0.5$ | Worms | N=9 | t | 6.22 | 8 | – | **2.52E-04** | – |
| 26 | 5B | t-test | Untrained, point *a*, mean $f\_H<0.5$ | Worms | N=12 | t | 8.29 | 11 | – | **4.66E-06** | – |
| 27 | 6A | Regression with replication slope test | *Figure 6A*, points *abd*, Trained slope ≠ 0 | Worms | N≥9 / ratio | F | 118.79 | 1 | 47 | **1.85E-14** | – |
| 28 | 6A | Regression with replication slope test | *Figure 6A*, points *abd*, Untrained slope ≠ 0 | Worms | N≥10 / ratio | F | 28.52 | 1 | 39 | **4.26E-06** | – |
| 29 | 6A | Regression with replication slope test | *Figure 6A*, points *cef*, Trained slope ≠ 0 | Worms | N≥10 / ratio | F | 20.29 | 1 | 46 | **4.54E-05** | – |
| 30 | 6A | Regression with replication slope test | *Figure 6A*, points *cef*, Untrained slope ≠ 0 | Worms | N≥14 / ratio | F | 26.56 | 1 | 49 | **4.55E-06** | – |
| 31 | 6A | Regression with replication slope test | *Figure 6A*, points *deg*, Trained slope ≠ 0 | Worms | N≥6 / ratio | F | 2.34 | 1 | 51 | 1.32E-01 | – |
| 32 | 6A | Regression with replication slope test | *Figure 6A*, points *deg*, Untrained slope ≠ 0 | Worms | N≥7 / ratio | F | 7.82 | 1 | 45 | **7.56E-03** | – |
| 33 | 7A | Linear correlation | Frequency ratio vs. $f\_H$ | Worms | N=142 | t | 6.64 | 141 | – | **6.53E-10** | – |
| 34 | 7B | Linear correlation | Dwell time ratio vs. $f\_H$ | Worms | N=203 | t | 39.60 | 202 | – | **6.95E-97** | – |
| 35 | 7C | Linear correlation | Dwell time ratio vs. mean head angle | Worms | N=203 | t | 35.96 | 202 | – | **2.62E-89** | – |
| 36 | 7D | t-test | *ceh-36* Untrained, $f\_H>0.5$ | Worms | N=11 | t | 4.20 | 10 | – | **1.84E-03** | – |
| 37 | 7D | Two-factor ANOVA | Treatment ×Strain interaction | Worms | N≥7 / treatment | F | 5.03 | 1 | 62 | **2.85E-02** | 0.075 |
| 38 | 7D* | t-test | N2, Trained vs. Untrained | Worms | N≥20 | t | 3.45 | 46 | – | **1.21E-03** | 0.186 |
| 39 | 7D | t-test | *ceh-36*, Trained vs. Untrained | Worms | N=7 / treatment | t | 0.58 | 16 | – | 5.83E-01 | – |
| 40 | 8B | Two-factor ANOVA | Main effect of food type | Worms | N≥6 / treatment | F | 3.56 | 1 | 23 | 7.20E-02 | – |
| 41 | 8D | Two-factor ANOVA | Main effect of food type | Worms | N≥7 / treatment | F | 18.42 | 1 | 25 | **2.00E-04** | 0.424 |
| 42 | 8 C* | t-test | Peak response, Untrained, H vs. M food | Worms | N≥7 / treatment | t | 2.98 | 12 | – | **1.30E-02** | 0.913 |
| 43 | 8 C× | t-test | Peak response, Trained, H vs. M food | Worms | N≥7 / treatment | t | 2.29 | 13 | – | **3.96E-02** | 1.184 |
| 44 | 8B | Two-factor ANOVA | Main effect of training | Worms | N≥6 / treatment | F | 0.00 | 1 | 23 | 9.90E-01 | – |

*Table 2 continued on next page*

*Table 2 continued*

| Row | Figure | Test | Effect or comparison tested | Units of replication or sampling | Number of replicates or samples | Statistic | | DF 1 or combined DF | DF 2 | p | Effect size |
|-----|--------|------|------------------------------|----------------------------------|----------------------------------|-----------|---|---------------------|------|---|-------------|
| 45 | 8D | Two-factor ANOVA | Main effect of training | Worms | N=7 / treatment | F | 7.52 | 1 | 25 | **1.10E-02** | 0.883 |
| 46 | 8D* | t-test | H food, Trained vs. Untrained | Worms | N=7 / treatment | t | 2.86 | 12 | – | **1.44E-02** | 1.528 |
| 47 | 8D | t-test | M food, Trained vs. Untrained | Worms | N≥7 / treatment | t | 0.80 | 13 | – | 4.39E-01 | – |
| 48 | 9 A* | t-test | H → M, peak response, Trained vs. Untrained | Worms | N≥14 / treatment | t | 2.66 | 28 | – | **1.29E-02** | 0.972 |
| 49 | 9A | t-test | M → H, area under the curve, Trained vs. Untrained | Worms | N≥15 / treatment | t | 0.74 | 29 | – | 4.67E-01 | – |
| 51 | 9C | Linear correlation | AWC activation vs. utility, Trained | Worms | N≥6 / mean | t | 0.10 | 5 | – | 9.27E-01 | – |
| 52 | 9C | Linear correlation | AWC activation vs. utility, Untrained | Worms | N≥8 / mean | t | 0.17 | 5 | – | 8.75E-01 | – |
| 53 | 9D | Linear correlation | AWC activation vs. preference, Trained | Worms | N≥6 / mean | t | 1.57 | 5 | – | 1.77E-01 | – |
| 54 | 9D | Linear correlation | AWC activation vs. preference, Untrained | Worms | N≥8 / mean | t | 0.44 | 5 | – | 6.78E-01 | – |
| 55 | 9C | t-test | Trained, point *e* vs. Untrained, point *d* | Worms | N≥8 / mean | t | 2.14 | 25 | – | **4.23E-02** | 0.889 |

p-values associated with significant results are shown in bold font. Sample size was determined by increasing the number of biological replicates until the coefficient of variation for each means converged. Each experiment was performed once, with the indicated number of biological replicates; the non-stationary nature of the organism precluded technical replicates. No data were censored or excluded.

azide in each food patch. The presence of sodium azide captures the preference of worms upon first approach to food. We found that trained and untrained worms preferred H food even when sodium azide was present (*Figure 2C*, *Table 2*[8,9]), albeit to a lesser degree than in the absence of sodium azide (*Figure 2C*, *Table 2*[11,12]), and that the effect of training was evident in both groups (*Figure 2C*, *Table 2*[7,10]). We draw three main conclusions from this experiment: (i) worms do not need to sample both patches before expressing a preference, (ii) training increases preference upon first approach, indicative of a change in olfactory sensitivity, and (iii) latent preferences exist and may contribute to overall preference.

*C. elegans* has 12 pairs of anterior chemosensory neurons that respond to bacteria conditioned medium (*Zaslaver et al., 2015*), acting either as on-cells (activated by onset), or off-cells (activated by offset). As a first step in identifying the locus of food quality learning, we measured food preferences in worms with a loss of function mutation in the Otx homeobox gene *ceh-36*. This gene is expressed specifically in two food-sensitive chemosensory neuron pairs, AWC and ASE, where it is required for normal expression levels of functionally essential genes, including chemoreceptors and ion channel subunits required for chemotransduction (*Lanjuin et al., 2003*; *Koga and Ohshima, 2004*). We found that *ceh-36* worms were nevertheless able to distinguish H from M food, as even untrained worms exhibited a marked preference for H food (*Figure 2B*, *ceh-36*, Untrained vs. $I = 0$, *Table 2*[3]). Moreover, this level of preference was indistinguishable from that exhibited by untrained N2 worms (*Figure 2B*, Trained or Untrained *ceh-36* vs. Untrained N2, *Table 2*[4,5]). In contrast, we were unable to detect an effect of training on food preference in *ceh-36* worms (*Figure 2B*, *ceh-36*, Trained vs. Untrained, *Table 2*[6]). The effect size associated with food quality training was reduced relative to wild type (Cohen's d: N2, 0.21; *ceh-36*=0.10). This drop was attributable to a reduction in the difference between means (N2, 1.25; *ceh-36*, 0.45) and an increase in the pooled standard deviation (N2, 0.17; *ceh-36*, 0.22). We conclude that training effects in *ceh-36*, if present, are probably weaker than in N2. Taken together, these results show that undiminished *ceh-36* function is likely required for

food preference acquired through food quality training, implicating AWC and/or ASE in this process; however, full *ceh-36* function is dispensable for latent food preferences, suggesting the other neurons may subserve this behavior. Below we present functional imaging data consistent with a role for AWC neurons in mediating the behavior effects of food quality training.

We next considered the mechanism of accumulation in food patches and how it may be altered by food quality training. The number of worms in a food patch depends on between patch entry and exit rates. In a simple experiment to study the effects of entry rate on preference index, we added a fast-acting metabolic poison (sodium azide) to each food patch to prevent worms from leaving (*Choi et al., 2016*). To increase the resolution of our preference measurements, we used a T-maze baited with H and M foods (*Figure 2—figure supplement 1*); the maze prevents worms from wandering out of range of food spots.

Trained N2 worms tested in the absence of sodium azide preferred H food to M food more strongly than untrained worms (*Figure 2C*, Azide –, Trained vs. Untrained, *Table 2*[7]) indicating a significant effect of food quality training. Preference in both groups continued to rise after the 40-min sample point. This rise could be refinement of preferences as worms go back and forth between patches, additional food-quality learning, or both. In the presence of sodium azide, trained and untrained worms still accumulated more strongly in H food than in M food (*Figure 2C*: Azide+, Trained and Untrained vs. $I = 0$, *Table 2*[8,9]). This finding shows that preferences can be established on the basis of entry rate alone. Furthermore, it shows that worms do not have to sample both patches before expressing a preference. This means at least some of the H-food preference seen in untrained worms is independent of sampling experience during the assay.

We also observed a significant effect of training on food preference in the azide condition (*Figure 2C*, Azide+, Trained vs. Untrained, *Table 2*[10]), indicating that food quality training increases entry rate. Finally, preference levels were substantially reduced by sodium azide (*Figure 2C*, Trained, Azide +vs. Azide–; *Table 2*[11] and Untrained, Azide +vs. Azide–; *Table 2*[12]). This result shows that additional mechanisms contribute to differential accumulation in H and M food, mostly likely differences in exit rate, shown previously to contribute to accumulation in open-field accumulation assays (*Shtonda and Avery, 2006*).

Learning to avoid the odors of pathogenic bacteria is reinforced by serotonin (*Zhang et al., 2005*), but less is known about how preferences for nonpathogenic foods are reinforced. We found that food preferences in general were reduced in *cat-2* mutants, which have substantially reduced levels of dopamine (*Lints and Emmons, 1999*; *Sawin et al., 2000*; *Calvo et al., 2011*). Preference for H food in untrained *cat-2* mutants was lower than in untrained N2 (*Figure 2D*, Untrained, N2 vs. *cat-2*, *Table 2*[13]) indicating impaired acquisition of latent preferences. Preference for H food in trained *cat-2* mutants was lower than in trained N2 worms (*Figure 2D*, Trained, N2 vs. *cat-2*, *Table 2*[14]), indicating an impairment in trained preferences. Finally, preferences in trained and untrained *cat-2* mutants were indistinguishable (*Figure 2D cat-2*, Trained vs. Untrained, *Table 2*[15]), another indication that trained preferences were impaired.

What accounts for these impairments? Worms move more slowly when in contact with food particles, an effect caused by mechanical activation of dopaminergic neurons (*Sawin et al., 2000*; *Tanimoto et al., 2016*). Slowing is reduced in *cat-2* mutants (*Sawin et al., 2000*; *Cermak et al., 2020*). During the training procedure, this reduction could cause *cat-2* mutants to exit food patches sooner than wild type worms. It is possible, therefore, that food quality learning in *cat-2* mutants is impaired simply because they spend less time in the food, hence have less experience of it. However, we found no differences between wild type and *cat-2* mutants in the proportion of time on food (*Figure 2—figure supplement 2*). This finding points to a requirement for dopamine signaling in the acquisition or expression of food memory, in accordance with substantial evidence showing a requirement for dopamine in several forms of associative learning in *C. elegans* (*Hukema et al., 2008*; *Voglis and Tavernarakis, 2008*; *Lee et al., 2009*; *Musselman et al., 2012*).

## GARP prerequisite (ii): Measurement of the consumption choices

Bacteria are ingested via the worm's pharynx, a rhythmically active, muscular pump comprising the animal's throat. Each pharyngeal contraction is called a 'pump'. To measure the relationship between price and consumption, we developed a system for presenting single worms with a pair of bacteria suspensions while recording the number of pumps the worm 'spends' on each (*Figure 3*). The system

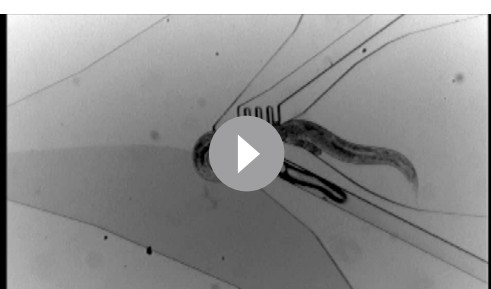

**Video 1.** Foraging behavior in the Y-chip. Simulated Y-chip experiment. The worm is held at its midsection by a vacuum activated clamp, leaving the head (left) and tail free to move. The both fluid streams contain bacteria-free buffer, flowing to the right to left. Food dye was added to the lower stream to visualize the interface between streams. Bubbles originating at the clamp are formed by air that has been pulled through the PDMS walls of the chip by the vacuum. The worm prefers the dyed stream as it contains potassium sorbate, which acts as a chemoattractant.

https://elifesciences.org/articles/69779/figures#video1

is based on a microfluidic chip (called the 'Y-chip') originally designed to investigate the neural mechanism of klinotaxis (*McCormick et al., 2011*), a common form of chemotaxis. *C. elegans* klinotaxis takes the form of accentuating or attenuating, respectively, locomotory head bends toward or away from attractive tastes and odors, including food (*Iino and Yoshida, 2009*). The Y-chip restrains the worm at the border between two streams of bacteria suspension (*Figure 3A and B*), representing contiguous patches of food as might occur in the natural environment (*Frézal and Félix, 2015*). Restraint is achieved by means of a vacuum clamp that leaves the worm's head, upper body, and tail free to move. The worm's head alternates between the two streams, making sinusoidal movements that resemble crawling on a standard agarose substrate in form and frequency (*Video 1*).

Large movements of the worm's head in the Y-chip made it impractical to count accurately the number of pumps in a feeding bout by optical methods (*Fang-Yen et al., 2009*; *Scholz et al., 2016*). Instead, we counted pumps by recording each worm's electropharyngeogram (*Raizen and Avery, 1994*) via electrodes inserted into the chip (*Faumont et al., 2012*). Despite movements of the worm's body, normal looking EPGs were obtained, with readily identifiable muscle excitation spikes (E) and relaxation spikes (R) (*Figure 3D*). We quantified consumption of H and M food in terms of the number of pumps that occurred in each food during a 12-min exposure to particular food offerings. This time limit was chosen as compromise between the need acquire reliable preference measures without allowing the worm to feed so long as to allow preferences to changes, for example, by onset of satiety.

We measured the consumption of each food in terms of the fraction of pumps that occurred while the worm's head was in that food's stream. In *C. elegans* feeding decisions, the muscular energy utilized while feeding, can be thought of as the functional equivalent of money, in human budgetary experiments. Following this logic, price $P$ can be defined as the cost incurred by swallowing the amount of bacteria ingested in a single pump. We assume that at equal density, approximately the same number of H and M bacteria cells are ingested per pump. This assumption is based on: (i) the observation that H and M bacteria are similar in size (*Avery and Shtonda, 2003*), and (ii) the likelihood that pump volume $v$ is approximately constant. We presume pump volume depends mainly on the maximum extent of contraction of pharyngeal muscles, which opens the pharyngeal lumen. This, in turn, should be most strongly influenced by the duration of the pharyngeal action potential, which is constant across the food densities utilized in our study (*Lee et al., 2017*). Assuming constant pump volume,

$$\text{cells/pump} = \nu\,[\text{nL/pump}]\,d\,[\text{cells/nL}], \tag{1}$$

where $d$ is the density of bacteria cells. Taking the inverse of cells/pump, and expressing volume in units of the volume of the fully-open pharyngeal lumen ($v = 1$) we can define a measure of price for bacteria X,

$$P_X(d_X) = 1/d_X\,[\text{pumps/cell}]. \tag{2}$$

*Equation 2* embodies the intuition that doubling bacteria density reduces the energetic cost of consumption by half.

Finally, consumption, $Q_X$, is the number of pumps in food X, $n_X$, times the number of cells/pump (*Equation 1*),

$$Q_X(n_X, d_X) = n_X d_X \text{ [cells]} \tag{3}$$

Our method of measuring consumption choices differs in operational terms from the method used in many experimental economics GARP studies. Participants in the experiment make instantaneous choices from a set of menu items. This type of experiment abstracts away from the classical revealed preference theory on which these experiments are based (*Samuelson, 1938*; *Houthakker, 1950*). In the classical theory, and essentially all economy-wide measurements that rest on it, consumption is defined in terms of the rate at which units of goods are acquired or consumed over time. In our study, we employ this more classical approach. Here, consumption is the product of the number of pumps and food density (*Equation 3*). The number of pumps is, in effect, the time integral of pumping rate. In other words, it is the integral of consumption rate. As such, it captures a key feature of the classical presentation missed by many in-lab experimental economics studies.

## GARP prerequisite (iii): consumption trade-offs

In GARP experiments with human participants, trade-offs between goods within a choice set are established by setting a fixed budget which each bundle must stay within. Given the assumption that more is preferred to less, the budget constraint will be satisfied by establishment of equality between expenditure and budget,

$$P_X Q_X + P_Y Q_Y = E \text{ [dollars]}, \tag{4}$$

where the left hand side is total expenditure and $E$ is the budget. Participants are not made aware of the budget constraint; it remains implicit in the set of available choices. For worms, the analogous equality is

$$P_H Q_H + P_M Q_M = N \text{ [pumps]}. \tag{5}$$

where $N$ is the total number of pumps ($N = n_H + n_M$). In human experiments the trade-off between consuming more of one good at the expense of less of another is enforced by the budget constraint. In *C. elegans* experiments, however, imposing a budget constraint is not necessary, as trade-offs are enforced by the Y-chip. The chip contains only two streams, one for H and one for M food, such that for every pump spent on H food, the worm necessarily forgoes a pump spent on M food, and *vice versa*.

As the total number of pumps emitted by individual worms over a fixed observation period was quite variable, and the observation period had to be brief to limit possible consumption changes due to satiety, we found it expedient to express consumption in terms of the proportion of pumps spent on each food, rather than number of pumps spent on each food, with

$$q_H(n_H, d_H) = \frac{n_H}{N} d_H$$

$$q_M(n_M, d_M) = \frac{n_M}{N} d_M \tag{6}$$

where $q_H$ and $q_M$ represent consumption in proportional terms. The corresponding equality is

$$P_H q_H + P_M q_M = 1 \tag{7}$$

Below, in the section Behavioral mechanisms of utility maximization, we show that worms pump at nearly the same rate in H and M food. Therefore, consumption is determined mainly by the fraction of time the worm pumps in each food. Note that by expressing consumption of a food as the fraction of total pumps spent in that food, rather than as the number of pumps spent on that food, the number of pumps $N$ (or equivalently the total time allowed for feeding) cancels out. This has the advantage that we do not have to assume that worms are aware of a pump or time budget.

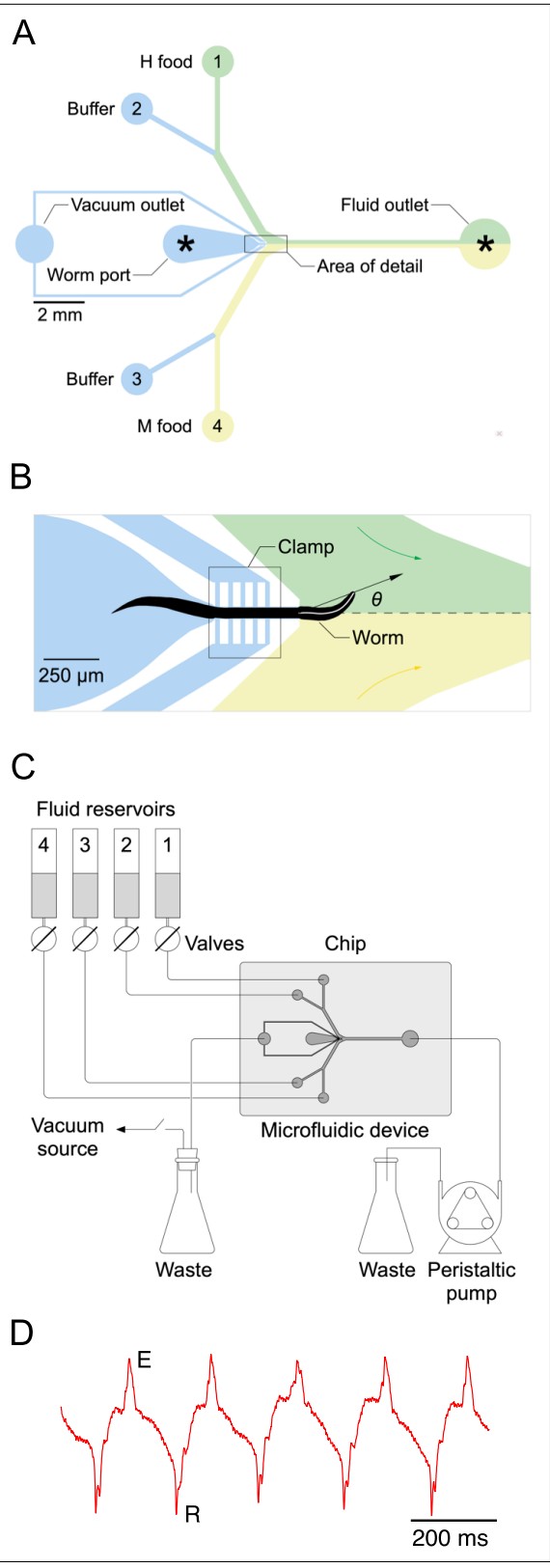

**Figure 3.** Single-worm food choice assays. (**A**) Layout of the Y-chip. *Asterisks* indicate the position of recording electrodes. Ground electrodes (not shown) were inserted into the food and buffer ports to reduce electrical interference. The chip is shown configured for the experiments in ***Figures 4B and 5B***. (**B**). Area of detail shown in A. The dashed line is the centerline of chip; the white line within the worm is its centerline. The black arrow

*Figure 3 continued on next page*

*Figure 3 continued*

connects the middle of the neck where it enters the food channel with anterior end of worm's centerline. Positive values of head angle ($\theta$) indicate displacement toward H food. Colored arrows show direction of flow. (**C**). Schematic overview of fluidic system. (**D**). Typical electropharyngeogram. Each pair of excitation (**E**) and relaxation (**R**) spikes constitutes one pharyngeal pump.

## GARP prerequisite (iv): observation of decision outcomes from intersecting budget lines

By systematically altering the densities of H and M bacteria in the Y-chip, we could alter their relative prices. Changing the density of both foods at the same time allowed us to create changes analogous to increases or decreases in the total consumption budget, which shifts the budget line in or out relative to the origin. This approach allowed us to create a series of intersecting choice sets.

## Basic feeding decisions are preserved in Y-chip

A concern at the outset was the possibility that feeding in the Y-chip is not representative of feeding under standard laboratory conditions such as on an agar substrate or in liquid culture. For example, the vacuum clamp likely stimulates the worm mechanically, which can inhibit feeding (*Keane and Avery, 2003*). It was necessary, therefore, to assess the degree to which feeding behavior in the chip is normal. We did this by comparing the worm's choices of what to eat and how avidly to eat it on agar plates and in the Y-chip.

## Food familiarity effect

*C. elegans* is reluctant to eat unfamiliar food. It pumps slower in unfamiliar food than in familiar food (*Song et al., 2013*). This effect is the result of feeding suppression triggered by the taste or smell of unfamiliar bacteria. To test for this effect in the Y-chip, we grew worms on H food or M food until young adulthood and measured pumping rates either in the same (familiar) or the converse (unfamiliar) food. In this experiment, both streams in the chip carried the same food. We found that mean pumping rate for a given type of food was lower when that food was unfamiliar, indicating that the food familiarity effect is intact in the Y-chip (*Figure 4A*, Familiar food vs. Unfamiliar food, *Table 2*[16]). Furthermore, we noted that pumping rate on familiar food was the same in the two cases. This allowed us to compare directly the extent to which unfamiliar H and M food suppressed pumping rate. We found that suppression was greater when the unfamiliar food was lower in quality than the familiar food (*Figure 4A*, Unfamiliar food, grown on H vs. grown on M, *Table 2*[17]). This result is consistent with a model in which worms are even more reluctant to feed on unfamiliar food when it is worse than what they have eaten in the recent past. A similar result has been seen in the case of food-patch leaving behavior (*Shtonda and Avery, 2006*).

The food familiarity effect supports a model in which H and M foods are qualitatively distinct to the worm. In the alternative model, the foods are qualitatively similar but one food generates a more intense perception than the other, perhaps by emitting more of the characteristic compound or compounds by which H and M are detected. Worms familiar with H or M food pump at the same rate in them (*Figure 4A*, left bars), indicating H and M are not distinct. By the alternative model, this means H and M emit the same amount of the characteristic compounds. In light of this result, the alternative model obviously fails to predict the familiarity effect for, as H and M are indistinguishable, neither of them can be unfamiliar to the worm under the conditions of our experiment. From the fact that worms pump at different rates on H and M food when they are familiar versus unfamiliar, we can conclude that the foods are qualitatively distinct.

## Food quality training

We tested groups of trained and untrained worms with H and M food at equal concentrations (OD 1.0) in their respective streams in the Y-chip. Preference, as indicated by the fraction of pumps in H food ($f_H$, see Materials and methods), in trained and untrained worms was greater than 0.5, indicating that both groups preferred H food in the chip, just as they do in accumulation assays (*Figure 4B*, *Table 2*[18,19]). Moreover, we found that this preference was enhanced by training, again consistent with

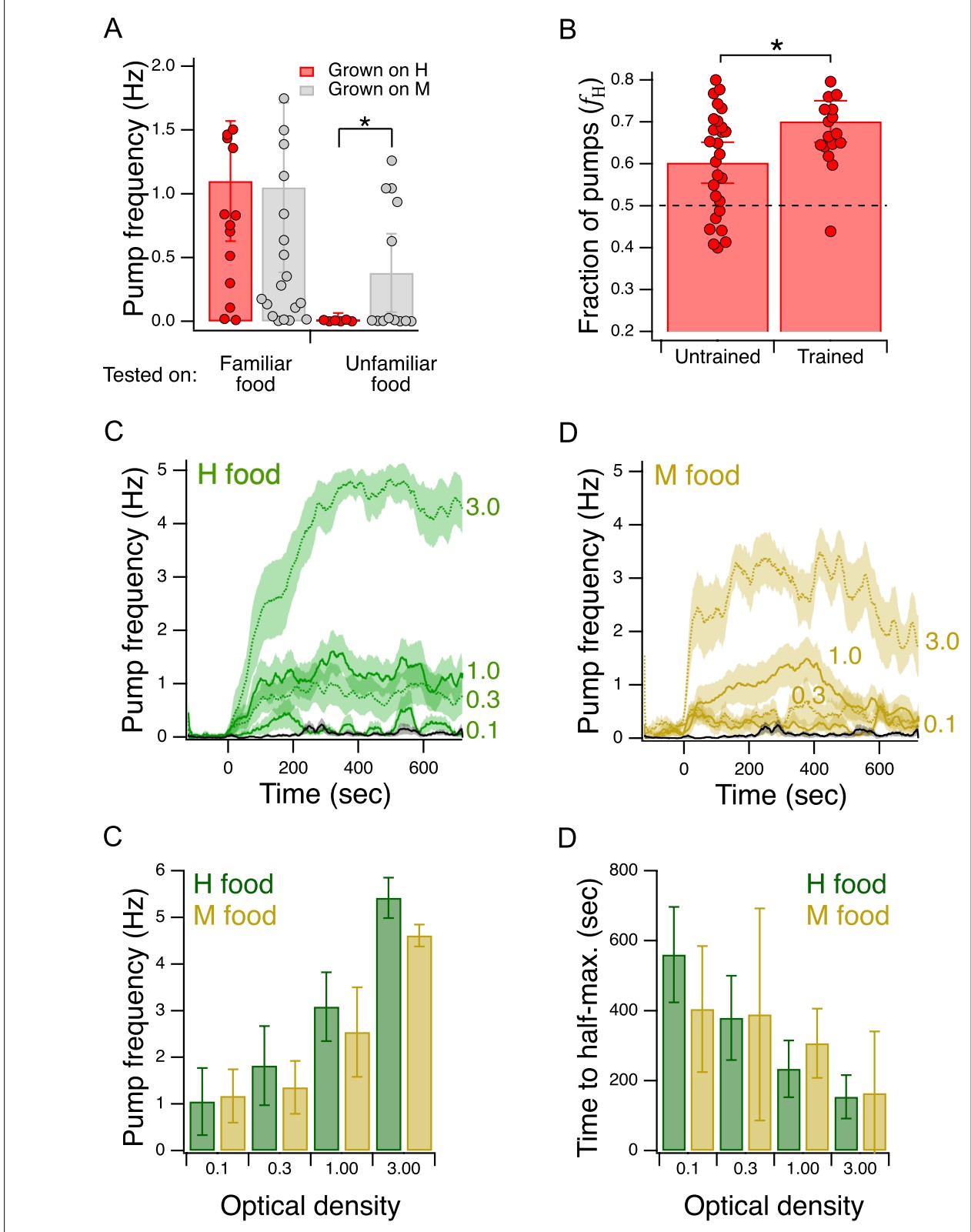

**Figure 4.** Validation of the Y-chip for measuring food preferences. (**A**) Familiar food effect. Mean pump frequency of worms grown on H or M food and tested on the same or the converse food. *Asterisk*, see *Table 2*[17]. Both foods were at OD 1. Pumping was recorded for 12 min. Replicates, Grown on H, *tested on* (*N*): *Familiar*(16), *Unfamiliar*(6). Replicates, Grown on M, *tested on*(*N*): *Familiar*(23), *Unfamiliar*(13). Error bars, 95% CI. (**B**). Food quality learning. Mean fraction of pumps in H food in trained and untrained worms. *Asterisk*, see *Table 2*[20]. The *dashed line* indicates equal preference for

*Figure 4 continued on next page*

Figure 4 continued

H and M food. Both foods were at OD 1. Error bars, 95% CI. (**C,D**). Time course of pump frequency at four different densities of familiar food. Food enters the chip at $t = 0$ sec. Optical density is indicated next to each trace. The *black trace* shows pumping in the absence of food. Shading, ± SEM E. Dependence of mean peak pump frequency density of familiar food. (**F**). Dependence of latency to half-maximal pump frequency on density of familiar food. (**E,F**). Error bars, 95% CI. For sample size (*N*), statistical methods used, and significance level, see *Table 2*, rows 16-000.

The online version of this article includes the following source data for figure 4:

**Source data 1.** Validation of the Y-chip for measuring food preferences.

accumulation assays (*Figure 4B*, Trained vs. Untrained, *Table 2*[20]). We conclude that the effects of food quality training are detectable in the Y-chip.

## Effect of food density on pumping rate

Although there appear to be no systematic studies of this effect when worms are feeding on bacteria lawns in petri plates, pumping rate has been shown to increase as a function of food density in liquid culture (*Avery and Horvitz, 1990*). This effect has also been demonstrated under conditions of mild restraint in microfluidic devices (*Scholz et al., 2016*; *Lee et al., 2017*; *Weeks et al., 2018*). To test for this effect in the Y-chip, we trained worms as in *Figure 2A*, except that the training plate contained a single food, H or M. During testing, both channels in the Y-chip carried the food on which the animals were trained (H or M) at an OD of 0.1, 0.3, 1.0, or 3.0. Pumping rate in H food was stable whereas pumping rate in M appeared to decline later in the experiment (*Figure 4C and D*); therefore, we quantified pumping in terms of its peak rate for both food types. Peak pumping rate was comparable to the rate recorded in similar densities of *E. coli* strain OP50 under mild restraint in microfluidic devices (*Scholz et al., 2016*; *Lee et al., 2017*; *Weeks et al., 2018*), and exhibited the expected increase with food density (*Figure 4E*; effect of OD, *Table 2*[21]). We conclude that the density dependence of pumping rate is intact in the Y-chip. This experiment also revealed previously unreported aspects of pumping kinetics. Regardless of food type, pumping rate rose slowly, on the time scale of 100 s of seconds (*Figure 4C and D*), Additionally, we observed an inverse relationship between the latency to half-maximum pumping rate and concentration (*Figure 4F*; effect of OD, *Table 2*[22]). Thus, worms encountering a richer food source eat sooner at higher rates, a coordinated response that is presumably adaptive in natural environments.

## Demand curves

We next turned to the question of whether *C. elegans* feeding behavior is altered by the relative price of food options that vary in quality. Economists identify several different types of goods according to how demand (or equivalently, consumption) is affected by changes in income or price. An *ordinary good* is one for which there is an inverse relationship between price and demand. To determine whether H and M food behave as ordinary goods in *C. elegans* feeding ecology, we constructed *demand curves*, in which consumption was plotted against pricefor H and M food (*Figure 5A*). We found an inverse relationship between consumption and price, indicating that H and M food act as normal goods and therefore are well suited to a GARP experiment.

More broadly, these results show that *C. elegans* obeys the classic law of demand, exhibiting the fundamental sensitivity of consumption to price seen in humans. The data of *Figure 5A* were well fit by

$$q_X(p_X) = A p_X{}^\epsilon \tag{8}$$

where $A$ is a positive constant and $\epsilon < 0$. This is the equation for a demand curve in which the percentage change in consumption from a given percentage change in price is constant at all prices, that is, there is constant *elasticity of demand* (*Varian, 1992*). We found $\epsilon \cong -2$ for both food types, indicating strong elasticity, a condition that typically arises when substitute goods of similar value are available. Interestingly, this may actually be the case for *C. elegans*, which grows robustly on approximately 80% of the hundreds of bacteria species in its natural habitat (*Samuel et al., 2016*). Elasticity is not always the case in foraging animals. Rats exhibit inelastic demand when offered essential commodities such as food pellets and water (*Kagel et al., 1975*; *Kagel et al., 1981*).

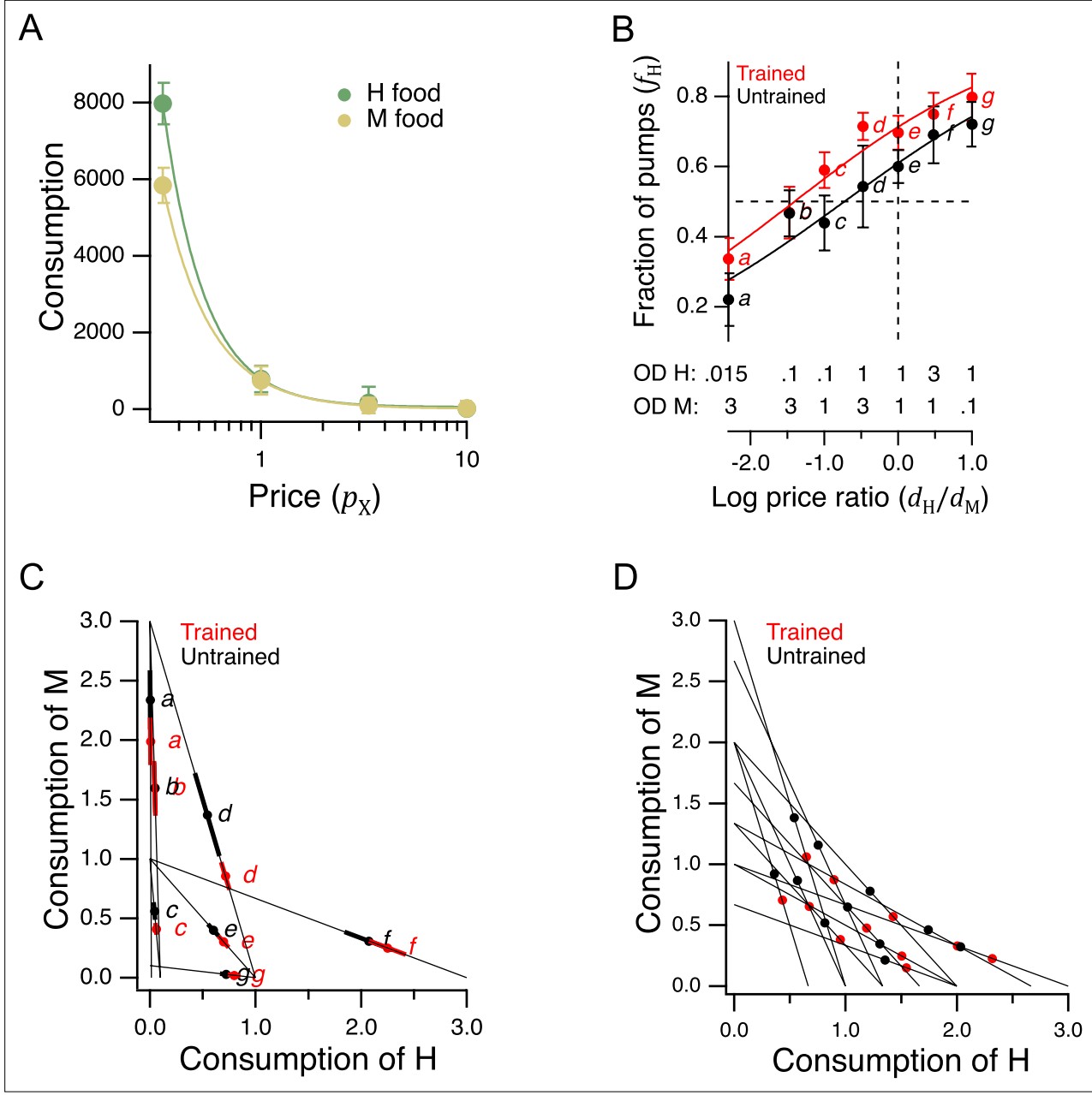

**Figure 5.** Economic analysis of food choice in *C. elegans*. (**A**) Demand curves. Mean consumption of familiar food versus its price. Consumption is computed as number of pumps times optical density of bacteria. Price is computed by *Equation 2*. The data are fit by *Equation 8* with $\epsilon = -2.2$ for H food and $\epsilon = -1.9$ for M food. H food price(*N*): 0.33(13), 1.0 (16), 3.3 (21), 10(15). M food price(*N*): 0.33(9), 1.0 (23), 3.3 (7), 10(10). (**B**). Price ratio curves. Food preference, measured as fraction of pumps in H food, versus price ratio for Trained and Untrained worms. *Horizontal dashed line*: indifference between H and M food; *vertical dashed line*: H and M food at equal price. Data at log price ratio = 0 are replotted from *Figure 4B*. Replicates, Trained, *point*(*N*): *a*(9), *b*(14), *c*(18), *d*(27), *e*(20), *f*(10), *g*(6). Replicates, Untrained, *point*(*N*): *a*(12), *b*(17), *c*(13), *d*(12), *e*(28), *f*(10), *g*(7). (**C**). GARP analysis of *C. elegans* food preferences. Plotted points show mean consumption of M food versus consumption of H food in Trained and Untrained worms. Lines are choice sets as in *Figure 1*. The *x* and *y* intercepts of each line indicate the amounts of H and M food that would have been consumed if the worm spent all its pumps on one or the other food type. Error bars, 95% CI. (**D**). Predicted consumption of H and M food in Trained and Untrained animals on a widely-used ensemble containing 11 budget lines (*Harbaugh et al., 2001*). (**A–C**). Error bars, 95% CI. For sample size (*N*), statistical methods used, and significance level, see *Table 2*, rows 23-26.

The online version of this article includes the following source data and figure supplement(s) for figure 5:

**Source data 1.** Economic analysis of food choice in *C. elegans*.

**Source data 2.** Distributions of preference values in trained and untrained animals in *Figure 5B*.

*Figure 5 continued on next page*

**Figure supplement 1.** Distributions of preference values in trained and untrained animals in *Figure 5B*.

**Figure supplement 2.** Indirectly revealed preferences inherent in the choices shown in *Figure 5C*.

## Integration of preference and price

In a GARP experiment, participants evaluate offerings in the choice sets by taking into account their preferences and the price of various goods. To determine if *C. elegans* takes preference and price into account, we repeated the experiment of *Figure 4B*, now for a broad range of relative H and M prices (*Figure 5B*). To avoid progressive effects of feeding and satiety on food choices, each worm experienced a single choice set, and we allowed worms to feed for only 12 min.

Data were analyzed by plotting preference, the mean fraction of pumps spent on H food, $f_H$, against the log of price ratio $\log(p_M/p_H)$ which, by *Equation 2*, is equal to $\log(d_H/d_M)$. The data were fit by an exponential sigmoid function of the form

$$f_H(r) = \frac{1}{1 + 10^{-(r-r_0)/k}} \tag{9}$$

where $r$ is log price ratio, $r_0$ is log price ratio at the point of indifference between H and M food ($f_H(r_0) = 0.5$), and $k$ set dynamic range of the function. We chose an exponential sigmoid because, like $f_H$, it is bounded between 0 and 1. We refer to this mathematical relationship as a *price-ratio curve*. We found that in trained and untrained groups alike, worms spent more pumps on H food as its relative density rose, that is, as its relative price was reduced (*Figure 5B*, effect of price ratio, *Table 2*[23]), showing that worms take relative price into account when choosing food. Training shifted the price-ratio curve (*Figure 5B*, Trained vs. Untrained, *Table 2*[24]), such that the inferred indifference point between H and M food (intersection of the fitted curve and the dashed horizontal line) moved leftward. That is, a higher relative concentration of M food was now required to make the two options equally preferred. In other words, training increased the relative preference for H food. Importantly, we also found that worms could be induced to spend the majority of pumps on non-preferred food M if the preferred food H was made sufficiently dilute, i.e., expensive (*Figure 5B*, point *a*, Trained and Untrained, $f_H < 0.5$, *Table 2*[25,26]). Therefore, price can overcome preference. Taken together, these data show that neither food preference nor price is the sole factor determining consumption. Worms appear to take both preference and price into account, as human consumers often do, and as required for a GARP experiment.

## Utility maximization

To apply the test for utility maximization to *C. elegans* food choice, we mapped the data of *Figure 5B* into the GARP framework by plotting mean consumption of M food against mean consumption of H food at each price ratio (*Figure 5C*). Under this mapping, the lines in *Figure 5C* correspond to choice sets like those in *Figure 1*. There is one line for each combination of price ratio and income. The $x$ and $y$ intercepts of these lines indicate the amount of food (up to a scale factor) that would have been consumed had the worms spent all of their pumps on H or M food, respectively. We found it impractical to standardize the number of pumps to impose a fixed budget on each worm because of individual differences in latency to feed and mean pumping rate during the recordings (*Figure 4C–F*). We therefore plotted proportional consumption, $q_H$ and $q_M$ (*Equation 6*).

Utility maximization was assessed according to the procedure outlined by *Varian, 1996*. This assessment yields a single number that captures the total degree of consistency of preferences. Using choices averaged over all trials, we found no violations of GARP in either the trained or untrained data set (*Figure 5C*). Therefore, worms were choosing as if they were maximizing utility on the seven budget lines in our study.

To assess the robustness of our finding of utility maximization, we first considered whether it could be attributed to sampling error. Error bars in *Figure 5C* show the confidence intervals for mean consumption of H food on each budget line. Given this degree of variability, it is conceivable that one or more violations was missed because of sampling error. We therefore constructed $10^8$ simulated data sets by sampling from the gaussian distributions implied by the means and standard errors of each point. Sampling from gaussian distributions was justified by the form of the distributions of

preference values (*Figure 5—figure supplement 1*). Finding no violations of utility maximization in either the trained or untrained group, we estimated the probability of at least one violation to be less than $10^{-8}$. It is therefore unlikely that the absence of violations was an accident of sampling error.

However, it is conceivable that violations might have been observed if we had used a larger ensemble of budget lines. To address this concern, we predicted the choices worms would make on a widely used ensemble (*Harbaugh et al., 2001*; *Camille et al., 2011*; *Chung et al., 2017*), which contains 11 budget lines and covers the choice space more uniformly than our 7-budget ensemble. We first assessed the stringency of the 11-budget ensemble as a test for the utility maximization. This was done by assuming a conservative null hypothesis: that worms choose completely randomly, such that the fraction of pumps in H food, $f_{\mathrm{H}}$, could be modeled by drawing from a flat distribution between 0 and 1 (fraction of pumps in M food was $1 - f_{\mathrm{H}}$). Based on $10^6$ random data sets constructed in this way, we estimated the probability of a false positive finding of utility maximization (no violations) in the 11-budget ensemble to be 0.06. The 7-budget ensemble, by comparison, has a false positive probability of 0.87. We conclude that the 11-budget ensemble is considerably more stringent.

To compute how *C. elegans* would be expected to perform when choosing on the stringent 11-budget ensemble, we used *Equation 9* to predict the expected behavior on the budget lines in *Figure 5D*. There were no violations of GARP. There were also no violations when, instead of *Equation 9*, we used piecewise linear representations of *Figure 5* data. Finally, to take into account the variance about the means in *Figure 5B*, we created $10^6$ random data sets by sampling from gaussian distributions centered on the means. The standard deviation of the gaussians was set to the highest value of the standard error of the mean in either the trained or untrained data set in *Figure 5B*. By this procedure, we estimated the probability that worms would exhibit utility maximization in the 11-budget ensemble to be $\geq 0.98$, regardless of training state or type of fit (*Equation 9* or piecewise linear). We conclude that the worm's price-ratio curve likely constitutes a robust utility maximization strategy.

## Higher order features of utility maximization

Evidence that *C. elegans* is a utility maximizer (*Figure 5C and D*), allowed us to investigate several higher order features of utility maximization considered to be properties of human decisions. Ultimately, we succeeded in establishing a utility model describing what may be the underlying valuation process guiding the worm's choices.

Economic theory distinguishes between two main classes of goods – *substitutes* versus *complements* – according to how changes in the price or quantity of one good that a consumer possesses affects consumption of a second good. In the case of substitutes, which are relatively interchangeable from the consumer's perspective, an increase in the price of one good causes a decrease in the consumption of that good and a compensatory increase in consumption of the substitute. This occurs as consumers trade some of the good whose price increased for more of the alternative good. Pairs of goods that are traded-off at a constant exchange rate, regardless of the amounts of goods on offer, are called *perfect substitutes*; black and blue pens are an example of perfect substitutes. In the case of complements, which are defined as goods that are more desirable when consumed together rather than separately, an increase in the price of one good causes a decrease in consumption of both goods. Left and right are an example of perfect complements. Wearing only one shoe has essentially zero utility, so increases in the price of left shoes leads to decreased shoe consumption overall. Although worms consistently ate some of both foods in our experiments, we predicted that H and M food should act, to some degree, as substitutes for each other, as each provides nutrition.

To test this prediction, we took advantage of the design of the experiment in *Figure 5B*. The seven choice sets can be arranged in groups in which the price of one food was held constant while the other food was offered at three different prices. There were two groups in which the price of M food was constant while the price H food changed (points *a*, *b*, *d*, and points *c*, *e*, *f*), and there was one group in which the price of H food as constant while the price of M food changed (points *d*, *e*, *g*); see *Supplementary file 1*. We analyzed these groups by plotting consumption of the food whose price was constant against the price of the other food (*Figure 6A*). In five out of six cases, consumption of the constant-price food increased in response to increases in the price of the other food (*Figure 6A*, non-zero positive slope, *Table 2*[27-32]). We conclude that H and M food are substitutes for *C. elegans* as predicted.

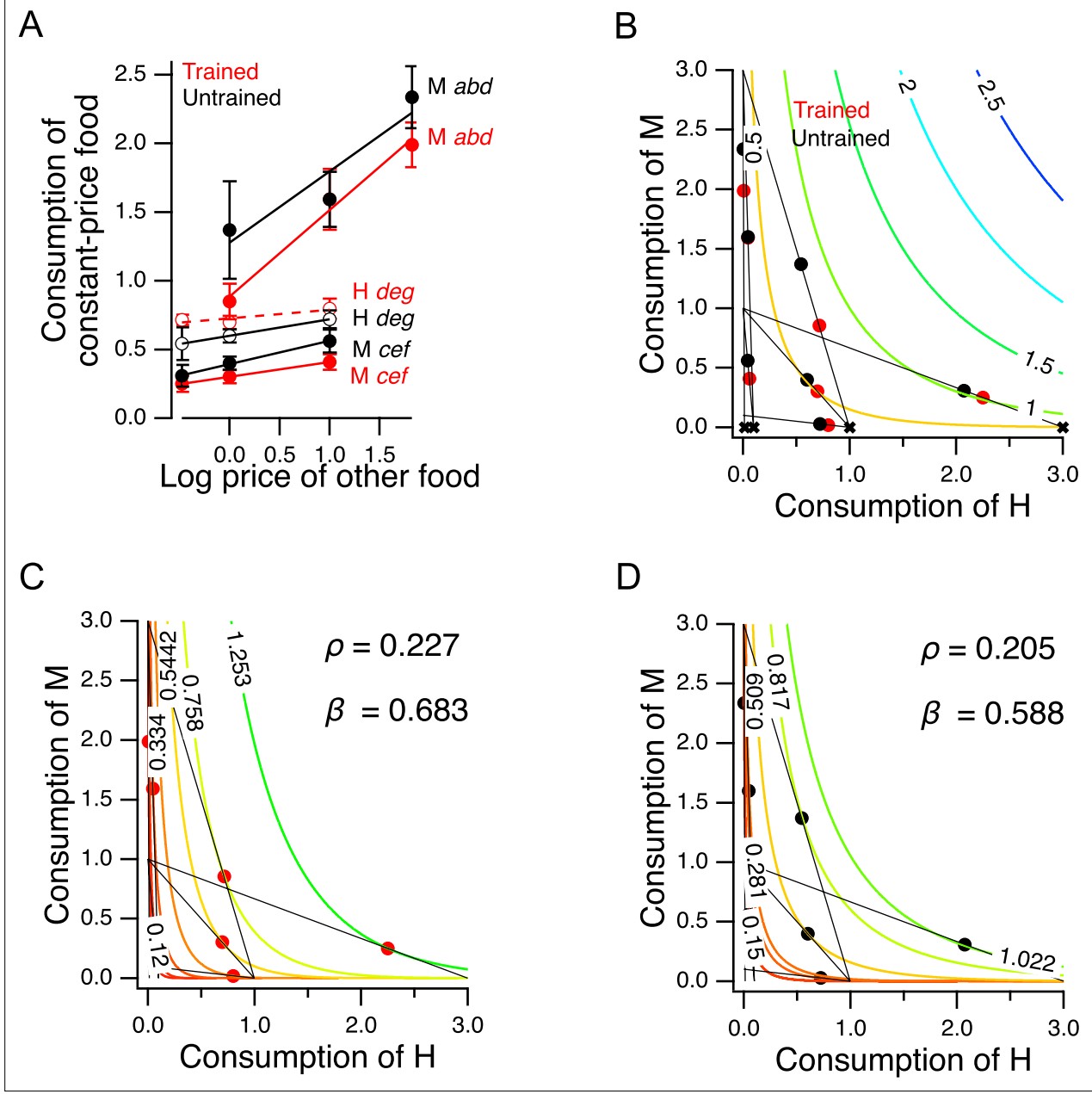

**Figure 6.** Higher order features of utility maximization. (**A**) H and M food act as substitutes. Consumption is plotted against price for triplets of cost ratios in which the concentration of one food was constant and the concentration of the other food was variable (see ***Supplementary file 1***). *Lower case italic letters*: data points in ***Figure 5B***. *Capital letters*: the food whose density was constant, the consumption of which is plotted on the $y$-axis. *Solid lines*: regression slope different from zero ($p \leq 0.01$); *dashed lines*: slope not different from zero. Error bars, 95% CI. (**B**). H and M food are not perfect substitutes. Colored contour lines are indifference curves in a perfect substitute model (***Equation 10***) with $\beta = 10/11$. Data points from ***Figure 5C*** are replotted for comparison, with associated budget lines, according to the conventions of that figure. 'X' symbols indicate the point of highest utility on each budget line. (**C–D**). Best fitting parameterizations of the CES function (***Equation 11***) for Trained and Untrained animals. Each panel shows the seven the iso-utility lines that are tangent to the budget lines. Goodness of fit can be assessed by observing that the iso-utility lines are tangent to the budget lines at, or near, the data points which indicate mean consumption of H and M food. For sample size (*N*), statistical methods used, and significance level, see ***Table 2***, rows 27-32.

The online version of this article includes the following source data for figure 6:

**Source data 1.** Higher order features of utility maximization.

Further, we can be reasonably certain that the H and M foods used in our experiments are not *perfect substitutes*. In the case of perfect substitutes, utility is the weighted sum of the consumed amount of each good. This relationship can be described with the equation

$$U(x, y) = \beta x + (1 - \beta)y \tag{10}$$

where $x$ and $y$ are the amounts of each good, and $\beta$ is a weighting factor ($0 \leq \beta \leq 1$). In that case, the exchange rate between the two goods, i.e., the amount of $y$ required to compensate for giving up one unit of $x$, is a constant, $\beta/(1 - \beta)$. For example, goods that can be substituted on a one-for-one basis have an exchange rate of unity ($\beta = 0.5$). *Equation 10* defines a planar utility surface that lies above the positive $x, y$ plane and passes through the origin. This plane is indicated by the colored contour lines in *Figure 6B*, which also contains the data of *Figure 5C* for comparison. The contour lines represent iso-utility lines within the plane. Such lines are called *indifference curves*, because a chooser would be indifferent between bundles located on the same curve, as these would have the same utility. In the example shown, $\beta = 10/11 > 0.5$, meaning that the slope of the plane is steeper in the direction parallel to the axis indicating consumption of good $x$, that is, H food in the figure. A utility maximizer will therefore choose the points labeled 'X' as these have the highest utility available on the associated budget line. Such points are called *corner solutions*, where the chooser devotes the entire budget to a one of the two options. In this example, the corner solutions are arrayed along the $x$ axis (and a hypothetical worm showing this pattern of perfect substitution would spend all of its pumps eating only the H food); for $\beta < 0.5$ the corner solutions would be arrayed along the $y$-axis. Worms in the Y-chip may be incapable of generating perfect corner solutions because they continuously wave their heads, frequently crossing into the opposite stream. We found that they are nevertheless capable of generating close approximations to corner solutions (e.g. point $g$ for trained worms in *Figure 5C*). Even so, we did not observe a consistent pattern of approximate corner solutions. We conclude that *C. elegans* does not perceive H and M food as perfect substitutes under our conditions, but rather as *imperfect substitutes*.

## A model of valuation in the worm

In widely used models of choices made by human consumers when offered imperfect substitutes, the exchange rate is not constant, but varies as a function of the amount of each good offered in a particular bundle. In economics this case is often modeled by the Constant Elasticity of Substitution (CES) function. The CES function takes the form

$$U(q_{\text{H}}, q_{\text{M}}) = \left(\beta q_{\text{H}}{}^{\rho} + (1 - \beta)q_{\text{M}}{}^{\rho}\right)^{1/\rho} \tag{11}$$

The exponent $\rho$ ($\neq 0$) represents the sensitivity of choosers to the fact that the more of a good they already possess, the less valuable each additional unit of the good becomes; in economics, this is called *diminishing marginal utility*. This sensitivity is inversely related to $\rho$. Here, as in *Equation 10*, $\beta$ captures the tradeoff between the goods, now after transformation by $\rho$. The CES utility function is quite flexible in that it can generate indifference curves for perfect substitutes ($\rho = 1$), imperfect substitutes ($-0.5 < \rho < 1$), and perfect complements ($\rho \ll -0.5$).

The best fitting parameterizations of the CES function ($\beta, \rho$) for the preference data in *Figure 5C* are shown for trained and untrained worms in *Figure 6C and D*, respectively. The highest contour reached by any budget line is the one that is tangent to it, and this contour constitutes the CES function's prediction of the worm's mean preference in that choice set. The close match between model and behavior for trained and untrained worms indicates that *C. elegans* food choice conforms to a widely used model of choice behavior in humans. We found that $\beta$ and $\rho$ increased following training. Therefore, training caused worms value H more and made diminishing marginal utility less relevant to their food valuations.

## Behavioral mechanisms of utility maximization

We next asked how utility is maximized at the behavioral level. As noted in above, we defined preference as the fraction of pumps in H food, $f_{\text{H}} = N_{\text{H}}/(N_{\text{H}} + N_{\text{M}})$, where $N_{\text{H}}$ and $N_{\text{M}}$ are the number of pumps in H and M food, respectively. In one model, the *pumping-rate model*, the behavioral expression of preference is a higher pumping rate in the preferred option. Alternatively, in the *dwell-time*

*model*, behavioral expression is increased amount of time spent feeding on the preferred side. Because the number of pumps in a given food type is equal to product of the time spent in that food and the mean pumping frequency in that food, an equivalent expression for preference is

$$f_H = \frac{F_H t_H}{F_H t_H + F_M t_M} \tag{12}$$

where $F$ and $t$ are, respectively, mean pumping frequency and mean total dwell time in the indicated food type. Limiting cases are informative here. If preference depends entirely on pumping frequency, then $t_H = t_M$, and *Equation 12* reduces to the pumping-rate model

$$f_H = \frac{F_H}{F_H + F_M} \tag{13}$$

in which preference for H food occurs when $F_H > F_M$, whereas preference for M food occurs when $F_M > F_H$; we refer to *Equation 13* as the *frequency index*. Plotting preference as defined by *Equation 13* for each animal against its actual preference, $f_H$, revealed a modest but significant negative correlation (*Figure 7A*, *Table 2*[33]). This result indicates a paradoxical but weak tendency to pump more slowly in H food as preference for it increases. The mean frequency index was 0.50±0.004 SEM and the extreme $y$-values of the linear fit to the data were only about 10% higher or lower. Therefore, we conclude that worms pump at approximately the same frequency in H and M food.

Conversely, if preference depends entirely on time in each food, then $F_H = F_M$, and *Equation 10* reduces to the dwell-time model

$$f_H = \frac{t_H}{t_H + t_M} \tag{14}$$

in which preference for H food occurs when $t_H > t_M$, whereas preference M food occurs when $t_M > t_H$; we refer to *Equation 14* as the *dwell-time index*. We found a strong positive correlation preference defined by the well-time index and actual preference, $f_H$ (*Figure 7B*, *Table 2*[34]). These findings favor the dwell-time model. We propose that once the worm detects the presence of food, pharyngeal pumping is activated at a fairly constant rate and preference is determined by the fraction of time the worm's head is in H food.

To determine how dwell time is biased toward the preferred food option, we measured mean head angle for each animal in the data set underlying *Figure 5B*. As expected, we found a strong positive correlation between the dwell-time index and mean head angle in the Y-chip (*Figure 7C*, *Table 2*[35]). Calcium imaging suggests that the angle of the worm's head with respect to the rest of the body is regulated by differential activation of dorsal and ventral neck muscle motor neurons (*Hendricks et al., 2012*). We propose, therefore, that the function of the neural circuit that maximizes utility is to generate asymmetric activation of these motor neurons during head bends.

## Role of chemosensory neurons in utility maximization

### *ceh-36* is required for the effects of food quality training measured in the Y-chip

In a final series of experiments, we began the search for neuronal representations of utility in *C. elegans*, beginning with its chemosensory neurons. As shown in *Figure 4B*, N2 worms preferred H to M food in the Y-chip, and there was a significant effect of training. In the experiment of *Figure 7D*, we found that whereas untrained *ceh-36* worms also preferred H to M food (*Figure 7D*, *ceh-36* Untrained, $f_H > 0.5$, *Table 2*[36]), the effect of food quality training differed between N2 and *ceh-36* (*Figure 7D*, training ×strain, *Table 2*[37]), such that trained and untrained *ceh-36* worms were indistinguishable (*Figure 7D*, *ceh-36* Trained vs. Untrained, *Table 2*[39]). These findings indicate a requirement for normal function of the *ceh-36* expressing neurons AWC and/or ASE in food quality learning.

## Characterization of AWC's response to delivery and removal of bacteria

In perhaps the most likely scenario, the worm's internal representation of H and M food is distributed across some or all of the 12 pairs of food sensitive chemosensory neurons in this organism. As a first step, we focused on one of these, AWC, because its role in directing chemotaxis to food is

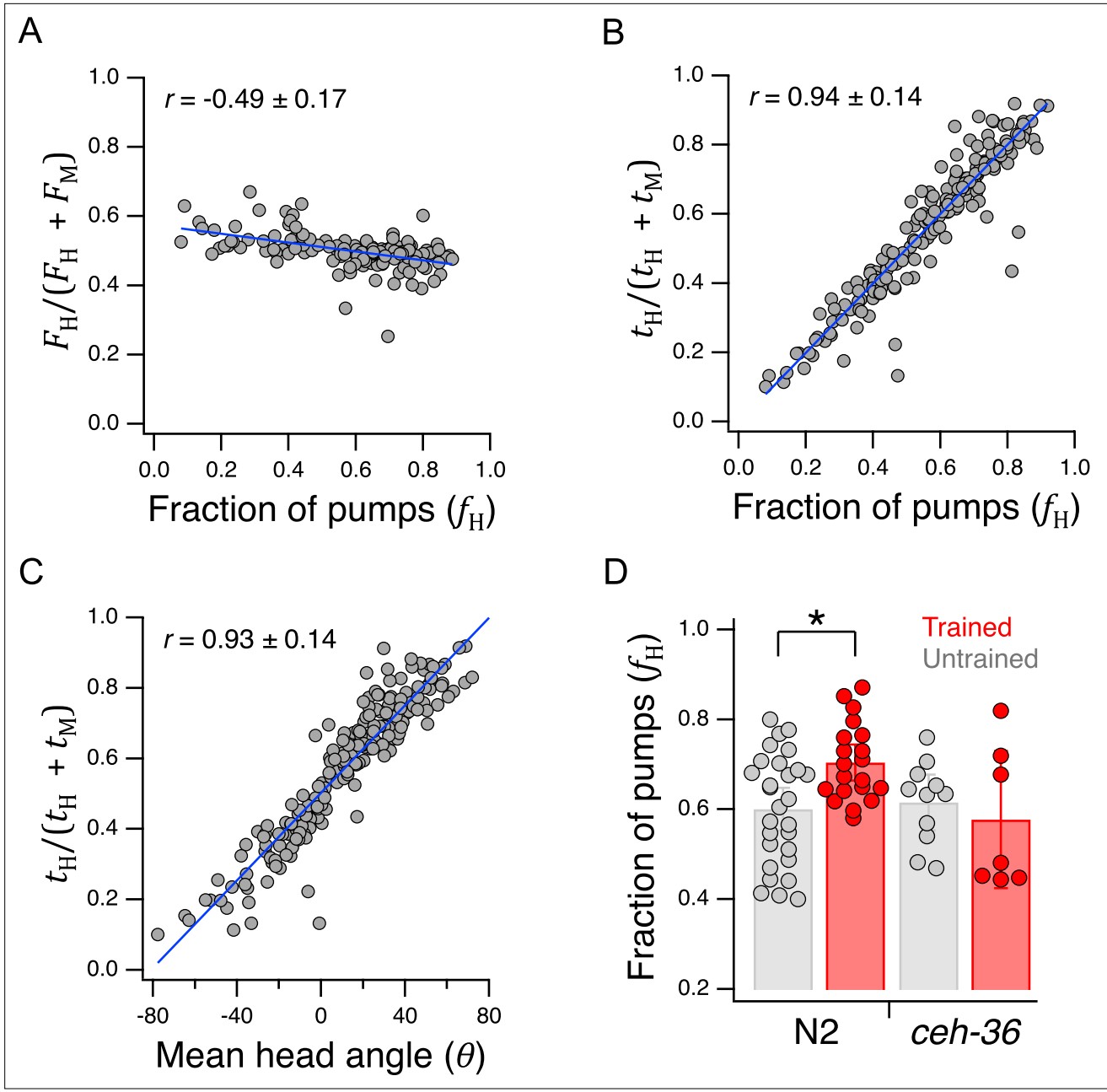

**Figure 7.** Behavioral mechanisms of utility maximization. (**A**) Pumping-rate model of preference. *Equation 13* is evaluated for each worm in *Figure 5B* and the result is plotted against preference in terms of fraction of pumps in H food for the same animal. (**B**). Dwell-time model of preference. Same as A but using *Equation 14*. (**C**). Regression of *Equation 14* against mean head angle as defined in *Figure 3B*. (**A–C**). *Blue lines*: regressions on the data. (**D**). Diminished *ceh-36* function eliminates the effect of food quality training on food preference. *Asterisk*, see *Table 2*[38]. (**H and M**) are at OD = 1. N2 data are from *Figure 4B*. Replicates, Trained, *strain*(*N*): N2(20), *ceh-36*(7). Replicates, Untrained, *strain*(*N*): N2(28), *ceh-36*(11). Error bars, 95% CI. Correlation coefficients are shown ±95% CI. For sample size (*N*), statistical methods used, and significance level, see *Table 2*, rows 33-39.

The online version of this article includes the following source data for figure 7:

**Source data 1.** Behavioral mechanisms of utility maximization.

particularly well established. Additionally, it is one of the few chemosensory neurons known from optogenetic manipulation to be capable of producing precisely the type of head-angle bias that underlies the expression of utility maximization in *C. elegans* seen in *Figure 7C* (*Kocabas et al., 2012*). The two AWC neurons are designated AWC[ON] and AWC[OFF] according to differences in gene expression (*Wes and Bargmann, 2001*). AWC[ON] and AWC[OFF] generate similar calcium transients to

odorants that they both detect (*Chalasani et al., 2007*); for consistency, we recorded from AWC[ON] (henceforth, 'AWC'). Calcium imaging shows that AWC is inhibited by bacteria conditioned medium or odorants, such as isoamyl alcohol, that are released by attractive bacteria (*Worthy et al., 2018a*) but then responds with a robust, positive-going transient when the stimulus is removed; AWC is therefore considered to be a food-off neuron. However, in one case, AWC is known to generate an off-transient when the stimulus is switched from the odor of a preferred food to the odor of a less preferred food (*Ha et al., 2010*; *Lin et al., 2023*). This finding indicates that AWC may be sensitive to food preferences.

To test whether AWC activity reflects preferences for the foods used in our study, we stimulated AWC with suspensions of H and M bacteria. We first characterized AWC's responses to the delivery and removal of H and M food (OD = 1) Such events mimic the experience of an AWC neuron when the worm moves in or out of a food patch, respectively, as in our accumulation assays (e.g. *Figure 2B–D*). We found, as expected from odorant experiments (*Chalasani et al., 2007*), that AWC is inhibited by food onset (*Figure 8A*) and excited by food offset (*Figure 8C*) regardless of food type, consistent with previously reported off-responses to *E. coli* (*Calhoun et al., 2015*).

Do AWC responses reflect the worm's overall preference for H? With respect to food onset, the apparently stronger inhibition in response to H than to M food (*Figure 8A*), when measured in terms of integrated calcium response, did not reach significance (*Figure 8B*, effect of food type, *Table 2*[40]). However, with respect to food offset, AWC was more strongly excited by H than M food (*Figure 8D*, effect of food type, *Table 2*[41]). At the more fine-grained level of peak responses within training groups, peak H food responses were significantly stronger than peak M food responses (*Figure 8C*, Untrained, effect of food type, *, *Table 2*[42]; *Figure 8C*, Trained, effect of food type, ×, *Table 2*[43]). Overall, we conclude AWC responds more strongly to removal of H food than M food. Exogenous activation of AWC increases the probability of a bout of reverse locomotion (*Gordus et al., 2015*). Therefore, AWC's response to food offset would be expected to increase reversal probability, leading to increased retention food patches. Our imaging data suggest that AWC-mediated retention would be stronger for H than M food, promoting greater preference for H food, as seen in *Figure 2B–D*.

What is the effect of food-quality training on AWC responses? Training had no effect on integrated response to food onset (*Figure 8B*, effect of training, *Table 2*[44]). In the case of food offset, training increased integrated response to H food (*Figure 8D*, Trained vs. Untrained, *Table 2*[45] and H food, Trained vs. Untrained, *, *Table 2*[46]), but not to M food (*Figure 8D*, *Table 2*[47]). Overall, our imaging data suggest that AWC-mediated retention in H food patches would be stronger in trained than untrained animals, promoting greater preference for H food after training, as seen in *Figure 2B–D*.

## AWC's response to transitions between H and M food

We next asked whether AWC responds to changes in food quality. If AWC were merely a food-off neuron, then switching between H and M food at the same density should produce no response, as food is always present. We found, on the contrary, that AWC was strongly excited by H→M transitions, and peak responses were amplified by food-quality training (*Figure 9A*, asterisk, Trained vs. Untrained, *Table 2*[48]). AWC appeared to be inhibited by M→H transitions. The extent of inhibition, quantified as the integrated calcium transient, was insensitive to training (*Figure 9A*, *Table 2*[49]). We conclude that AWC is capable of reporting not only the mere presence or absence of food, but also changes in food quality.

The fact that normal *ceh-36* function is required for food preference in the Y-chip (*Figure 7D*), and that AWC is sensitive to food quality (*Figure 9A*), suggests that it could provide sensory input to the worm's utility maximization circuit. If so, it should respond to changes in food quality on the time scale of individual head bends (seconds). We therefore presented worms with an alternating series of 2 s. presentations of each food. (*Figure 9B*), which approximate the time to peak calcium responses in *Figure 9A*. The imaging trace in *Figure 9B* illustrates a typical fluorescence waveform in the experiment. AWC responded at two distinct time scales. On the longer time scale, it exhibited a positive transient that decayed over tens of seconds, reminiscent of the time course of responses to sustained changes from H to M food (*Figure 9A*). On the shorter time scale, AWC responded with positive-going transients at each H→M transition, and negative-going transients at each M→H transition. In control experiments, when each fluid channel contained bacteria-free buffer, AWC did not respond (*Figure 9—figure supplement 1*), making it unlikely that pressure artifacts associated

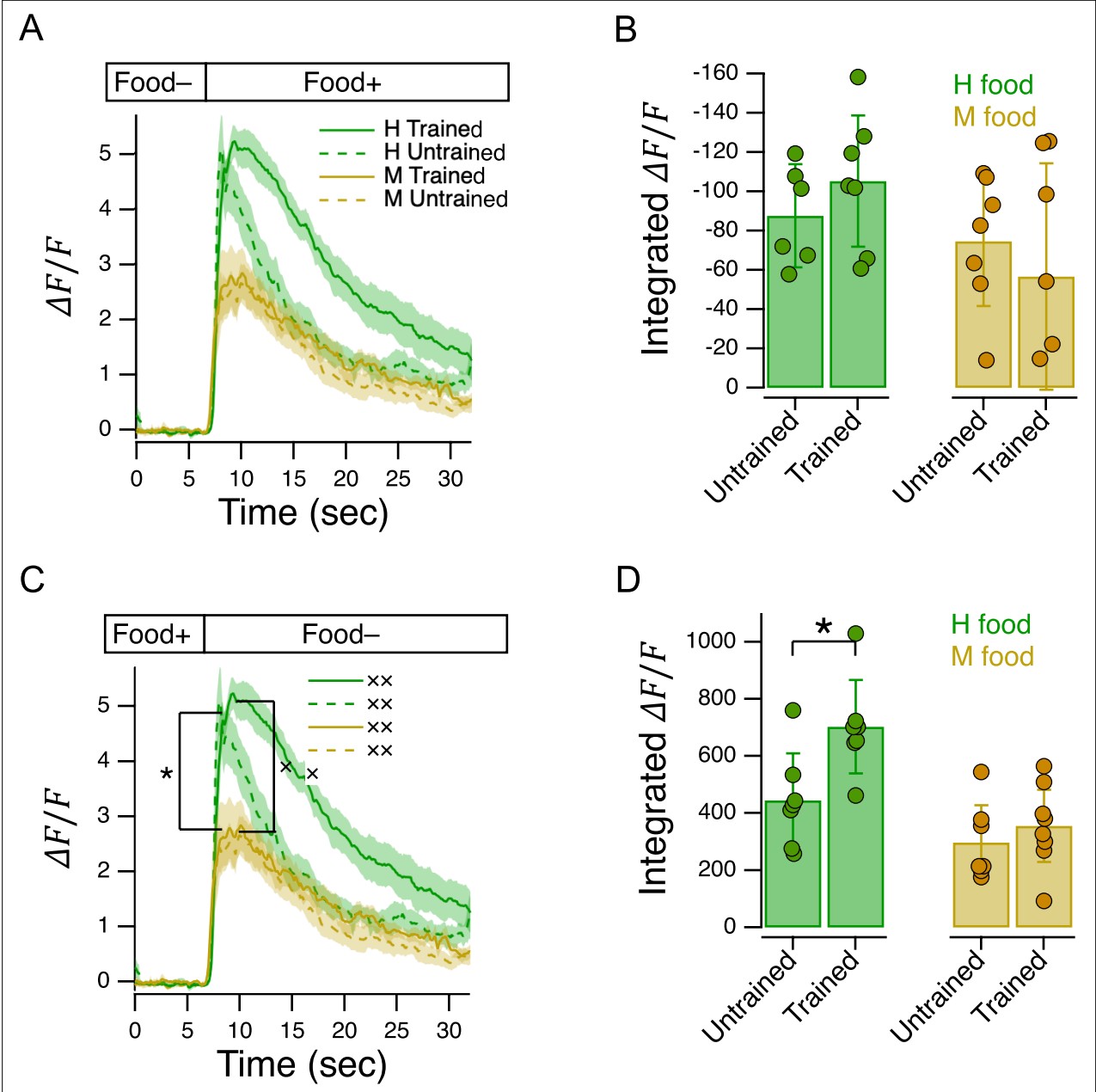

**Figure 8.** Characterization of AWC's response to delivery and removal of food. (**A**) Ensemble averages of relative fluorescence versus time in response to onset of the indicated food in trained and untrained animals. (**B**). Summary of data in A, showing mean integrated calcium transients. (**C**). Ensemble averages of relative fluorescence versus time in response to removal of the indicated food in trained and untrained animals. *Asterisk*: untrained group, mean peak response, H vs. M food, see *Table 2*[42]. *Cross*: trained group, mean peak response, H vs. M food, see *Table 2*[43]. (**D**). Summary of data in C, showing mean integrated calcium transients *Asterisk*, see *Table 2*[46]. (**A–D**). OD = 1 for H and M food. Shading, ± SEM. Error bars, 95% CI. For sample size (*N*), statistical methods used, and significance level, see *Table 2*, rows 40-47.

The online version of this article includes the following source data and figure supplement(s) for figure 8:

**Source data 1.** Characterization of AWC's response to delivery and removal of food.

**Figure supplement 1.** Imaging chip.

with valve activation contributed to fluorescence transients. We conclude that AWC is capable of responding to changes in food quality on the time scale of individual head bends.

Exogenous activation of AWC truncates head bends (*Kocabas et al., 2012*). The fact that positive transients occurred at the H→M transitions is consistent with AWC having a role in truncating head

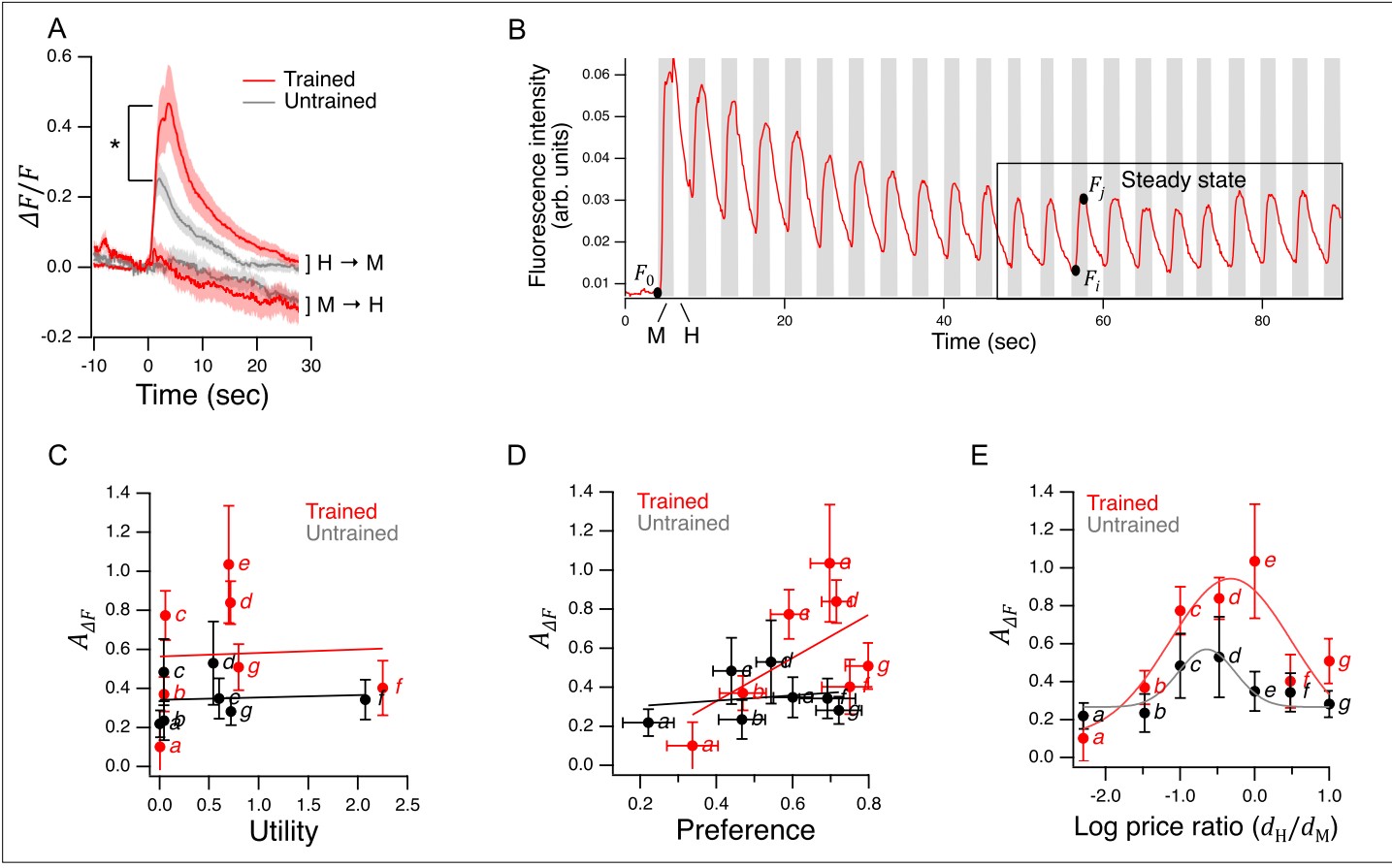

**Figure 9.** Characterization of AWC's response to bacterial foods in Y-chip assays. (**A**) Ensemble average of relative fluorescence versus time in response to transitions from H to M (H→M) or M to H (M→H) food. *Asterisk*, see **Table 2**[48]. Both foods were presented at OD = 1. Food was switched at $t$ = 0. H→M, Trained, $N$ = 16; H→M, Untrained $N$ = 14. M→H, Trained, $N$ = 16; M→H, Untrained, $N$ = 15. Shading,±SEM. (**B**). Typical fluorescence waveform in response to a series of transitions between H and M food. Notation: $F_0$, baseline fluorescence after sustained exposure to H food (≅ 2 min.); $F_M$, maximum H→M fluorescence; $F_H$, minimum M→H fluorescence. The box delineates the time period of presumptive steady-state responses over which mean response amplitudes $A_{\Delta F}$ were computed for each recorded worm. (**C–E**). $A_{\Delta F}$ versus utility, preference, and log price ratio. Lines and curves are, respectively, linear and gaussian fits. *Italic letters* refer to labeled points in **Figure 5B**. Replicates, Trained, *point*($N$): $a$(6), $b$(10), $c$(18), $d$(12), $e$(19), $f$(10), $g$(9). Replicates, Untrained. *point*($N$): $a$(8), $b$(11), $c$(10), $d$(8), $e$(12), $f$(8), $g$(10). Error bars, 95% CI. For sample size ($N$), statistical methods used, and significance level, see **Table 2**, rows 48-55.

The online version of this article includes the following source data and figure supplement(s) for figure 9:

**Source data 1.** Characterization of AWC's response to bacterial foods in Y-chip assays.

**Figure supplement 1.** Control for mechanical artifacts in the experiment of **Figure 9B** assays.

bends into the non-preferred food, as a means of biasing mean head angle toward the preferred side (**Figure 7C**). The waveforms we obtained closely resemble the response to alternating presentations of plain buffer and buffer containing a food-related odor sensed by AWC (**Kato et al., 2013**), suggesting the AWC perceives the transition to lower food quality as being similar to the transition to food-free medium.

## Representational content of AWC's responses

To investigate a possible role of AWC in utility maximization, we mapped its response function across the seven price ratios used in the GARP analysis of **Figure 5C**. We assumed that the steady-state region of the imaging trace is a reasonably accurate representation of AWC activity throughout the 12 min. behavioral recordings in the Y-chip (**Figure 5B**). We defined the amplitude of the calcium transients at steady-state as the mean fractional change in fluorescence between the peak at the end of an M step and its preceding trough at the end of an H step ($A_{\Delta F} = \langle (F_M - F_H)/F_0 \rangle$). In perhaps the simplest model of utility maximization, individual chemosensory neurons like AWC report changes in

the utility of the local environment. In that case, AWC activation would be well correlated with the utility of the chosen mixture of H and M food at each price ratio in *Figure 5B*. However, plotting $A_{\Delta F}$ against the utility values predicted by fits of utility functions to the preference data in *Figure 6C and D* showed no such relationship (*Figure 9C*, *Table 2*[50, 51]). We conclude that AWC is not representing the utility of chosen food. A similar analysis showed AWC activation was also not correlated with preference for H food (*Figure 9D*, *Table 2*[52,53]).

Given that AWC represents neither to chosen utility nor to preference, perhaps it represents some other economic quantity. Plotting $A_{\Delta F}$ against price ratio revealed an inverted U-shaped response function (*Figure 9E*). Such response functions have been observed in primate orbitofrontal cortex (*Padoa-Schioppa and Assad, 2006*) in plots of mean firing rate (analogous to our $A_{\Delta F}$ measure) against the ratio of offered amounts of two desirable juice rewards (analogous to price ratio in our study). By regressing mean firing rate against a wide range of value-related economic quantities, it was found that these neurons represent the relative value of the chosen juice, termed *chosen value*.

We therefore sought correlations between $A_{\Delta F}$ and offered food value (H or M), chosen food value, and 15 additional quantities that depend on them (*Supplementary file 2*). Food value, referenced to M food, was computed as

$$V_{\mathrm{M}} = d_{\mathrm{M}}$$
$$V_{\mathrm{H}} = w_{\mathrm{T}} d_{\mathrm{H}} \text{ or } w_{\mathrm{U}} d_{\mathrm{H}}, \tag{15}$$

where $w_{\mathrm{T}}$ and $w_{\mathrm{U}}$ specify the number of units of M food that are equivalent to one unit of H food assuming a linear valuation function in the vicinity of the indifference point (where $f_{\mathrm{H}} = 0.5$), in trained and untrained animals, respectively. These quantities were computed as $1/r_0$ from the fit of *Equation 9* to the preference data in *Figure 5B*. To test whether AWC represents any of these quantities, we computed correlation coefficients between $A_{\Delta F}$ each quantity, in trained and untrained animals. We found no significant correlations (*Supplementary file 3*). In a separate analysis, we also considered offered utility and change in chosen utility, again finding no significant correlations. We conclude that AWC represents neither the value of food nor utility.

A potential clue to the function of AWC in food choice stems from the observation that the peak of its response function lies at or near the point on the price-ratio axis where the two foods are presented at the same density. This is the point where foraging decisions are the most difficult, as food density is no longer available as cue to identify the better option. Moreover, food quality training greatly increased the height of the function's peak (*Figure 9E*, effect of training, *Table 2*[54]), improving AWC's ability to make such distinctions. Accordingly, we propose that AWC's role is to report changes in relative food quality as defined by prior experience.

## Discussion

There is substantial evidence that *C. elegans* is capable of several forms of valuation. These include valuations inherent in cost-benefit decisions, transitivity of binary preferences, and independence of irrelevant alternatives (*Barrios et al., 2008*; *Bendesky et al., 2011*; *Shinkai et al., 2011*; *Ghosh et al., 2016*; *Cohen et al., 2019*; *Iwanir et al., 2019*). The present work establishes a new reference point in the study of valuation in *C. elegans* in four key respects. (i) *C. elegans* food choices obey the classic law of supply and demand (*Figure 5A*). This law has been shown to apply not just to humans but to a variety of model organisms (*Kagel et al., 1975*; *Kagel et al., 1995*; *Bickel et al., 1995*), but none as simple as *C. elegans*. (i) *C. elegans* behaves as if maximizing utility (*Figure 5C*). When challenged with multiple trade-offs between food quality and price, its choices satisfy the necessary and sufficient conditions for utility maximization. An organism that maximizes utility also maximizes subjective value, for it is reasonable to maximize only what is valued. We believe this may be the first demonstration of value-based decision making according to GARP in an invertebrate. (iii) *C. elegans* food-consumption decisions are well fit by the CES utility function (*Figure 6C and D*). This function is widely used to model human consumers, but the extent to which it applies to other species remains an open question (*Fréchette, 2016*). Here, we show that the CES function accurately models consumption decisions in an organism that diverged from the line leading to humans 600 million years ago (*Raible and Arendt, 2004*; *Figure 6C and D*), which provides new evidence of the function's universality. (iv) In addition to demonstrating utility maximization in *C. elegans*, we have outlined a plausible mechanism for it.

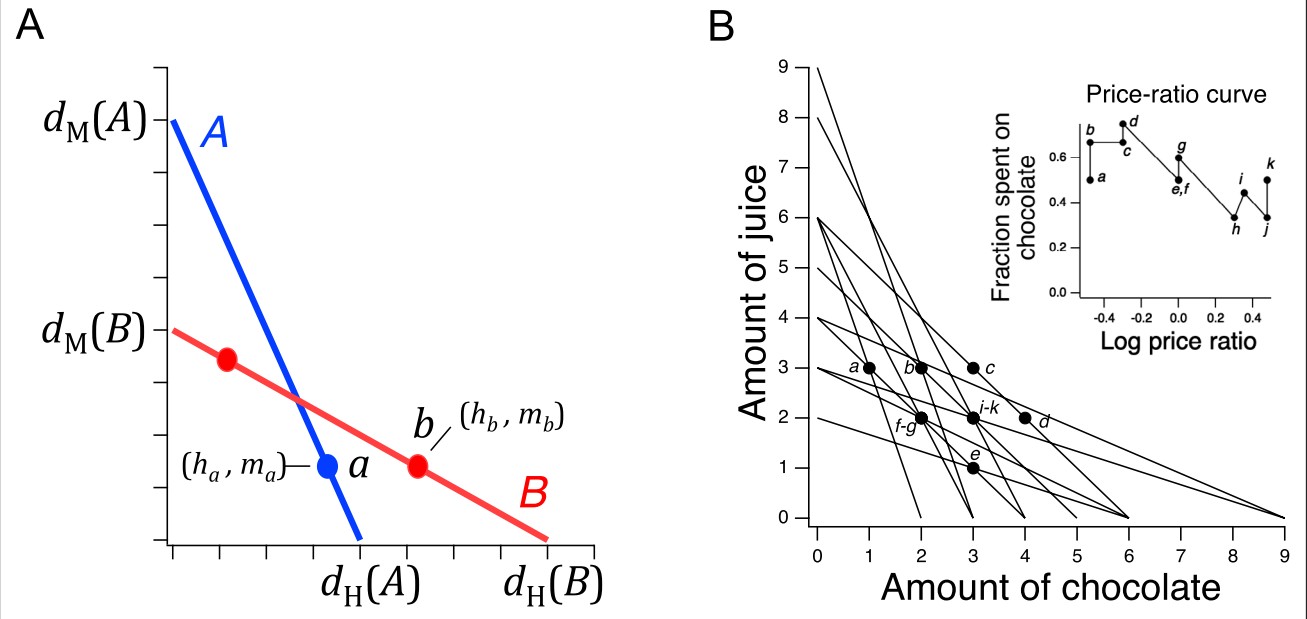

**Figure 10.** Price ratio curves and utility maximization. (**A**) A pair of intersecting budget lines wherein choices *a* and *b* are governed by a price-ratio curve (not shown) that is monotonic-increasing. To support a direct violation of GARP, *a* must be to the right of the intersection of lines *A* and *B*, and *b* must be to the left of it. However, by price-ratio monotonicity, *b* must lie to the right of *a*, for reasons described in the text. This constraint precludes direct violations of GARP. (**B**). Choice data from a human participant in a GARP experiment (*Camille et al., 2011*). The data are consistent with utility maximization. *Inset*, price ratio data inferred from the experiment. Log price ratios for points *a-k* are calculated as $logX_Z/Y_Z$, where $X_Z$ and $Y_Z$ are, respectively, the *x* and *y* intercepts of budget line *Z*. The fraction of total budget spent on good X is computed as $x_Z/X_Z$, where $x_Z$ is the amount of good X chosen on line *Z*.

The *C. elegans* price-ratio curve is monotonic-increasing (*Figure 5B*). Below we offer a proof that monotonicity of the price-ratio curve guarantees adherence to GARP (*Figure 10*). Importantly, such a price-ratio curve seems simple to implement at the neuronal level, requiring only that chemosensory neurons are able to regulate the amplitude of locomotory head-bends as they occur, and to do so monotonically in response to differences in food value on either side of the body.

A key challenge in the present study was to develop the methodology for presenting worms with choices that can be analyzed using revealed preference theory. This meant finding a method of offering worms qualitatively distinct goods whose prices could be varied systemically under the constraint of an explicit trade-off between them. It also meant finding a way to measure consumption of each good. We chose species of high- and medium-quality bacteria as goods. We showed that these species have different olfactory fingerprints, making it conceivable that they are qualitatively distinct (*Table 1*). Furthermore, we showed that worms feed differently on these species in a way that can only be explained by a model in which the foods are indeed qualitatively distinct (*Figure 4A*). Price differences were modeled as differences in bacteria density under the assumption that doubling food density reduces by half the number of pharyngeal pumps required to ingest a given mass of food. In human experiments, trade-offs are established by means of a budget constraint ensuring that choosing more of one good means getting less of the other. That is the only purpose of the budget constraint, and participants need not and are not made aware of it. In the case of *C. elegans*, it was not necessary to impose a budget constraint because trade-offs could be established simply by the geometry of the microfluidic device, which ensured that more a pump spent in one food is a pump not spent in the other food (*Figure 3*). Finally, to measure relative consumption of the two bacteria, we computed the fraction of pumps emitted in each food stream (*Figure 5B*). Overall, this methodology results in decision-making paradigm that is formally analogous to GARP experiments in humans.

There are several possible objections to this methodology. (i) There is the concern that revealed preference theory, created to model human consumers, is unnecessarily complex for modeling worm behavior. However, our study uses this framework not as a model but as litmus test for utility maximization. (ii) We have not shown that qualitative distinctions between bacteria (*Table 1*, *Figure 4A*)

are mirrored by distinct representations in the worm's nervous system, as is presumably the case in the human nervous system. However, revealed preference theory makes no such stipulation as to how utility maximization physically occurs, including neural representations of goods. In worms, cells other neurons, such as gut cells, could be part of the representation without violating theoretical assumptions. (iii) Finally, there is the concern that that conscious awareness of the budget constraint is a necessary condition for application of the theory. However, revealed preference theory merely requires the ability to choose between different bundles of goods. There is no requirement that that these choices be made consciously after considering prices and income and calculating the range of possible choices. The budget constraint can instead be imposed physically, as in this experiment.

Our study has several experimental limitations. (i) There was a high probability of a false-positive finding of utility maximization in *Figure 5C*. A contributing factor to this problem is the comparatively small number of budget lines in our budget ensemble. Another factor is the somewhat uneven coverage of choice space, with two budget lines close to the $y$ axis (*a* and *b*). We included these budget lines to make a stronger case for complex decision making in the worm by demonstrating preference reversals (*Figure 5B*, point *a*); this required presenting H food at very low density. We were able to address this limitation by showing that the price-ratio curve of *Figure 5B* predicted the absence of transitivity violations in a larger budget ensemble that covered choice space completely (*Figure 5D*). Moreover, we show below that the absence of violations in the larger ensemble is essentially guaranteed by the monotonic form of the worm's price-ratio curve. (ii) In the calcium imaging experiment of *Figure 9B–E*, dwell times in the two foods were equal (2 s). This stimulus pattern is somewhat unrealistic because in the behavioral experiments to measure food preference in the Y-chip, dwell times were rarely equal (*Figure 7B*). We nevertheless used equal dwell times because we were primarily interested in the question of whether AWC is capable of reporting food utility via its inherent sensitivity to the food quality, that is, without the complicating factor of dwell time, which is behaviorally determined and can increase or decrease peak calcium responses. (iii) Our demonstration of utility maximization is currently limited to the klinotaxis component of *C. elegans* foraging behavior. It does not necessary extend to other components, such as *klinokinesis* (biased random walk *Pierce-Shimomura et al., 1999*) and *orthokinesis* (slowing in the presence food *Sawin et al., 2000*). (iv) Finally, it will now be important to test whether utility maximization is observed in choices between other pairs of bacteria strains.

## Distinctions between GARP and binary transitivity

Assessment of utility maximization by GARP involves demonstration of internally consistent preferences. This means non-violation of transitivity relationships inherent in directly revealed preferences, as well as those revealed indirectly through sequences of directly revealed preferences. The scientific literature on transitive choice in animals is extensive, encompassing a wide range of vertebrates: primates (*Addessi et al., 2008*), birds (*Mazur and Coe, 1987*; *Sumpter et al., 1999*; *Schuck-Paim and Kacelnik, 2002*), and fish (*Dechaume-Moncharmont et al., 2013*). Many invertebrates also exhibit transitivity: bees (*Shafir, 1994*), fruit flies (*Arbuthnott et al., 2017*), nematodes (*Cohen et al., 2019*; *Iwanir et al., 2019*), and even slime molds (*Latty and Beekman, 2011*). However, these studies have focused almost entirely on binary transitivity, meaning either-or decisions between pairs of items. Transitivity in this sense can be impressive, even in simple organisms. Choices of male fruit flies presented with pairs of genetically distinct females drawn from 10 divergent inbred strains form a transitive hierarchy of order 10 (*Arbuthnott et al., 2017*). Such examples notwithstanding, binary transitivity is not a sufficient condition for utility maximization. That is because in binary choices there is no notion of amount. It is therefore impossible to assess whether choices between different goods are consistent with the rule 'more is better' which, as illustrated in *Figure 1A*, is a necessary condition for utility maximization. Here, we have demonstrated a form of transitivity in which not only the quality of goods but also their offered amount (hence price) is taken into account by the organism. The full set of transitivity relationships exhibited by worms in our study is illustrated in *Figure 5—figure supplement 1*.

## What is being maximized?

Utility maximization raises the ineluctable question of what is being maximized. Revealed preference theory, including GARP, cannot answer this question. At present, it is easier to identify what is

*not* being maximized. Three ethologically plausible maximization targets can be ruled out by simple inspection of *Figure 5C*. (i) It is unlikely that the worm is maximizing the amount of high quality food consumed, as that would have resulted in corner solutions. (ii) It is also unlikely that the worm is maximizing the overall amount of food consumed. That would have resulted in feeding exclusively in whichever of the two streams carried denser food, again yielding corner solutions (except at point *e*, where densities are equal and so the worm would be indifferent, which is was not). (iii) Finally, and perhaps surprisingly, the worm is probably not maximizing its potential for growth, at least in terms of a simple model in which this potential is the product the characteristic growth rates of worm in H or M food, 0.50 and 0.43 days$^{-10}$, respectively (*Shtonda, 2004*), and the densities of those foods. The growth-potential model fails to explain our data because H and M growth rates are not sufficiently different to make the less dense food preferred under any of the price ratios tested. It predicts an outcome similar to maximizing amount of food consumed (except at point *e*, where instead of indifference, it predicts that H food would be preferred). Together, considerations (i)-(iii) imply that the worm is not maximizing fitness, at least in terms of food consumption. It is conceivable that the worm is maximizing some combination of fitness-related quantities which, in addition to food consumption, might include knowledge about the local food environment (*Calhoun et al., 2014*), food safety (*Song et al., 2013*), and the foraging benefits of continuous locomotion while feeding (*Shtonda and Avery, 2006*), but we have no evidence of these as yet.

The worm's preference for non-corner solutions, called 'interior solutions', is consistent with at least two hypotheses concerning what is being maximized. The worm might be maximizing something having to do with mixtures of the two foods, reminiscent of 'a taste for variety' in economics (*Senior, 1836*; *Jevons, 1871*). This would be an appropriate strategy if, for example, the two foods had unique essential nutrients, but we currently have no evidence for or against this. A second hypothesis is that interior solutions arise because the worm is balancing the benefit of consuming the better food option against the cost of forgoing the possibility of the sudden appearance of an even better option on the other side; this hypothesis mirrors an exploitation-exploration trade-off. Alternatively, neuromechanical constraints such as chemosensory transduction delays coupled with behavioral momentum (inability to reverse the current head bend).

## Establishment of subjective value assignments

It is well established that learning alters food preferences in *C. elegans* (*Ardiel and Rankin, 2010*). We extended these findings by showing that worms are capable of food quality learning for palatable foods at least as late as larval stage L4 (*Figure 2*) and that this process requires intact dopamine signaling. A requirement for dopamine signaling in *C. elegans* has also been demonstrated in the formation of enhanced preferences for salt ions that were paired with cocaine or methamphetamine, drugs that increase the amount of dopamine in the synaptic cleft (*Musselman et al., 2012*). Furthermore, pairing these drugs with a particular food increases preference for it. These findings are consistent with model in which trained preferences arise by a form of classical conditioning that is driven by the rewarding properties of food. However, our data do not exclude possible non-associative effects during training, such as sensitization and habituation to H and M food, respectively. Studying untrained worms, we found evidence for acquisition of latent food preferences that occurred in the absence of direct experience of the newly preferred foods. The fact that acquisition of latent preferences also requires intact dopamine signaling (*Figure 2D*) is consistent with this process being reward-driven, but developmental effects cannot be ruled out on the basis of the available data.

Training and the latent effects of food experience may be dissociable at the circuit level in that training effects, but not latent effects, exhibit a requirement for *ceh-36* (*Figure 2B*). In one simple model, at least one of two chemosensory neurons, AWC or ASE, is a locus of food memories implanted by sampling both foods repeatedly over time, whereas some or all of the other chemosensory neurons mediate latent preferences. In partial support of this model, AWC's response to removal of H food (*Figure 8C*) and to H→M transitions (*Figure 9A and E*), is strongly enhanced by training. It will now be important to determine whether food responses in ASE neurons are also modified by training, and whether the responses of other chemosensory neurons are exhibit latent effects.

The fact that in *cat-2* mutants training has no effect on food preferences (*Figure 2D*) implicates dopamine signaling in the acquisition or expression of food quality memories. One hypothesis is that during training, dopamine neuron activity in *C. elegans* signals reward, as it does in many other

organisms. In support of this model, *cat-2* is required for learned associations between salt cues and drugs of abuse that co-opt the reward systems, such as cocaine and methamphetamine (*Musselman et al., 2012*) as well acquisition of alcohol preference (*Lee et al., 2009*). Similarly, the *C. elegans* gene *asic-1*, a member of the degenerin/epithelial sodium channel family that is mainly expressed at presynaptic terminals of dopaminergic neurons, is required for learned associations between tastes or odors and the presence of food (*Voglis and Tavernarakis, 2008*).

## Monotonic-increasing price-ratio curves guarantee adherence to GARP

The preference values $f_H$, plotted against price ratio $r = d_H/d_M$ (*Figure 5B*), were well fit by smooth curves that are monotonic-increasing (strictly, *non-decreasing*; *Equation 9*). Here we offer a proof that a monotonic-increasing price-ratio curve precludes GARP violations. In *Figure 10A*, Point $a$ represents any location on *A* such that $b \succ a$ could be revealed directly (meaning $a$ is between the *A-B* intersection and the $x$ axis; see *Figure 1A*). Point $b$, on the other hand, represents any location on *B* that is consistent with monotonicity, conditional on the location of $a$. By *Equation 6*, the $x$ coordinates of points $a$ and $b$, representing the consumed quanties of H, are

$$h_a = f_H(r_A) \cdot d_H(A)$$
$$h_b = f_H(r_B) \cdot d_H(B) \tag{16}$$

Budget line *A* has the steeper slope of the two lines which means that $d_M(A)/d_H(A) > d_M(B)/d_H(B)$. These ratios are the inverse of price ratio. Therefore, $r_B > r_A$ and, by monotonicity,

$$f_H(r_B) > f_H(r_A) \tag{17}$$

Furthermore, by construction,

$$d_H(B) > d_H(A) \tag{18}$$

Given these inequalities, $h_b > h_a$, meaning that point $b$ must always lie to the right of $a$. Therefore, $b$ can never reside to the left of the *A-B* intersection, as needed to generate a direct violation. Monotonicity of the price-ratio curve thereby precludes direct violations of GARP. Furthermore, for indirect violations to occur, there must be at least one direct violation (*Rose, 1958*; *Heufer, 2009*). Therefore, we can conclude that monotonic price-ratio curves guarantee adherence to GARP. This conclusion explains our finding that when modeling *C. elegans* food choice by the worm's price-ratio curve, the probability of finding utility maximization in the 11-budget ensemble was ≥0.98 (the exceptions being due to sampling noise). Although monotonic-increasing price-ratio curves guarantee utility maximization, they are not necessary for it. *Figure 10B* shows data from a human subject who exhibited no utility violations, yet whose price-ratio curve was clearly non-monotonic. Thus, in relying upon monotonic price-ratio curves, the worm's choice strategy may be an adaptation for utility maximization under the constraint of limited computational capacity.

## A neuronal mechanism of utility maximization

The following simple model illustrates how a monotonic price-ratio curve, and thus utility maximization, could be achieved by the *C. elegans* klinotaxis circuit. The food-sensitive chemosensory off-cells (AWC, ASER, AWB, ASH, ASK) and on-cells (ASEL, AFD, AWA, ASJ, BAG, ASI, ADF) activate, respectively, when the cell's preferred stimulus is removed or delivered (*Zaslaver et al., 2015*). The off-cells AWC and ASER are known to truncate head bends when exogenously activated (*Kocabas et al., 2012*). The model assumes that other off-cells do likewise, whereas on-cells extend head bends. Provided that summed activations of all off-cells and on-cells are, respectively, monotonic functions of price ratio, the amplitude of head-bend truncations and extensions will also be monotonic with respect to price ratio. Given the sinusoidal kinematics of *C. elegans* locomotion, this relationship necessarily extends to relative dwell time on the side of the preferred food option, thereby increasing the fraction of pumps on that side. The non-monotonic activation function of AWC (*Figure 9E*) could be compensated by activations of other chemosensory neurons at price ratios greater than unity. This model can now be tested by recording and stimulating neurons in the klinotaxis circuit during choices of the type studied here.

## Conclusion

All animals forage to obtain sufficient food at minimal cost. Omnivores like humans, *C. elegans*, and other species face the additional challenge of reconciling trade-offs between food quality and a variety of costs, such as energy, time (lost opportunity), and risk. Our findings expand the scope of comparative studies already in progress in ethology, behavioral ecology, and neuroeconomics to probe the limits of classical economic theory in explaining such fundamental trade-offs (*Kalenscher and van Wingerden, 2011*; *Pearson et al., 2014*). Is the classical theory universally applicable to animal behavior? To what extent can this theory be grounded in neurophysiology? The small size and unequalled annotation of the *C. elegans* nervous system (*White et al., 1986*; *Jarrell et al., 2012*; *Hammarlund et al., 2018*; *Cook et al., 2019*; *Brittin et al., 2021*; *Hobert, 2021*), coupled with recent advances in brain wide imaging in this organism (*Kato et al., 2015*; *Nguyen et al., 2016*; *Venkatachalam et al., 2016*) offer unique advantages in answering these questions.

# Materials and methods

**Key resources table**

| Reagent type (species) or resource | Designation | Source or reference | Identifiers | Additional information |
|---|---|---|---|---|
| Strain, strain background (*E. coli*) | OP50 | CGC (*C. elegans* Genetic Center) | RRID:WB-STRAIN:WBStrain00041969 | |
| Strain, strain background (*Comamonas sp.*) | DA1877 | CGC | RRID:WB-STRAIN:WBStrain00040995 | |
| Strain, strain background (*Bacillus simplex*) | DA1885 | CGC | RRID:WB-STRAIN:WBStrain00040997 | |
| Genetic reagent (*C. elegans*) | N2, Bristol | CGC | RRID:WB-STRAIN:WBStrain00000001 | |
| Genetic reagent (*C. elegans*) | CX5893 | CGC | RRID:WB-Strain00005275 | |
| Genetic reagent (*C. elegans*) | JP5651 | National Resource Project (Japan) | None | *cat-2(tm2261)* |
| Genetic reagent (*C. elegans*) | MT15620 | CGC | RRID:WB-Strain00027527 | *cat-2(n4547)* |
| Genetic reagent (*C. elegans*) | CB1112 | *Stern et al., 2017*; PMID:29198526 | RRID:WB-Strain00004246 | *cat-2(e1112)* |
| Genetic reagent (*C. elegans*) | XL322 | Lockery lab | None | ntIs1703[str-2::GCaMP6s-wcherry; unc-122::dsred2] |
| Recombinant DNA reagent | GCaMP-6s::wCherry | Zhen lab | None | |
| Software, algorithm | Igor Pro | Wavemetrics https://www.wavemetrics.com/ | Version 9.01 | |

## Worm strains

The following *C. elegans* genotypes were used:

| Experiment | Figure | Genotype |
|---|---|---|
| Wild type decision making | 2, 4–9 | N2 |
| Requirement for AWC/ASE neurons | 2B, 7D | *ceh-36(ky646)* |
| Requirement for dopamine signaling | 2D | *cat-2(tm2261)* |
| Calcium imaging from AWC neurons | 9 | *ntIs1703[str-2::GCamp6s-wcherry; unc-122::dsred2]* |
| Proportion of time on food | 2 (suppl.) | *cat-2(n4547), cat-2(e1112)* |

## Bacteria strains and suspensions

Streptomycin-resistant strains of three species were used: *E. coli* (OP50 DA 837) representing standard laboratory food, *Comamonas* sp. (DA1877) representing high-quality food (H), and Bacillus simplex (DA1885), representing mediumquality food (M). To culture bacteria, a small scraping of frozen stock

was transferred to 400 mL of LB broth in a 500 mL flask to which streptomycin (50 μg/mL) added to inhibit competitive bacterial growth. Cultures were grown overnight in on a shaker at 35 C. Bacteria cultured as described above were washed three times by centrifugation and resuspension of the pellet in 10–20 mL minimal buffer which contained (in mM): 1 MgSO4, 10 HEPES adjusted 350–360 mOsm (glycerol). The final wash pellet was resuspended in a smaller volume of buffer to create a stock suspension whose bacteria density was greater than the desired value. The density of the stock suspension was measured by dilution of a small sample into the linear range of the densitometer (0–1 $OD_{600}$). The final suspension at the desired density $d$ was obtained by dilution of the stock suspension.

## Identification of volatile organic compounds released by bacteria

Bacteria were prepared for Thermal Desorption Gas Chromatography-Mass Spectrometry (TD-GC-MS) analysis in a similar method as described (*Worthy et al., 2018a*; *Worthy et al., 2018b*) Bacteria were grown overnight in LB with 50 mg/mL streptomycin at 37 °C, centrifuged, and then resuspended at an $OD_{600}$=10. Two peptone-free NGM plates were prepared each with 9 spots of 25 μL of bacterial suspension. For the controls, 25 μL of LB media without bacteria was spotted on NGM plate. Plates were incubated for 1 hr at 20 °C. Then 18 squares of the NGM agar with 25 μL bacteria suspension were placed in a GC-MS glass vial for 24 hr. Samples were analyzed by Thermal Desorption (TD) GC-MS using the Agilent 6890 GC System equipped with a Markes Unity II Thermal Desorption System on the GC inlet, a Restek, Rtx-5 column, and Agilent 5973 Mass Selective Detector. The temperature program was: hold 8 min at 35 °C, increased to 130 °C at a rate of 10 °C/min, hold 5 min at 130 °C then increased to 300 °C at a rate of 15 °C/min, and hold at 300 °C for 1 min. MS ranged from 30 to 550 in full scan mode. VOCs were identified with the NIST 11 (National Institute of Standards and Technology) mass spectral library and pure chemical standards run following the same parameters as for bacterial samples. Samples were prepared for analysis in duplicate from a single stock of each bacteria and LB control samples were run immediately before or after each bacterial sample.

## Worm cultivation and training

Worms were synchronized by isolating eggs from 10 gravid adults allowed to lay eggs for five hours. Progeny were cultivated until late larval stage L3 at 20 °C on 50 cm plates containing low-density nematode growth medium (NGM; *Brenner, 1974*) seeded with standard laboratory food (*E. coli* OP50). In experiments involving *food quality training*, worms were washed three times in M9 buffer, pelleted by sedimentation, then transferred in a 2 μL aliquot (~150 worms) to a training plate (50 mm diam.) filled with bactopeptone-free NGM containing 200 μg/mL streptomycin. Training plates were prepared one day in advance by spotting them with eight patches each of H and M food. Patches were formed by pipetting 10 μL aliquots of bacteria culture, suspended in LB broth at OD = 10, in a 4×4 grid with 10 mm between patch centers (*Figure 2A*). Control plates were spotted with patches of OP50 in the same pattern. Worms resided on training plates for 18–24 hr before testing. In experiments involving the *food familiarity effect*, training plates contained 16 patches of either H or M food prepared in the same way.

## Behavioral assays

### Accumulation assays

In *open-field assays*, NGM test plates (50 mm diameter) were spotted with one patch each of H and M food (100 μL, OD = 10) separated by 25 mm. In *T-maze assays*, a mask was formed by laser cutting 2-mm-thick ethylene-vinyl acetate foam sheets (*Figure 2—figure supplement 1*; Supplemental CAD file 1). The mask was placed on the NGM surface and the maze was baited with the same amounts of H and M food, which were placed within the maze on the agar surface at the end of each arm. After training, worms were washed and transferred as above to the plate center or the starting point in the T-maze. Accumulation was scored for 60 min at 15 min. intervals. At each time point, preference $I$ was computed as $(N_H - N_M)/(N_H + N_M)$, where $N$ is the number of worms in contact with the food-type indicated by the subscript; worms not in contact with food, and small numbers of escaped worms (~5%), were not counted. Worms were counted by eye with the aid of a tally counter; the experimenter was blind to condition.

### Food dwell-time assays

Dwell-time on food patches was measured as described previously (*Cermak et al., 2020*). Assay plates were standard 10 cm diameter petri dishes filled with low-peptone (0.2 g/L) nematode growth

medium, seeded with 200 mL *E. coli* OP50. Low-peptone medium ensured thin bacterial lawns for improved tracking optics. Roughly circular lawns were created with a spreader, and plates were left to dry overnight before use. For recordings, a single 72 hr old adult animal was picked to an assay plate, allowed to accommodate for 10 min, and then tracked for approximately six hours at 20 fps using the described automated tracking microscope. Prior to tracking, the lawn boundary was annotated by manually circumscribing it with the microscope. This procedure enabled post-hoc determination of when the worm was on or off the lawn.

## System for recording food choice in semi-restrained worms

The system comprised (1) a food delivery system, (2) instrumentation for electrophysiological recording, and (3) a video camera for recording behavior (*Figure 3*).

(1) Food delivery system. Bacteria suspensions, and bacteria-free buffer, were held in 20 mL reservoirs (syringes with plungers removed) fitted with stopcocks. Reservoirs were suspended 50 cm above the chip and connected to it via polyethylene tubing (PE-9) fitted with 1.5 mm diameter × 12.5 mm stainless steel nipples (17 Ga, × 0.500", New England Small Tube, Litchfield, New Hampshire). To minimize settling of bacteria, a miniature magnetic stir bar in each reservoir was agitated periodically during experiments by moving a small hand-held magnet. The layout of the Y-chip was similar in all main respects to a previous design (*McCormick et al., 2011*); feature height was 55 µm. Flow rate in the chip was regulated by a peristaltic pump (model 426-2000, Labconco, Kansas City, MO, USA) attached to chip's outlet port.

(2) Electrophysiology. Electropharyngeograms were recorded by means of electrodes (stainless steel nipples, see above) inserted into the worm port and fluid outlet. In this configuration they were used to measure the voltage differences that occur between the worm's head and tail during pharyngeal muscle action potentials. The vacuum clamp accentuates these voltage differences by increasing the electrical resistance between head and tail. Voltage differences between electrodes were amplified by a differential amplifier (model 1700, AM Systems, Sequim, Washington) and digitized (2 kHz) for later analysis (USB-9215A, National Instruments, Austin, Texas). Digitized recordings were band-pass filtered between 5 and 200 Hz. E and R spikes were detected offline by a manually adjusted threshold in custom data analysis software. Instantaneous pumping rate was computed using a 5 sec. sliding window average (forward looking).

(3) Videography. The worm was imaged using a stereomicroscope (Wild M3C, Leica, Buffalo Grove, Illinois) with a 1.5× objective, and a video camera (VE-CCDTX, DATG MTI, Michigan City, Illinois) with a frame rate of 30 Hz. Individual frames were analyzed by MATLAB scripts (Mathworks, Natick, MA) to extract head angle of the worm as previously described (*McCormick et al., 2011*). First, each frame was thresholded to identify the worm's head and neck region, which was then skeletonized to obtain its centerline (white line, *Figure 3B*) and head angle ($\theta$) was defined as described *Figure 3B*. Values of head angle when the worm was exhibiting presumptive reversal behavior (*Faumont and Lockery, 2006*) where excluded manually, without knowledge of experimental condition.

## Single-worm food choice assays

Bacteria cultured as described above (late stationary phase) were washed three times by centrifugation and resuspension in ~5 mL minimal buffer; the buffer contained (in mM): 1 MgSO$_4$, 10 HEPES adjusted 350–360 mOsm (glycerol). After resuspension of the final wash, bacteria cell density d was measured by diluting a sample of known volume by a factor $1/a$ such that the optical density of the sample, OD$_s$, was in in the linear range relative to cell density, that is, OD$_s$ <1 unit (*Stevenson et al., 2016*). We computed $d$ as $a \times$ OD$_s$; typical values of $d$ were 4–6 units. The final resuspension was then diluted to the desired density for each experiment. We found that OD = 1 corresponds to approximately $2.35 \times 10^9$ and $2.00 \times 10^9$ colony forming units/mL of *Comamonas* and *Simplex*, respectively.

After training, worms were washed and transferred to foodless plates for 1–2 hr of food deprivation. At the start of the assay, the Y-chip was filled with bacteria-free buffer solution and a worm was inserted into the chip by liquid transfer using a syringe fitting with PE tubing and a steel nipple. During a 2 min. accommodation period, both streams carried bacteria-free buffer and the video and electrophysiological recordings were initiated. After accommodation, both streams were switched to particular food types and food densities according to the design of the experiment. In *food-familiarity*

(*Figure 4A*) and *food-density* experiments (*Figure 4C–D*) channels 1 and 4 carried the same food (H or M). In *food quality learning* (*Figure 4B*) and *integration of preference and price* (*Figure 5B*), channel 1 carried H food and channel 4 carried M food at optical densities given in the legends or indicated in the figures. Feeding was recorded for 12 min after food onset. Mean pumping frequency was computed as the number of pumps (paired E and R spikes) divided by total observation time. Preference $(f_H)$ was defined as the fraction of pumps emitted when the tip of the worm's head, where the mouth is located, was in H food, as detected in the synchronized video recording. Specifically, $f_H = N_H / (N \| H \| + N_M)$, where $N_H$ and $N_M$ are the number of pumps in H and M food, respectively; on this scale, $f_H = 0.5$ constitutes equal preference for the two foods or, in economic terms, indifference between the two options. No animals were censored.

### Fitting utility functions to preference data

To fit the CES function (*Equation 11*) to the choice data in *Figure 6C and D*, we estimated the parameters $\beta$ and $\rho$ using two-limit tobit maximum likelihood. Values of the error term were drawn from identical and independent normal distributions (*Andreoni and Miller, 2002*).

### System for calcium imaging of neuronal activity

For recording from AWC neurons, the genetically encoded calcium indicator GCaMP6s was expressed under control of the *str-2* promoter. Late L4 or early adult worms were immobilized in a newly designed microfluidic imaging chip (*Figure 8—figure supplement 1*; Supplemental CAD file 2) based on a previous device in which the worm's nose protrudes into a switchable stimulus stream (*Chronis et al., 2007*). Chip feature-height was 30 μm. The chip was adapted for rapid switching. The switching time constant in a microfluidic chip that is driven by a constant current source (e.g. a syringe pump) is equal to the product of the system's compliance ($C$) and fluidic resistance of the chip ($R$). To reduce compliance we used rigid inlet tubing and to reduce resistance we greatly enlarged the food-carrying channels. An additional advantage of reduced resistance is a reduction in mechanical switching artifacts. Although edible bacteria were perfused through the chip, worms did not feed while being imaged, presumably as a result of being tightly constrained in the chip.

To improve stimulus stability, fluid flow was driven by a syringe pump rather than gravity or pressure. Two of the four syringes in the pump were filled with H food and two with M food, prepared in the same was as for single-worm food choice assays. Syringes were connected to the chip such that if all four channels were flowing into the chip, the cross-sectional flow at the point of confluence near the worm would be $H_2 H_1 M_1 M_2$. Stimulus presentation was automated utilizing a microprocessor to control a pair of two-way solenoid valves (LFAA1201610H, The Lee Company, Westbrook, Connecticut) in series with the syringes $H_1$ and $M_1$; the outer channels $H_2$ and $M_2$ flowed continuously. To present H food, the stimulus pattern was $H_2 H_1 M_1$; to present M food the pattern was $H_1 M_1 M_2$. As all channels flowed at the same rate, each occupied 1/3 of the cross-sectional flow at the worm's position with the result that small fluctuations in the position of fluidic interfaces were kept far from the worm's nose.

A Hamamatsu CCD camera (model C11254) controlled by HCImage was used to capture stacks of TIFF images at 10 frames/sec. Images were analyzed by manually drawing a region of interest (ROI) comprising a tightly cropped segment of the neurite connecting the cilium and soma. Mean background fluorescence was estimated from a 2-pixel thick margin situated 2 pixels outside the ROI in each frame. Absolute neuronal fluorescence was quantified as the mean of the 200 brightest pixels in the ROI, minus mean background. Finally, fluorescence values were expressed as fractional change relative to pre-stimulus baseline absolute fluorescence. Traces shown were not bleach-corrected.

### Microfabrication

Devices were fabricated using standard soft lithography (*Xia and Whitesides, 1998*; *Xia, 2008*). Silicon-wafer masters were created by exposing a layer of SU-8 2025 resist (Microchem, Newton, MA) through a transparency mask and developing the master in a bath of glycol monomethyl ether acetate (PGMEA). CAD files for the Y-chip and the imaging chip are provided in Supplemental CAD file 2 and 3. Masters were treated with tridecafluoro-1,1,2,2-tetrahydrooctyl trichlorosilane (Gelest, Morrisville, Pennsylvania) vapor to facilitate release. Devices were formed by casting polydimethylsiloxane prepolymer (PDMS Sylgard 184, Dow Corning, Corning, NY) against masters. After curing and mold

release, holes for external connections (fluidic inlets and outlets, worm injection, and electrodes) were formed using a 1.5 diameter punch. Devices were exposed to an air plasma then bonded to glass slides (Y-chip) or coverslips (calcium imaging chip).

## Statistics

The following statistical tests were employed according to experimental design. We used *t*-tests to determine whether the mean differed from a specific value, and to compare means from two samples or experimental groups (strains, treatments, optical densities, and price ratios). When the number of experimental groups exceeded two we used a two-factor analysis of variance. When the number of experimental groups exceeded two and individuals within groups were measured successively over time, we used a two-factor analysis of variance with repeated measures. For all comparisons, the center of distributions were their arithmetic mean. For dispersion measures, we used 95% confidence intervals, except in the case of time-series data where we used standard errors of the mean. None of the experimental designs required correction for multiple comparisons. Statistical details for each test (statistical test, effect or comparison tested, definition of units of replication, number of replicates, statistic value, degrees of freedom, p-value, and effect size) are compiled in *Table 2*. These details are cited by the notation "*Table 2*, where *n* is the row number in the table. Because all assays were novel, sample sizes were determined empirically, by noting how variance changed as a function of number of replicates. All replicates were biological replicates. In the case of 'assay plates', a replicate is unique cohort of worms on an assay plate. In the case of 'worms', a replicate is a unique animal tested once. No outliers were excluded.

## Acknowledgements

This research was supported by MH051383 and GM129576 from the National Institutes of Health (SRL). We thank Mei Zhen for the GCaMP-6s::wCherry probe used in imaging experiments, Jonathan Millet for assistance, and Anastasia Levichev for comments on the manuscript.

## Additional information

### Competing interests

Shawn R Lockery: is co-founder and Chief Technology Officer of InVivo Biosystems, Inc, which manufactures instrumentation for recording electropharyngeograms. The other authors declare that no competing interests exist.

### Funding

| Funder | Grant reference number | Author |
| --- | --- | --- |
| National Institute of Mental Health | MH051383 | Shawn R Lockery |
| National Institute of General Medical Sciences | GM129576 | Shawn R Lockery |

The funders had no role in study design, data collection and interpretation, or the decision to submit the work for publication.

### Author contributions

Abraham Katzen, Software, Investigation, Methodology, Writing – original draft; Hui-Kuan Chung, Software, Methodology, Writing – review and editing; William T Harbaugh, Formal analysis, Methodology, Writing – review and editing; Christina Della Iacono, Nicholas Jackson, Stephanie K Yu, Investigation; Elizabeth E Glater, Analysis of volatile organic compounds; Charles J Taylor, Analysis of volatile organic compounds; Steven W Flavell, Supervision, Investigation; Paul W Glimcher, Supervision, Investigation, Writing – review and editing; James Andreoni, Formal analysis; Shawn R Lockery, Conceptualization, Data curation, Software, Formal analysis, Supervision, Funding acquisition, Methodology, Writing – original draft, Writing – review and editing

## Author ORCIDs

Elizabeth E Glater ⓘ http://orcid.org/0000-0003-0205-8209
Steven W Flavell ⓘ http://orcid.org/0000-0001-9464-1877
Shawn R Lockery ⓘ http://orcid.org/0000-0001-8535-7989

## Decision letter and Author response

Decision letter https://doi.org/10.7554/eLife.69779.sa1
Author response https://doi.org/10.7554/eLife.69779.sa2

---

## Additional files

### Supplementary files

• Supplementary file 1. Design of the analysis in *Figure 6A*. Boxes are groups of choice sets in which the price of one food was constant (*shading*) while the other was variable. Numbers are optical density from which price was computed by *Equation 2*. Lower case letters refer to data points in *Figure 5B*.

• Supplementary file 2. Linear correlation equations. These equations show how the economic variables in the left column were computed based on food type, density and preference in trained and untrained animals. The quantities $w_T$ and $w_U$ are defined in *Equation 15* in the main text. The quantities $U_T$ and $U_U$ are, respectively, utility in trained and untrained animals computed according to the CES function as fitted to the data in *Figure 6C and D*.

• Supplementary file 3. Tests of linear correlations between AWC activation and economic variable it might hypothetically represent. Significance of correlations was tested using the *F* distribution. This table shows, for each economic variable tested, the correlation coefficient, value of the *F* statistic, its two degrees of freedom, and the corresponding p-value. No significant correlations where found. For definitions of variables see *Supplementary file 2*.

• Supplementary file 4. T-maze CAD file.

• Supplementary file 5. Y-chip CAD file.

• Supplementary file 6. Imaging chip CAD file.

• MDAR checklist

### Data availability

Data of Figures 2, 4, 5-9 are available as source data files associated with this publication.

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
