## [Editor Report]

In this thought-provoking study, the authors adopt a framework from economic decision-making theory to food choice behaviours in the nematode *C. elegans*. Based on their findings, they propose that worms behave consistently with the Generalized Axiom of Revealed Preference, a classic measure of utility maximization.

---

## [Decision Letter]

**Decision letter after peer review:**

Thank you for submitting your article "The nematode worm *C. elegans* chooses between bacterial foods exactly as if maximizing economic utility" for consideration by *eLife*. Your article has been reviewed by 3 peer reviewers, one of whom is a member of our Board of Reviewing Editors, and the evaluation has been overseen by Ronald Calabrese as the Senior Editor. The following individual involved in review of your submission has agreed to reveal their identity: Doug Portman (Reviewer #2).

All reviewers are excited about the manuscript, however, reviewer #2 raises a major technical concern and the main comments by reviewers #1 and #3 are at a conceptual level. They have discussed their reviews with one another, and the Reviewing Editor has drafted this to help you prepare a revised submission.

Essential revisions:

*Reviewer #1 (Recommendations for the authors):*

1) The authors developed a clever experimental design that allows to precisely monitor the food consumption of worms presented with two streams of different bacterial strains of different nutritious value, H(igh) vs M(medium), and at varying concentrations. Previous work showed that worms prefer H bacteria. Animals are clamped into a microfluidic device where they are free to choose between the two streams by positioning their heads in either one of them; they can also modulate the rate of food ingestion by varying the rate of pharyngeal pumping. Under the assumption that the two types of bacteria are equally bite-sized, the "price" of food is directly related to the dilution of the bacterial suspension. I.e., the more diluted the food is the more pumps need to be invested to intake the same amount. A central finding of the study is presented in Figure 5B, showing that the intake of H monotonically increases with the price ratio of the H and M streams. The authors claim that this finding precludes violations of GARP and therefore strongly supports utility maximization within this GARP framework. However, I wonder how comparable the experimental design is to the GARP experiment shown in Figure 1. Here, subjects are presented with choosing between two option bundles of qualitatively different goods (apples and oranges); humans are consciously aware of their limited budget and the trade-off between the two goods. I don't think that worms are faced with the same task in the present experimental design. (1) there is no evidence that *C. elegans* worms qualitatively distinguish H and M bacteria, i.e. we do not know whether in their brains H and M are represented as qualitatively different objects, like humans perceive apples and oranges. Most likely H produces more of an attractive odorant X; therefore, worms likely are sensitive to the concentration difference of X between the two streams. The AWC imaging experiments are supporting this; they were shown previously to be sensitive to food-borne odorant concentrations in a gradual manner. Thus, AWC neuronal activity does not represent H and M as qualitatively different objects (see Figure 8). Therefore, unlike in Figure 1, worms do not make a trade-off between H and M, they more likely detect high and low concentrations of X. This simple mechanism is sufficient to explain the data in Figure 5B. This would lead also to consistent choices that maximize utility, i.e. foraging in areas that contain more of X which is likely to increase welfare in form of more nutritious food intake. The experiment in Figure 5B could be done using just H bacteria with different dilution. (2) In the experiments, worms are not faced with a binary choice among two bundles of goods that they can take home. They rather continuously forage the two streams by varying pumping rate and waving their heads. This means they are free to choose of how much they can intake from the H and M streams. Worms are also not aware of the duration of the experiment. This means there is no fixed budget, and their choices are not on a fixed budget line. I am therefore not convinced that the statement in lines 531-532: " Under this mapping, the lines in Figure 5C correspond to choice sets like those in Figure 1." is correct.

These (1-2) major differences to the GARP experimental design in Figure 1 largely confuse me and I wonder how useful for understanding neuronal mechanisms is it to rigidly analyse the data with the mathematical tools of this framework. Vice versa, a reader from the field of economic decision making might find the explanations in (1) too trivial and must be similarly concerned of how much the experimental design fits into the GARP framework. The authors spend a good amount of effort justifying the various GARP requisites. I think they should also discuss these obvious differences and provide clear justification why their experimental design is a good worm equivalent to what is shown in Figure 1.

2) The animals show a baseline preference for H bacteria, which is interpreted by the authors as "latent learning". There is no evidence that this preference is due to a latent learning mechanism, it could be just an innate preference. This is also not important for the major conclusions of the paper, so I recommend to avoid this terminology and leave it only to the final discussion.

3) The authors use the term "explicit" learning for the enhanced H preference after training. Explicit learning and explicit memory are differently understood in the field, like abilities or events that humans consciously acquire or remember. I strongly suggest to avoid this term as this could cause some confusion.

Note -> (2-3) was also raised by reviewers #2-3

4) The behavioural defect of cat-2 is interesting but on its own it does not help much to better understand the mechanism that establishes food preference. Since the paper is quite complex already, I would recommend the authors keep these results for future mechanistic studies. – just a recommendation.

5) line 434: you cannot conclude that food quality learning is intact in the Y-chip, since the animals never learned in the chip; acquired food preference upon learning can robustly be measured under Y-chip conditions.

6) Line 461: text abruptly terminates and some important explanatory text to understand the law of demand is missing.

7) Figure 4C-D: what is t=0? Switching from buffer to food streams? What is the time course of pump frequency when no food (OD 0.0) is supplied?

8) Figure 4: I suggest to add legend for colour code in C-F, green = H food, yellow = M food.

9) Line 536: you mean Figure 4?

10) Ca++ imaging experiments lack negative controls. Since switches in microfluidic devices unavoidably come with pressure profiles, proper control experiments should be included.

11) do worms pump/feed in Ca++ imaging assays? Please discuss.

12) Line 964: since worm locomotion needs to perform dorsal to ventral bending patterns under unconstrained conditions, they have a strong internal drive to keep undulating their heads. They just can't keep still. Here, they cannot leave and crawl out of the position in the device. I think the constraints in the device thus could be a major explanation why you don't see corner solutions (see also reviewer #2).

13) line 167: aren't oranges in A double the price of apples?

14) Figure 2 panel B does not show what is described in main text and Figure caption

*Reviewer #2 (Recommendations for the authors):*

1. Bacterial concentration: precision and accuracy in the measurement of bacterial concentration is critical to essentially all analyses carried out here. One of the most fundamental variables in the model, price, is a simple function of this. Exactly how bacterial suspensions of different concentrations/densities are prepared is not clear, but the authors repeatedly use OD600 as a measurement of bacterial concentration (this is explicitly stated in lines 341-2). This has the potential to be a significant concern, as OD600 has a linear relationship with bacterial density only in very narrow range of relatively low density -- above this (certainly ODs of 1 or 3, which are used regularly here) there can be vast deviations from linearity. As noted by one of the references this paper cites (Stevenson et al. 2016), OD600 can even have a parabolic relationship to concentration in some density regimes because of the complexities of light scattering by bacteria. It's hard to tell whether this has been accounted for here – it seems that the authors are assuming, for example, that a suspension of OD 3 has 30x as much food as one of OD 0.1, which almost certainly not case. Further, when considering how this applies to H vs. M food, the authors calculate the difference in mass/OD ratio between H and M food at OD = 1, but this ratio seems very unlikely to hold at different ODs. If bacterial concentration has been mis-estimated because of these problems, it's possible that errors in price would propagate through all of the subsequent analyses. If the authors have indeed accounted for these complications, that's great – in this case, they'd need provide additional details about how this was done. If not, however, additional work will have to be done to figure this out.

2. In Figure 9, the authors provide evidence that AWC physiology does not encode a simple economic quantity like value. I have a few comments/concerns about this analysis. First, if bacterial concentrations are being mis-estimated as described above, it may be the case that the conclusions here will change, possibly dramatically. Second, it looks to me like the red line in 9D indicates a clear relationship between preference and ∆F in trained animals. The statistics might not bear this out because of the limited power of the analysis, but I'm surprised that the authors don't find this at least suggestive of what's going on. Third, authors make an argument for calculating ∆F from its steady state responses, but it would seem that the initial responses might be informative too – this is when relative values are first being assessed. Also, the presentation of the two stimuli with equal times for each doesn't really mimic the steady-state situation when an animal has a strong preference for one of them (in this case, the nose would be exposed to the preferred stimulus for most of the time). Maybe it would be useful to examine AWC responses with different stimulus patterns? Fourth, in rationalizing 9E, authors say that they asked whether AWC might encode "some other economic quantity". How many such quantities were examined? Is this a case where correction for multiple comparisons would be appropriate?

*Reviewer #3 (Recommendations for the authors):*

Introduction: GARP applies to consumer's choices, and importantly, assumes rational agents. The leap to nematodes is huge. Can the authors comment on this?

Line 102: Citation that can potentially be added: Ghosh et al., 2016, Neuron, PMID: 27866800 (for risk assessment).

Line 120: "it is reasonable to expect that *C. elegans* food choices maximize fitness"◊ The notion that an organism's food choices maximize fitness is directly related to the Optimal Foraging Theory. Among the major criticisms this notion has received, is that it relies on the assumption that natural selection and evolution have somehow resulted in a decision-making process that maximizes fitness. However, natural selection has not necessarily led to such a result when it comes to the feeding strategy or food choices of an organism, because it is a process of selection and not a mechanism that produces optimal results, e.g., behaviors. Therefore, it is debatable whether *C. elegans* food choices maximize fitness.

Reviewer #1 and #3 discussed and reviewer #3 recommends to rephrase, e.g. "promote fitness" instead of "maximize".

Line 145: redundant word: during

Line 146: bends, should be plural

Line 150: a period is missing at the end of the sentence.

Line 154: "GARP for worms": although this is the title of this Results section, the section itself is about explaining in detail how utility maximization and its violations can be manifested in graphic representation of bundle choices in experiments with human subjects. This section is not about *C. elegans* and GARP. I suggest revising the title. It is also unclear why this section is part of the Results, since there is practically no result presented.

Line 167: it seems to me that the opposite is inferred from the text (and from Figure 1 legend): apples are half the price of oranges. Is this a typo? - see also reviewer #1

What is the budget of a *C. elegans* nematode? In line 535 and onward authors state that "We found it impractical to standardize the number of pumps to impose a fixed energy budget on each worm because […]", and then they explain that they have been plotting the fractional consumption and equation 5. Does this mean that the budget changes depending on the food source? Can the authors elaborate on this? Are worms' budget units consistent with the goods' prices units? -> see also reviewer #1

Why do the authors consider the abundance of a bacterial strain equivalent to price? In human experiments, the ratio of, e.g., oranges vs apples is one of the variables (amount of fruit per weight), and the dollars per unit is another.

Line 236: Worms were cultured on *E. coli* OP50 before the H/M training took place. What is the nutritional value of this food source compared to H and to M?

Line 253: "If H food smells to the worm more like *E. coli* than M food does, this preference might be explained by the so-called food familiarity effect (Song et al., 2013), in which worms eat familiar food more readily than novel food." In this case, worms do not prefer H food based on its "quality" but based on its similarity to the food source they have been used to. In this case, maybe the sequence of preference choices does not constitute a violation of the utility maximization. In this case, worms do not choose driven by an urge to maximize their fitness, but rather by a tendency to maintain a familiar environment. How does this fit with the authors claims and assumption?

Figure 2B: The caption reads "Mean preference versus time for trained and untrained ceh-36 mutants and N2 controls in open-field accumulation assays", however, I am not sure that this is what the figure shows, given also that the y axis is labeled as "fraction of pumps". Please explain. -> see also reviewer #1

Given the issue with Figure 2B, it is hard to tell, at this point, which plot represents the open field assay and which the maze choice assay, therefore it is hard to compare the two or evaluate the findings in a combined way.

Line 298: "The decision to accumulate in a particular food can be made, at least in part, before the animal enters the patch." ◊ I think that, for the sake of accuracy, the decision in question is not about deciding to accumulate; a worm does not decide to accumulate, but rather to approach or reach or forage on food H or M. How do the authors comment on this?

Have the Authors performed the open-field accumulation assay in the presence of sodium azide?

The authors speak about latent learning (=a type of learning which is not apparent in the learner's behavior at the time of learning, but which manifests later when a suitable motivation and circumstances appear) in case of the non-trained animals, and about explicit learning (=a more conscious process where the individual makes and tests hypotheses in a deliberate search for answers) in case of the trained animals. This implies a dipole scheme of latent vs explicit. However, the authors do not justify the use of these two terms, nor do they provide evidence that in non-trained animals the learning is latent; in addition, there is no justification for the use of the term explicit in the second case. The worms have been simply conditioned or not conditioned, respectively. Even in lines 307-309 when they cite two papers by Worthy et al., stating that "Explicit food quality learning in *C. elegans* is formally equivalent to a type of classical conditioning in which an association is formed between the mélange of odors characteristic of particular bacteria species (Worthy, Haynes, et al., 2018; Worthy, Rojas, et al., 2018)", this is perplexing, because I am not sure that these papers claim or demonstrate that explicit learning in *C. elegans* is equivalent to classical conditioning. Despite that, the authors state that they are trying to find the locus of explicit food quality learning (line 259). I think the use of the two terms, explicit and latent, should be better justified or other terms should be used instead. -> see also reviewers #1/2

In relation to that, in lines 323-333, it is implied that food learning occurs while the animals forage on the food patches. What is it exactly that worms learn? Is this some form of non-associative learning? In addition, the effect on their preference developed as they forage on a specific food patch is something that should be present in both trained and non-trained animals. The authors state that "It is possible, therefore, that food quality learning in cat-2 mutants is impaired simply because they spend less time in the food, hence have less experience of it", line 321. Does this mean the authors believe that food quality learning occurs during the worms' stay on the food patch during the open field or maze assay and not during training?

Have the authors tested only two types of food, i.e., two bacterial strains characterized as medium and high quality? If so, then the reported results could be due to other properties of the bacteria and not their nutritional value (case-specific results). Note that the nutritional value (energy, joules) is what the authors take into account in equation 1. Wouldn't their findings be stronger if the observed preferences were repeated for other pairs of food choices, i.e., by testing other/more bacterial strains?

Figure 4C and 4D: what is the difference between the two? Do they refer to M and H?

Similarly for 4E and 4F: do different colors (green and gold) represent different foods e.g., H and M? Please clarify.

Line 524: Maybe these data should be shown, as they constitute an important link in the authors' train of thought.

Line 550: Can the authors explain why this is expected to be a gaussian distribution?

Line 551-554: Is there a plot/figure to illustrate this?

Line 561: I don't think this is a particularly conservative null hypothesis; in most cases in behavioral assays the null hypothesis is exactly this: that animals choose randomly, i.e., that the treatment has no effect.

Lines 604-606 and Figure 6A: I do not understand why e and d are grouped both in the group in which H changes and in the one H stays constant. Can the authors please explain?

Line 673: "worms require more of it to be satisfied": How do the authors define or measure satisfaction in this case? Without a definition this sentence is misleading.

Line 700: eliminate or support?

Line 766-767, and the entire paragraph: Authors say that "As exogenous activation of AWC produces a bout of reverse locomotion (Gordus et al., 2015), its response to food offset promotes reversals, leading to increased retention patches. Our imaging data suggest that AWC-mediated retention would be stronger for H than M food, promoting greater preference for H food, as seen in Figure 2B-D." The authors rely on their finding that AWC was more strongly excited by H than M food, to jump to the conclusion that "AWC-mediated retention would be stronger for H than M food, promoting greater preference for H food". In my view, this last claim could constitute a well-defined hypothesis to be tested, and not a conclusion based on their findings, even in combination to Gordus et al. In fact, it is Gordus and colleagues who highlight in their 2015 paper that although the activation pattern of AWC is deterministic ("highly reliable"), the resulting reversing behavior is, by contrast, probabilistic: "The AWC calcium response, which is likely correlated with depolarization, is highly reliable from trial to trial, even after dozens of odor presentations (Larsch et al., 2013). By contrast, the reversal response is probabilistic. Even under well-controlled conditions, animals may or may not reverse on individual trials, regardless of the strength of the AWC calcium response (Larsch et al., 2013)" and that "Thus, the variability in the behavioral response results from variable transmission of information from the AWC sensory neuron to AVA command neurons." Based on the above, I do not understand the authors' claim. Maybe they can run an experiment to show that the claim is valid? The results presented in Figure 2, although they show increased retention, they do not allow the conclusion that the animals with stronger AWC activation are the ones that present increased retention. That individuality in the response is, in my view, what Gordus et al. indicate in their work.

Similarly, in the following paragraph, based only on the strength of AWC activation, they conclude that "retention in H food patches would be stronger in trained than untrained animals, promoting greater preference for H food after training". However, stronger activation of AWC does not mean stronger retention (behavioral outcome), based on Gordus et al.

Line 1114: The mask was baited or the maze-shaped NGM area? Where there any escaping animals? Where there any animals that did not choose either of the two (e.g., not reached neither H or M in the allowed time)? How long were the worms allowed to make a decision inside the maze? Were there any animals censored, and if so, does this affect the results? Did the authors alternate the position of H/M food in the two maze arms to eliminate any side preference effect?

Line 914: what do the authors mean by "infra-human species"? Maybe non-human?

Line 919: "literally guarantees", and elsewhere in the text: the authors tend to express their claims using absolute language (see also title, "exactly as"). I feel that it might be more appropriate if the language used is firm and confident, as the authors see fit, but milder.

Lines 914-922, and elsewhere in the text: The authors are trying to convince the reader that *C. elegans* makes consumption decisions in the same way that a human consumer does. What's more, they claim that such a human-conforming behavior can be captured in nematodes by just mapping the neuronal circuit that steers head bending. This is in the core of the paper, and in my view, it is problematic in many ways. The axioms of the revealed preference theory are justified by the assumption that humans are rational agents. This means that they make rational decisions. Do the authors claim the same for *C. elegans* nematodes? In parallel, the fact that *C. elegans* feed on two different food sources and they choose one over the other in a way that the goods qualify as substitutes, is a finding presented as of extreme importance. This is indeed an interesting conclusion, well supported by the data. It becomes more interesting because of the untangling of the neuronal circuit involved. At the same time, this is really not an unexpected result, although providing evidence for it is definitely useful. Any animal that feeds on multiple food sources would alternate between two of them, based on their availability, the effort required, the nutrition provided. Indeed, the authors themselves do not claim that this is unexpected. However, they claim that they "break grounds" because worms' behavior appears to conform with utility maximization principles and can be described with human-centered terms of substitute goods. This is misleading, first, because of the agent rationality assumed, which we have no basis for, and second, because identifying two goods as a substitutes pair is not ground-breaking on its own. Behavioral economics labels pairs of goods as such in order to move on with more complicated claims, theorems and analyses. The authors are interpreting *C. elegans* feeding behavior using behavioral economics terms that sound extravagant when speaking of nematodes, but in reality, their findings are not extravagant and certainly do not need to be dressed as such in order to be significant. In this reviewer's view, the findings presented in the manuscript are interesting and they constitute a significant contribution. Attempting to attribute to them a dimension disproportionate to their real depth, dampens the initial enthusiasm in an unnecessary way.

Line 979: What are the worms maximizing? This is a question that the authors admit comes up inevitably, but interestingly (and honestly, of course) at the same time they state that they do not have a plausible answer to it. Therefore, even the notion of utility (which is supposed to be maximized) remains obscure. In my understanding, this confirms the fundamental problem with the way this study is presented. The authors are not working toward testing a stated hypothesis (e.g., nematodes' behavior is such that maximizes x), but they rather quantify a nematode behavior in a way that fits the equations used in behavioral economics. What they don't take into account is that, sometimes, even if phenomenon A can be phenomenologically described by a set of equations that has been developed to describe (part of) phenomenon B, this does not mean that A is explained by the same physical or biological principles that steer phenomenon B. This is even more prominent if the organism involved in A is so very different regarding its brain faculties and societal construct than organism B. And even more so if the researchers fail to provide a satisfactory answer for the biological (in lack of maybe a psychological, societal or other) explanation for this behavior.

Finally, a few thoughts, which the authors are not requested to address, neither in the revised manuscript, nor in their rebuttal letter: Any scientific paper creates, upon its publication, new premises. Hence, even if the authors do not continue this work, there will be others who will, one way or another. I wonder what the premises of the present manuscript are/will be. What is the precedent created here? That *C. elegans* is a valid system to test behavioral economics hypotheses and apply axioms? That *C. elegans* can be used as a model system to understand, and maybe predict, human economic decisions? To establish *C. elegans* as an experimental system for behavioral economics? To what extend and to what end? The authors state in the introduction, "A positive result [of this study] would establish a simple experimental system in which neuronal activity correlated with utility could be manipulated both physiologically and genetically to establish behavioral causality. Such a finding would also be interesting from a comparative perspective, extending the domain of utility-based decision making far beyond the boundaries of organisms that are generally considered to have cognition." Do they envision their work as the precursor for manipulating or directing human decision making? I truly appreciate that the authors made me reflect on all these (and many more) issues, and I trust that this work, when published, will instigate a lot of interesting discussions. The way we do research and the questions we choose to pose and answer can have a lot of impact, that goes beyond the citations a paper might receive. To conclude, I would like to share with the authors a note by Claude Shannon, the founder of Information Theory, written back in 1956: https://ieeexplore.ieee.org/stamp/stamp.jsp?arnumber=1056774. He comments on the attempt to apply information theory to more distant fields, like biology, psychology, physics, etc. The term information theory can potentially be substituted with other terms with interdisciplinary appeal, such as game theory, behavioral economics, microeconomics, etc.

[Editors' note: further revisions were suggested prior to acceptance, as described below.]

Thank you for resubmitting your work entitled "The nematode worm *C. elegans* chooses between bacterial foods as if maximizing economic utility" for further consideration by *eLife*. Your revised article has been evaluated by Piali Sengupta (Senior Editor) and a Reviewing Editor.

The revised manuscript has undergone extensive discussion among the reviewers. While reviewers #1 and #2 are enthusiastic about publishing it in *eLife*, reviewer #3 has raised a strong concern that the Y-chip configuration precludes a GARP conclusion. We are excited that your study is thought-provoking and will likely spark a debate among *eLife* readers. However, it is essential that major concerns about the appropriateness of your approach be addressed before publication.

Please find below excerpts from the reviewers' comments that we request be addressed in a revised version of the main text. We are aware of the time pressure you have communicated to us and will do our best to expedite the process once we receive a revision, along with a point-by-point response.

*Reviewer #1 (Recommendations for the authors):*

The authors have made substantial efforts in addressing my previous comments. They show that H and M bacteria release different volatile odorant profiles, indicating that worms potentially sense them as distinct olfactory objects, also supported by the pharyngeal pumping experiments using the food familiarity paradigm. Moreover, they improved the discussion to address some of my conceptual concerns when comparing their assay design with the human apple-orange experiment. However, the data are just indicative and to fully address my concern they would need to show, both with neuronal activity measurements, (I) that worms perceive H & M as distinct odorant objects in the Y-chip and (II) that worms calculate utility based on the relative abundance of H and M. I did not mean to say in my previous comment that (I) and (II) have to be conscious processes (like in humans), but they should have explicit neuronal representations. I think the alternative and simpler model of detecting concentration differences between odorant X in the Y-chip is still a possibility. I understand that addressing this would be out of the scope of the current manuscript and could be interesting for future experiments. At the current stage, I find the here reported approach applied to *C. elegans* creative, novel, and thought-provoking and since I cannot detect (within my expertise) any technical problems in how the framework is implemented, I would enthusiastically support publication in eLife.

*Reviewer #2 (Recommendations for the authors):*

The authors have nicely addressed my previous concerns, which were primarily technical. I find the paper to be a fascinating, rigorous, and creative approach to understanding decision neuroscience using *C. elegans*.

*Reviewer #3 (Recommendations for the authors):*

The authors have extensively revised large parts of the manuscript, especially in the Results and Discussion sections. The revised manuscript also includes updated figures and captions, new supplementary figures, etc. The manuscript is in much better shape, and much more information is provided, information that is critical to evaluate the results, their interpretation, and the final conclusions. The addition of new authors and expertise, as well as the change in the title are very welcome, too. Many of my concerns were addressed, and I appreciate the detailed explanations provided in some of the authors' responses. My understanding of their work is now much better. Nevertheless, the work continues to present fundamental problems. These are reflected mainly in comments #1, #5, and #11, which can be considered my major concerns. The rest of the comments can be considered minor or easier to answer.

1. Comment 1 of first revision and response 3.1:

The authors write in their rebuttal letter that they adopt the stance that if the agents' behavior adheres to the GARP, then they are rational (and not that rationality is a prerequisite). In their response, the authors list 3 papers to support the stance they claim to adopt, and all three of them have to do with humans, and the work reported has to do with reduced or altered observed rationality because of trauma, aging, etc. Displaying reduced rationality is fundamentally different than potentially lacking the very ability to be rational. I still feel that the issue of rationality needs to be addressed clearly in the manuscript. I explain:

In the revised manuscript, apart from two references to published work on *C. elegans* observed occasional rationality with many exceptions (lines 106-110), the only important part that addresses rationality is the new text in lines 228-234. Here, the authors mention (line 228) that "tests of adherence to GARP have been utilized to assess the degree to which decision-making agents are rational in the economic sense", and later in the paper, they claim that nematodes adhere to GARP. The authors need to make clear whether they claim rationality for *C. elegans*, based on their findings, and if so, how is this rationality defined. And in addition, they need to make clear that whatever rationality is claimed for worms, it is not to be confused with what is broadly understood as rationality but is rather a term to describe a mathematically defined concept, the criteria for which are met by nematodes. If this is not absolutely clear, then the reader can be easily misled.

5. Line 301 (related also to response 3.18): " [The preference of naïve worms] could be the result of prior experience or developmental maturation": I do not understand this. What kind of prior experience? If I understand correctly, the worms have been grown in the lab for their entire life, and for a number of prior generations. Have they been fed anything else than *E. coli* during some part of their life? Have they been exposed to H or M food before? Have their ancestors been exposed to H or M food? If not, then the "prior experience" argument cannot possibly apply. If yes, then this should be mentioned, as it would dramatically change the perspective. As for developmental maturation: If the authors believe this is true, then assessing worms ranging from L3 to Day 1 is obscuring, and distinct age cohorts should be assayed separately (L3, L4, Day 1). Moreover, if this were true, it would mean that older or younger animals prefer, for example, H. Therefore, the ratio of younger/older animals in the tested population could skew the results this way or the other. How do the authors comment on that? As for the trained worms, acquiring a preference because of sampling the two foods during the 60 min training assay seems the most reasonable thing to assume. However, the authors claim that this scenario does not apply, because of the findings of the familiarity experiments in the Y chip, but it is not clear to me why this is the case. Besides, the conclusion in line 522 that the foods are qualitatively distinct, would explain the development of a preference during the 60 min of the assay.

11. The new information provided in the revised manuscript on the budget constraint and the price of goods H and M, and how the budget constraint is imposed because of the Y-chip design, allowed me to get a new perspective of the Y-chip experiments. In consequence, several concerns arise. The process is as follows, roughly: A worm is placed in the chip, and two streams, one of H and one of M food, are provided. The preference is revealed by the worm bending its head toward the M or the H stream, and by performing pumps. The combination of pumps of H or M food is the preferred bundle. The whole process lasts for 12 mins. Therefore, it takes 12 min for the worms to behave in a way that reveals their preference. During these 12 mins, the preference is revealed through a sequential decision-making process, in which the worms sample the goods.

i) The sequential nature of this process is fundamentally different than the one-time decision made by human subjects in analogous experiments. In those, humans select the bundle of preference in a single decision. Nematodes, however, make many decisions, sequentially, that define the preferred bundle. How do the authors believe that this affects the interpretation of the results, with respect to adhering to the GARP prerequisites? I posit that this difference can have many implications as someone is trying to determine whether nematodes meet the same prerequisites that humans meet but while under very different experimental conditions. Some additional methodological concerns are listed below.

ii) During the sequential sampling, the worm needs to bend its body toward this or the other side, in order to perform a pump. Consequently, switching between sides, therefore foods, has a cost for the worm (note that bending just the head does not allow for switching between the two flows, as evident in Video 1). How do we know that this cost does not affect its decision-making with respect to switching between foods? In addition, even though the authors say that the body bending while in the chip resembles the body bending during the sinusoidal locomotion on a GNM plate, in Video 1 we can clearly see the worm bending multiple times toward one side before switching to the other side. This is not what happens during regular locomotion. This can be related to the worm preferring to feed on one food, but also to the cost of the effort required in order to bend to the other side. Given the constraint along the mid-body, this effort seems not to be negligible.

iii) How do we know that the worm is not biased by the content of the first few pumps? Is there a correlation between the first few pumps and the final preferred bundle?

iv) The initial body orientation of the worm is not controlled, as I would guess. How do we know that it does introduce a bias with respect to the initial conditions of the system?

The fact that the worm develops a preference over the course of 12 min, as it is tasting (sampling) the two foods, cannot be overlooked. According to the literature, this can be enough time, for worm standards, to develop behavioral plasticity. How could this affect the "revealed preference" and the adherence of the worm to the GARP prerequisites? Especially since human subjects, who participate in carefully controlled experiments, do not undergo a similar (sequential) process.

v) The (iv) relates to the development of a preference during training in a plate with patches, for the accumulation assay, see comment #5. The authors are reluctant to admit that worms can develop a preference because of sampling the two foods during the 60min training assay. They attempt to justify this reluctance based on the results of the familiarity experiments on the Y-chip. As mentioned in comment #5, I am not sure how the familiarity experiments lead to the conclusion that worms do not develop a preference during training. Does this have to do with the authors not acknowledging that worms might be developing a preference during the Y-chip experiments?

All the above (i-iv) leave the key claim "Overall, this methodology results in decision-making paradigm that is formally analogous to GARP experiments in humans", line 1054, unsupported.

To conclude: the revised manuscript is in much better shape than the original draft, and I recognize the authors' efforts to respond to the reviewers' many comments. The amount of work performed is impressive. However, this paper still presents fundamental problems. These are reflected mainly in comments #1, #5, and #11. Worm's adherence to GARP prerequisites is proposed to be the case, but the methodologies through which this conclusion is drawn and the interpretation of some of the findings are problematic. Trying to adjust human GARP behavioral experiments so that they can be run with nematodes does not seem to work, in this reviewer's opinion. Many of the parameters that experimenters are trying to control when studying humans, in order for them to be able to reliably claim adherence to GARP, are not controlled in the nematode experiments presented here (comment #11), something that can dramatically affect the interpretation of the results. Furthermore, important clarifications need to be made so that the reader is not misled (comment #1), and some explanations the authors provide for some of the observed behaviors are inadequately substantiated (comment #5).

In light of the above, even if the authors addressed everything else satisfactorily, with concern #11 unresolved, I do not see how the main claim can be sufficiently supported and how this paper can be published in *eLife*. The claims that the authors are making are such that their substantiation should be above any ambiguity.

Sequential decision-making in the Y-chip is a fundamental concern. Here is why: The main claim the authors are making is that *C. elegans*' behavior abides by the GARP requirements. They claim so, based on their interpretation of experimental findings. By implementing a certain experimental design, they experimentally pose a question to the system, and the system (nematode consumers) responds by demonstrating a certain behavior, and this behavior is interpreted as GARP behavior because presumably, it is similar to the behavior displayed by the human counterparts. As a reference, when behavioral economists run experiments with human subjects, they apply a certain experimental design, they experimentally pose a question to the system (human consumers), and the system responds by demonstrating a certain behavior, and this behavior is defined as GARP behavior. Note that the experimental design followed by behavioral economists is very strictly controlled, to make sure that no undesired variables interfere with the results. However, the Y-chip experiments pose a different question to nematodes, than the question posed to human consumers: the Y-chip asks what the outcome is of a sequential decision-making process, whereas the GARP-defining human experiments ask what is the outcome of a one-time decision-making process. In other words, the problem is that Katzen et al. introduce a variable that is not present in the experimental design used in behavioral economics to decide whether the subject behaves as GARP: the variable of time. The worms display sequential decision-making, but GARP is supposed to explain a one-time decision-making behavior, not a sequential one. This discrepancy, ignoring the extra variable in the Katzen et al. experiment, is problematic and potentially very misleading.

I am particularly concerned because of the potential impact of the claim that authors make: nematodes make economic decisions that can be described by the same set of principles that describe human economic decisions. For such a huge claim to stand, it has to be supported in a concrete way, without gaps in the thought process.

Bottom line: I understand that the experimental design is what it is and that there is no way the Y-chip experiments are performed differently so that the worms could make a one-time decision. Therefore, I see two options: (A) In the current version of the manuscript, because of this discrepancy, the claim is weak and is not adequately supported, in my view. Therefore, I cannot suggest the manuscript be published in *eLife*. (B) The authors update the manuscript to admit the discrepancy, and they make it absolutely clear to the reader that the nematode system makes its decision according to a process very different than the one that human systems make their decision, in corresponding reference experiments that decide the subjects' GARP behavior. If the authors do that, then everything will be transparent, and the readers can decide for themselves whether the claim stands or not. In such a case, I am fine with publishing the work in *eLife*, if such is the suggestion made by everyone else.

---

## [Author Response]

Reviewer #1 (Recommendations for the authors):1) The authors developed a clever experimental design that allows to precisely monitor the food consumption of worms presented with two streams of different bacterial strains of different nutritious value, H(igh) vs M(medium), and at varying concentrations. Previous work showed that worms prefer H bacteria. Animals are clamped into a microfluidic device where they are free to choose between the two streams by positioning their heads in either one of them; they can also modulate the rate of food ingestion by varying the rate of pharyngeal pumping. Under the assumption that the two types of bacteria are equally bite-sized, the "price" of food is directly related to the dilution of the bacterial suspension. I.e., the more diluted the food is the more pumps need to be invested to intake the same amount. A central finding of the study is presented in Figure 5B, showing that the intake of H monotonically increases with the price ratio of the H and M streams. The authors claim that this finding precludes violations of GARP and therefore strongly supports utility maximization within this GARP framework. However, I wonder how comparable the experimental design is to the GARP experiment shown in Figure 1. Here, subjects are presented with choosing between two option bundles of qualitatively different goods (apples and oranges); (a) humans are consciously aware of their limited budget and the trade-off between the two goods. I don't think that worms are faced with the same task in the present experimental design. (b) (1) there is no evidence that *C. elegans* worms qualitatively distinguish H and M bacteria, i.e. (c) we do not know whether in their brains H and M are represented as qualitatively different objects, like humans perceive apples and oranges. (d) Most likely H produces more of an attractive odorant X; therefore, worms likely are sensitive to the concentration difference of X between the two streams. (e) The AWC imaging experiments are supporting this; they were shown previously to be sensitive to food-borne odorant concentrations in a gradual manner. Thus, AWC neuronal activity does not represent H and M as qualitatively different objects (see Figure 8).

1.1.a. Economists do not regard conscious awareness of the budget as a prerequisite for testing adherence to GARP. In fact, human participants are not informed of the budget constraint in GARP tests (Harbaugh, Krause and Berry, 2001; Camille *et al.*, 2011). That is because the purpose of this constraint is solely to ensure that every option in a choice set cost the same amount, thereby establishing trade-offs between goods. This point is now explicitly made in the text, “Participants need not be made aware of the budget constraint; it can remain implicit in the set of available choices presented.”

1.1.b. The experiment of Figure 4A provides evidence for a model in which worms *do* qualitatively distinguish H from M bacteria. According to the alternative model (see d), the foods are qualitatively similar but one food generates a more intense perception than the other, perhaps by emitting more of the main compound (*X*) by which H and M are detected. Worms familiar with H or M food pump at the same rate in them (Figure 4A, left bars), indicating H and M are not perceptually distinct, which means they emit the same amount of X. In light of this result, the alternative model obviously fails to predict the familiarity effect for, as H and M are indistinguishable, neither of them can be unfamiliar to the worm. From the fact that worms pump at different rates on H and M when cross-cultured on the other food, we can conclude they are qualitatively distinct. The text has been edited to make this point, “The food familiarity effect supports a model in which H and M foods are qualitatively distinct to the worm. According to the alternative model, the foods are qualitatively similar…”.

1.1.c. The nature of the physical representations underlying the distinct perceptions is at present unknown, although neuronal representations are the obvious candidate. Nevertheless, this question is irrelevant to conclusions drawn from adherence to GARP as GARP makes no assumptions about choice mechanisms.

1.1.d. The manuscript now includes new data showing a chemical analysis of the volatile organic compounds emitted by the H and M bacteria used in our study (Table 1). Each of these bacteria emit many such compounds. This finding provides the olfactory substrate for qualitative differences in the worm’s perception of H and M bacteria, similar to the qualitative differences by which people distinguish between goods (e.g., colors and flavors of fruits in Figure 1). Three of the five main released compounds have been identified as chemoattracts in *C. elegans*; the others have yet to be tested. Surprisingly, M releases more of each attractant than H. Therefore, if worms choose based solely on odors, M should be more attractive than H.

1.1.e. We did not mean to imply that the neuronal representation of H and M food is restricted to a single chemosensory neuron class. In particular, we stated, “*C. elegans* has 12 pairs of anterior chemosensory neurons that respond to bacteria conditioned medium (Zaslaver *et al.*, 2015).” This complexity provides the substrate for population-code representations of H and M, which could make *C. elegans* an effective modal for investigating population codes in decision making. We now state, “In perhaps the most likely scenario, the worm’s internal representation of H and M food is distributed across some or all of the 12 pairs of foods sensitive chemosensory neurons in this organism. As a first step, we focused on one of these, AWC…”

Therefore, unlike in Figure 1, (f) worms do not make a trade-off between H and M, they more likely detect high and low concentrations of X. This simple mechanism is sufficient to explain the data in Figure 5B. (g) This would lead also to consistent choices that maximize utility, i.e. foraging in areas that contain more of X which is likely to increase welfare in form of more nutritious food intake. (h) The experiment in Figure 5B could be done using just H bacteria with different dilution. (2) (i) In the experiments, worms are not faced with a binary choice among two bundles of goods that they can take home. They rather continuously forage the two streams by varying pumping rate and waving their heads. This means they are free to choose of how much they can intake from the H and M streams. (j) Worms are also not aware of the duration of the experiment. (k) This means there is no fixed budget, and their choices are not on a fixed budget line. I am therefore not convinced that the statement in lines 531-532: " Under this mapping, the lines in Figure 5C correspond to choice sets like those in Figure 1." is correct.

1.1.f. Please see response 1.1.b.

1.1.g. We have argued above that the single-odor model is inconsistent with the familiar food effect (Figure 4A) and is not supported by our chemical analysis of released compounds. Furthermore, whereas efficient foraging generally increases food intake, it does not necessarily maximize utility. In economics, utility maximization has a specific technical meaning: that the individual’s choices over distinct commodities are consistent and transitive across a range of price-budget structures, i.e., choice sets. The proposed foraging example does not allow for this type of decision making because it entails neither price nor budget.

1.1.h. Doing the experiment of Figure 5B with a single species of bacteria would be interesting, but that would be very different paper, one that might fit with the probability matching literature (the matching law), whereas the present paper is concerned with rational choice between different goods.

1.1.i. We agree worms were free to eat as much as desired. This would be a potential concern relative to establishing a fixed budget only if a fixed budget were required in order to ensure that trade-offs occur. But in our study, trade-offs were introduced by the geometry of the Y-chip. A conventional budget constraint was not required.

1.1.j. We agree that worms are not aware of the duration of the experiment. Not only is time awareness an inappropriate assumption for nematodes, awareness of time is not necessary in order to establish the required trade-off between H and M food. That is because the trade-off is established by the geometry of the Y-chip.

1.1.k. The purpose of the budget constraint in human experiments is to ensure that there are trade-offs between goods in each choice set. In our experiments, however, trade-offs are ensured by the geometry of the Y-chip. As the two fluid streams are in contact and completely fill the food channel, the worm is always feeding in one stream or the other. Therefore, for every second spent pumping in H food, the worm forgoes a second pumping in M food, and *vice versa*. Therefore, there are trade-offs in the experiment and, because of these, all possible choices lie one of the budget lines. In human experiments the trade-off between consuming more of one good at the expense of less of another is enforced by the budget constraint. In *C. elegans* experiments, however, imposing a budget constraint is not necessary, as trade-offs are enforced by the Y-chip. The chip contains only two streams, one for H and one for M food, such that for every pump spent on H food, the worm necessarily forgoes a pump spent on M food, and *vice versa*.

These (1-2) major differences to the GARP experimental design in Figure 1 largely confuse me and I wonder how useful for understanding neuronal mechanisms is it to rigidly analyse the data with the mathematical tools of this framework. Vice versa, a reader from the field of economic decision making might find the explanations in (1) too trivial and must be similarly concerned of how much the experimental design fits into the GARP framework. The authors spend a good amount of effort justifying the various GARP requisites. I think they should also discuss these obvious differences and provide clear justification why their experimental design is a good worm equivalent to what is shown in Figure 1.

1.1.l. We have added such a paragraph to the Discussion.

2) (a) The animals show a baseline preference for H bacteria, which is interpreted by the authors as "latent learning". There is no evidence that this preference is due to a latent learning mechanism, it could be just an innate preference. This is also not important for the major conclusions of the paper, so I recommend to avoid this terminology and leave it only to the final discussion.3) The authors use the term "explicit" learning for the enhanced H preference after training. (b) Explicit learning and explicit memory are differently understood in the field, like abilities or events that humans consciously acquire or remember. I strongly suggest to avoid this term as this could cause some confusion.

1.2.a. We agree that referring to the acquisition of baseline preference for H bacteria as *latent learning* overstates what is known. As originally stated, hatchlings do not prefer H bacteria over others (Shtonda, 2006), so the baseline preference for H cannot be called innate. However, we do not know whether the acquisition of this baseline preference during development is the result of learning about standard laboratory food (strain OP50) that then generalizes to H food, or simply the result for developmental changes in the nervous system. We have replaced the term latent learning with acquisition of *latent preferences*.

1.2.b. We now refer to preferences acquired through food quality training as *trained preferences*, thereby eliminating the confusing term *explicit.*

4) The behavioural defect of cat-2 is interesting but on its own it does not help much to better understand the mechanism that establishes food preference. Since the paper is quite complex already, I would recommend the authors keep these results for future mechanistic studies. – just a recommendation.

1.4. We propose to keep this result as it supports the hypothesis that food quality can be learned by adults.

5) line 434: you cannot conclude that food quality learning is intact in the Y-chip, since the animals never learned in the chip; acquired food preference upon learning can robustly be measured under Y-chip conditions.

1.5. Good point; we agree completely. We now conclude that the effects of food quality training are detectable in the Y-chip.

6) Line 461: text abruptly terminates and some important explanatory text to understand the law of demand is missing.

1.6. Apologies. the missing text has been inserted.

7) (a)Figure 4C-D: what is t=0? Switching from buffer to food streams? (b) What is the time course of pump frequency when no food (OD 0.0) is supplied?

1.7.a. Legend has been revised to define t = 0.

1.7.b. The OD 0 trace has been added.

8) Figure 4: I suggest to add legend for colour code in C-F, green = H food, yellow = M food.

1.8. Done.

9) Line 536: you mean Figure 4?

1.9. Fixed.

10) Ca++ imaging experiments lack negative controls. Since switches in microfluidic devices unavoidably come with pressure profiles, proper control experiments should be included.

1.10. A typical buffer-only control imaging trace has been added (Figure 9—figure supplement 1).

11) Do worms pump/feed in Ca++ imaging assays? Please discuss.

1.11. They do not. We now say this in the methods section.

12) Line 964: since worm locomotion needs to perform dorsal to ventral bending patterns under unconstrained conditions, they have a strong internal drive to keep undulating their heads. They just can't keep still. Here, they cannot leave and crawl out of the position in the device. I think the constraints in the device thus could be a major explanation why you don't see corner solutions (see also reviewer #2).

1.12. We agree worms are probably incapable of generating *perfect* corner solutions, for the reason the reviewer suggests. However, they are capable of generating close approximations to corner solutions. See for example point *g* for trained worms in Figure 5C, where H food is ten times denser than M food; the worm’s choice here is a close approximation to a corner solution. Accordingly, the device is not a limiting factor in observing corner solutions. This point has been added to the text.

13) line 167: aren't oranges in A double the price of apples?

1.13. Silly mistake, thanks for catching it.

14) Figure 2 panel B does not show what is described in main text and Figure caption

1.14. Fixed.

Reviewer #2 (Recommendations for the authors):1. Bacterial concentration: precision and accuracy in the measurement of bacterial concentration is critical to essentially all analyses carried out here. One of the most fundamental variables in the model, price, is a simple function of this. Exactly how bacterial suspensions of different concentrations/densities are prepared is not clear, but the authors repeatedly use OD600 as a measurement of bacterial concentration (this is explicitly stated in lines 341-2). This has the potential to be a significant concern, as OD600 has a linear relationship with bacterial density only in very narrow range of relatively low density -- above this (certainly ODs of 1 or 3, which are used regularly here) there can be vast deviations from linearity. As noted by one of the references this paper cites (Stevenson et al. 2016), OD600 can even have a parabolic relationship to concentration in some density regimes because of the complexities of light scattering by bacteria. It's hard to tell whether this has been accounted for here – it seems that the authors are assuming, for example, that a suspension of OD 3 has 30x as much food as one of OD 0.1, which almost certainly not case. Further, when considering how this applies to H vs. M food, the authors calculate the difference in mass/OD ratio between H and M food at OD = 1, but this ratio seems very unlikely to hold at different ODs. If bacterial concentration has been mis-estimated because of these problems, it's possible that errors in price would propagate through all of the subsequent analyses. If the authors have indeed accounted for these complications, that's great – in this case, they'd need provide additional details about how this was done. If not, however, additional work will have to be done to figure this out.

2.1. Bacteria cell density in food suspensions was always measured in the linear range of OD relative to cell density (OD < 1). For suspensions outside this range, OD was measured in diluted samples. The Materials and methods section has been revised accordingly.

2. In Figure 9, the authors provide evidence that AWC physiology does not encode a simple economic quantity like value. I have a few comments/concerns about this analysis.a) First, if bacterial concentrations are being mis-estimated as described above, it may be the case that the conclusions here will change, possibly dramatically.

2.2.a. Please see response 2.1.

b) Second, it looks to me like the red line in 9D indicates a clear relationship between preference and ∆F in trained animals. The statistics might not bear this out because of the limited power of the analysis, but I'm surprised that the authors don't find this at least suggestive of what's going on.

2.2.b. As the p-value for this correlation is 0.177, we prefer to be conservative and leave the text as it stands.

c) Third, authors make an argument for calculating ∆F from its steady state responses, but it would seem that the initial responses might be informative too – this is when relative values are first being assessed.

2.2.c. The initial transients died out in 1 min or less, well before most animals had begun to feed. As feeding is almost certainly required to assess value, at least in the case of untrained animals, we believe these transients are likely irrelevant.

d) Also, the presentation of the two stimuli with equal times for each doesn't really mimic the steady-state situation when an animal has a strong preference for one of them (in this case, the nose would be exposed to the preferred stimulus for most of the time). Maybe it would be useful to examine AWC responses with different stimulus patterns?

2.2.d. We agree that it would be interesting to image under conditions of asymmetric dwell times, as these occur in Y-chip experiments. However, in considering whether to implement asymmetric or equal dwell times in the imaging experiments, we chose the latter because we were primarily interested in the question of whether AWC is capable of reporting food utility via its inherent sensitivity to each food, that is, before the rest of the worm acts on this information; in other words, without the confound of the feedback loop between stimulus and behavior. We now explain this reasoning more in the Discussion.

e) Fourth, in rationalizing 9E, authors say that they asked whether AWC might encode "some other economic quantity". How many such quantities were examined? Is this a case where correction for multiple comparisons would be appropriate?

2.2.e. We tested 18 other economic quantities (please see Supplementary file 3). None showed a significant correlation with AWC activation. This means there are no positive findings requiring correction for multiple comparisons.

Reviewer #3 (Recommendations for the authors):Introduction: GARP applies to consumer's choices, and importantly, assumes rational agents. The leap to nematodes is huge. Can the authors comment on this?

3.1. The implications of revealed preference theory can be viewed from two distance stances. The reviewer appears to adopt the first stance, “If an agent is rational, then his/her preferences will be consistent and transitive, they will satisfy GARP.” The second stance derives from the fact that having preferences that satisfy GARP is the necessary and sufficient condition for rationality. The force of necessity and sufficiency here is that GARP, can be used as a test for rationality, “If an agent’s preferences satisfy GARP, then he/her is rational.” Our study takes this second stance. We use GARP as, literally, a *litmus test* for utility maximization, one among many forms of rationality (Kacelnik, 2006). We use it solely for this limited purpose, following precedents set by other experimentalists (Camille *et al.*, 2011; Chung, Tymula and Glimcher, 2017; Pastor-Bernier, Stasiak and Schultz, 2019)

Line 102: Citation that can potentially be added: Ghosh et al., 2016, Neuron, PMID: 27866800 (for risk assessment).

3.2. Added.

Line 120: "it is reasonable to expect that C. elegans food choices maximize fitness"◊ The notion that an organism's food choices maximize fitness is directly related to the Optimal Foraging Theory. Among the major criticisms this notion has received, is that it relies on the assumption that natural selection and evolution have somehow resulted in a decision-making process that maximizes fitness. However, natural selection has not necessarily led to such a result when it comes to the feeding strategy or food choices of an organism, because it is a process of selection and not a mechanism that produces optimal results, e.g., behaviors. Therefore, it is debatable whether *C. elegans* food choices maximize fitness.Reviewer #1 and #3 discussed and reviewer #3 recommends to rephrase, e.g. "promote fitness" instead of "maximize".

3.3. Satisfaction of GARP indicates that there exists at least one utility function that renders the observed behavior perfectly maximizing. One of the limits of this approach is that we don’t get an exact specification of the utility function. This is true in two different respects. First, in mathematical terms, there can be a variety of different utility functions that fit the choice data equally well. Second, in mechanistic terms (psychological, biological), the units of the z-axis of the utility function remain obscure. One of the exciting aspects of the neuroeconomics approach, here and in general, is that by identifying the neural correlates of utility representations we can gain insights into the physical units of utility.

Line 145: redundant word: duringLine 146: bends, should be pluralLine 150: a period is missing at the end of the sentence.

3.4-3.7. All fixed.

Line 154: "GARP for worms": although this is the title of this Results section, the section itself is about explaining in detail how utility maximization and its violations can be manifested in graphic representation of bundle choices in experiments with human subjects. This section is not about *C. elegans* and GARP. I suggest revising the title. It is also unclear why this section is part of the Results, since there is practically no result presented.

3.8. We renamed this section “Revealed preference theory.”

Line 167: it seems to me that the opposite is inferred from the text (and from Figure 1 legend): apples are half the price of oranges. Is this a typo? -> see also reviewer #1

3.9. Silly mistake, thanks for catching it.

What is the budget of a *C. elegans* nematode? In line 535 and onward authors state that "We found it impractical to standardize the number of pumps to impose a fixed energy budget on each worm because […]", and then they explain that they have been plotting the fractional consumption and equation 5. Does this mean that the budget changes depending on the food source? Can the authors elaborate on this? Are worms' budget units consistent with the goods' prices units? -> see also reviewer #1

3.10. The sole purpose of the budget constraint in a human GARP experiment is to establish a trade-off between offered goods. There was no budget in our *C. elegans* experiments. That is because the trade-off was established by the geometry of the Y-chip.

Why do the authors consider the abundance of a bacterial strain equivalent to price? In human experiments, the ratio of, e.g., oranges vs apples is one of the variables (amount of fruit per weight), and the dollars per unit is another.

3.11. Actually, as stated in the original manuscript, we model price as the *inverse* of abundance, i.e., the inverse of density. This model is based on the intuition that food that is twice as dilute takes twice as much energy to consume per unit mass of food; the model views energy as cost. This point is now made explicitly in the text.

Line 236: Worms were cultured on *E. coli* OP50 before the H/M training took place. What is the nutritional value of this food source compared to H and to M?

3.12. This is an interesting question. In the original (and only) study of comparative food quality (Avery and Shtonda, 2003; Shtonda, 2004), quality was defined in terms of growth rate, computed as the inverse of the number of days to reach adulthood. Growth rates ranged from 0.28 day^-1^ to 0.50 day^-1^ (Shtonda, 2004). The H and M foods in our study, DA1877 and DA1885, have growth rates of 0.50 and 0.43, respectively. OP50 was not included in that study, but a derivative of OP50, DA837, has a growth rate of 0.45. By this measure, OP50 is a medium quality food, slightly better than our M food (DA1885).

Line 253: "If H food smells to the worm more like *E. coli* than M food does, this preference might be explained by the so-called food familiarity effect (Song et al., 2013), in which worms eat familiar food more readily than novel food." In this case, worms do not prefer H food based on its "quality" but based on its similarity to the food source they have been used to. In this case, maybe the sequence of preference choices does not constitute a violation of the utility maximization. In this case, worms do not choose driven by an urge to maximize their fitness, but rather by a tendency to maintain a familiar environment. How does this fit with the authors claims and assumption?

3.13. We do not claim that worms maximize fitness, at lease in the simple sense of optimizing food consumption. We claim that worms maximize utility or, in other terms, *welfare*, defined as fulfillment of their *subjective* preferences. Subjective preferences can be aligned with fitness, but they can also diverge from that norm. In humans, for example, drugs of abuse cause divergent preferences. In worms, the preference for familiar food even when it is lower in quality than other foods can be seen as a divergent preference. Adherence to GARP means only that preferences are internally consistent, regardless of whether they are aligned with or divergent from fitness. Therefore, we see no contradiction in our claims. Comments on fitness maximization have been added to the discussion.

Figure 2B: The caption reads "Mean preference versus time for trained and untrained ceh-36 mutants and N2 controls in open-field accumulation assays", however, I am not sure that this is what the figure shows, given also that the y axis is labeled as "fraction of pumps". Please explain. - see also reviewer #1

3.14. Apologies for this error, now corrected.

Given the issue with Figure 2B, it is hard to tell, at this point, which plot represents the open field assay and which the maze choice assay, therefore it is hard to compare the two or evaluate the findings in a combined way.

3.15. Caption now contains this information.

Line 298: "The decision to accumulate in a particular food can be made, at least in part, before the animal enters the patch." ◊ I think that, for the sake of accuracy, the decision in question is not about deciding to accumulate; a worm does not decide to accumulate, but rather to approach or reach or forage on food H or M. How do the authors comment on this?

3.16. We replaced decision to *accumulate* with decision to *approach*.

Have the Authors performed the open-field accumulation assay in the presence of sodium azide?

3.17. Yes. Please see Figure 4C.

The authors speak about latent learning (=a type of learning which is not apparent in the learner's behavior at the time of learning, but which manifests later when a suitable motivation and circumstances appear) in case of the non-trained animals, and about explicit learning (=a more conscious process where the individual makes and tests hypotheses in a deliberate search for answers) in case of the trained animals. This implies a dipole scheme of latent vs explicit. However, the authors do not justify the use of these two terms, nor do they provide evidence that in non-trained animals the learning is latent; in addition, there is no justification for the use of the term explicit in the second case. The worms have been simply conditioned or not conditioned, respectively. Even in lines 307-309 when they cite two papers by Worthy et al., stating that "Explicit food quality learning in C. elegans is formally equivalent to a type of classical conditioning in which an association is formed between the mélange of odors characteristic of particular bacteria species (Worthy, Haynes, et al., 2018; Worthy, Rojas, et al., 2018)", this is perplexing, because I am not sure that these papers claim or demonstrate that explicit learning in *C. elegans* is equivalent to classical conditioning. Despite that, the authors state that they are trying to find the locus of explicit food quality learning (line 259). I think the use of the two terms, explicit and latent, should be better justified or other terms should be used instead. -> see also reviewers #1/2

3.18. We have the terms *latent* and *explicit learning*. We now refer to preferences acquired through food quality training as *trained preferences* and preferences acquired in the absence of prior exposure to a particular food as *latent preferences*. Regarding the latter, we make it clear that latent preferences do not necessarily reflect learning; they could be developmental in origin.

In relation to that, in lines 323-333, it is implied that food learning occurs while the animals forage on the food patches. What is it exactly that worms learn?

3.19. The learning we observe is an increase in the normal bias to accumulate in H vs. M food (Figure 2B-D), and also increased consumption of H vs. M food (Figure 5B). In other words, trained worms learn to prefer H food more strongly than untrained worms.

Is this some form of non-associative learning?

3.20. We understand non-associative learning to be a process in which an organism’s behavior in response to a specific stimulus changes over time in the absence of any evident link to (or association with) consequences or other stimuli that would induce such change. In a simple non-associative model, increased bias for H could be the result of sensitization to H food, habituation/adaptation to M food, or both. In an associative model (e.g., classical conditioning), odors specifically emitted by H and M food would serve as conditioned stimuli, CS1 and CS2. The unconditioned stimulus is the presumptive reinforcing effects of consuming H or M food. At present, we cannot definitively distinguish between these two possibilities. The section “Establishment of subjective value assignments” in the Discussion now includes these points.

In addition, the effect on their preference developed as they forage on a specific food patch is something that should be present in both trained and non-trained animals. The authors state that "It is possible, therefore, that food quality learning in cat-2 mutants is impaired simply because they spend less time in the food, hence have less experience of it", line 321. Does this mean the authors believe that food quality learning occurs during the worms' stay on the food patch during the open field or maze assay and not during training?

3.21. We believe that the majority of the change in food preference occurs during the training phase, which lasts 18-24 hours as opposed to the testing phase, which lasts only 1 hr.

Have the authors tested only two types of food, i.e., two bacterial strains characterized as medium and high quality?

3.22. Yes, in these laborious experiments, we have tested only one pair of food. This limitation of our study is now pointed out in the Discussion.

If so, then the reported results could be due to other properties of the bacteria and not their nutritional value (case-specific results).

3.23. In the original manuscript, we noted that although adherence to GARP does not reveal what is being maximized, we were able to identify three aspects of bacterial food that are *not* being maximized: amount of high quality food consumed, overall amount of food consumed, and potential for rapid growth. So, we agree with the reviewer: the reported results are likely due to phenomena other than nutritional value. This point is now made in the text.

Note that the nutritional value (energy, joules) is what the authors take into account in equation 1.

3.24. There seems to be some confusion here. In the original manuscript, the term in equation 1 representing energy refers to energy required to swallow food, not the energy gained by swallowing food. That being said, equation 1 in the revised manuscript no longer includes an energy term.

Wouldn't their findings be stronger if the observed preferences were repeated for other pairs of food choices, i.e., by testing other/more bacterial strains?

3.25. Please see our response to 3.22.

Figure 4C and 4D: what is the difference between the two? Do they refer to M and H?Similarly for 4E and 4F: do different colors (green and gold) represent different foods e.g., H and M? Please clarify.

3.26. Thanks. We have altered the figure to remove this ambiguity.

Line 524: Maybe these data should be shown, as they constitute an important link in the authors' train of thought.

3.27. These data appear in Figure 5B in the original and revised manuscript. The accompanying paragraph has been edited for clarity.

Line 550: Can the authors explain why this is expected to be a gaussian distribution?Line 551-554: Is there a plot/figure to illustrate this?

3.28. The distributions of preference values in contributing to the means in Figure 5C were well fit by gaussians. See new Figure 5—figure supplement 1.

Line 561: I don't think this is a particularly conservative null hypothesis; in most cases in behavioral assays the null hypothesis is exactly this: that animals choose randomly, i.e., that the treatment has no effect.

3.29. We are a bit confused here, as simulated worms *did* choose randomly. We stated, “… the fraction of pumps in H food is drawn from a flat distribution between 0 and 1.” Nevertheless, we have eliminated the qualifying adjective, “highly.”

Lines 604-606 and Figure 6A: I do not understand why e and d are grouped both in the group in which H changes and in the one H stays constant. Can the authors please explain?

3.30. We have added a table that clarifies the design of this analysis (Supplementary file 1).

Line 673: "worms require more of it to be satisfied": How do the authors define or measure satisfaction in this case? Without a definition this sentence is misleading.

3.31. We agree. We eliminated “satisfaction.” We now say, “Thus, not only did training make H food more valuable, it also made diminishing marginal utility less relevant to valuation.”

Line 700: eliminate or support?

3.32. Typo. Text now states, “Together, these findings strongly favor the dwell-time model over the frequency model.”

Line 766-767, and the entire paragraph: Authors say that "As exogenous activation of AWC produces a bout of reverse locomotion (Gordus et al., 2015), its response to food offset promotes reversals, leading to increased retention patches. Our imaging data suggest that AWC-mediated retention would be stronger for H than M food, promoting greater preference for H food, as seen in Figure 2B-D." The authors rely on their finding that AWC was more strongly excited by H than M food, to jump to the conclusion that "AWC-mediated retention would be stronger for H than M food, promoting greater preference for H food". In my view, this last claim could constitute a well-defined hypothesis to be tested, and not a conclusion based on their findings, even in combination to Gordus et al. In fact, it is Gordus and colleagues who highlight in their 2015 paper that although the activation pattern of AWC is deterministic ("highly reliable"), the resulting reversing behavior is, by contrast, probabilistic: "The AWC calcium response, which is likely correlated with depolarization, is highly reliable from trial to trial, even after dozens of odor presentations (Larsch et al., 2013). By contrast, the reversal response is probabilistic. Even under well-controlled conditions, animals may or may not reverse on individual trials, regardless of the strength of the AWC calcium response (Larsch et al., 2013)" and that "Thus, the variability in the behavioral response results from variable transmission of information from the AWC sensory neuron to AVA command neurons." Based on the above, I do not understand the authors' claim. Maybe they can run an experiment to show that the claim is valid? The results presented in Figure 2, although they show increased retention, they do not allow the conclusion that the animals with stronger AWC activation are the ones that present increased retention. That individuality in the response is, in my view, what Gordus et al. indicate in their work.Similarly, in the following paragraph, based only on the strength of AWC activation, they conclude that "retention in H food patches would be stronger in trained than untrained animals, promoting greater preference for H food after training". However, stronger activation of AWC does not mean stronger retention (behavioral outcome), based on Gordus et al.

3.33. We should have made it more clear that our argument is probabilistic in nature. Our model is that greater activation of AWC upon removal of H food is consistent with a higher *probability* of being retained in H vs. M food patches. The text has been revised to emphasize the fact that our argument is probabilistic.

Line 1114: The mask was baited or the maze-shaped NGM area?

3.34.a. The latter. Methods section now states, “The mask was placed on the NGM surface and the maze was baited with the same amounts of H and M food, which were placed on the agar surface at the end of each arm.”

Where there any escaping animals? Where there any animals that did not choose either of the two (e.g., not reached neither H or M in the allowed time)?

3.34.b. Methods now states, “Preference at each time point was computed as (NH−NM)/(NH+ NM), where N is the number of worms in contact with the food-type indicated by the subscript; worms not in contact with food, and the small number of escaped worms (~5%), were not counted.”

How long were the worms allowed to make a decision inside the maze?

3.34c. In open-field and maze experiments, worms were allowed to reveal their food preferences for 60 min as indicated in original versions of (Figure 2C,D). methods section now includes this information … “Accumulation was scored for 60 min. at 15 min. intervals”

Were there any animals censored, and if so, does this affect the results?

3.34.d. No animals were censored. Text now states this this.

Did the authors alternate the position of H/M food in the two maze arms to eliminate any side preference effect?

3.34.e. We did not alternate the position of H/M food. The consistent differences between trained and untrained animals across manipulations (azide and *cat-2*) testify to the fact that worms were responding to food rather than extraneous stimuli.

Line 914: what do the authors mean by "infra-human species"? Maybe non-human?

3.35. “Infra-human” removed.

Line 919: "literally guarantees", and elsewhere in the text: the authors tend to express their claims using absolute language (see also title, "exactly as"). I feel that it might be more appropriate if the language used is firm and confident, as the authors see fit, but milder.

3.36. “Literally” and “exactly” have been removed.

Lines 914-922, and elsewhere in the text:a. The authors are trying to convince the reader that *C. elegans* makes consumption decisions in the same way that a human consumer does. What's more, they claim that such a human-conforming behavior can be captured in nematodes by just mapping the neuronal circuit that steers head bending. This is in the core of the paper, and in my view, it is problematic in many ways. The axioms of the revealed preference theory are justified by the assumption that humans are rational agents. This means that they make rational decisions. Do the authors claim the same for *C. elegans* nematodes?

3.37.a. This concern seems similar to point 3.1. As noted there, we are using revealed preference theory as a *litmus test* for utility maximization. In doing so, we are following well-established precedents cited in the original manuscript. A litmus test for utility maximization does not assume utility maximization or any form of rationality. On the contrary, it is a test for utility maximization.

b. In parallel, the fact that *C. elegans* feed on two different food sources and they choose one over the other in a way that the goods qualify as substitutes, is a finding presented as of extreme importance. This is indeed an interesting conclusion, well supported by the data. It becomes more interesting because of the untangling of the neuronal circuit involved. At the same time, this is really not an unexpected result, although providing evidence for it is definitely useful. Any animal that feeds on multiple food sources would alternate between two of them, based on their availability, the effort required, the nutrition provided. Indeed, the authors themselves do not claim that this is unexpected. However, they claim that they "break grounds" because worms' behavior appears to conform with utility maximization principles and can be described with human-centered terms of substitute goods. This is misleading, first, because of the agent rationality assumed, which we have no basis for, and second, because identifying two goods as a substitutes pair is not ground-breaking on its own. Behavioral economics labels pairs of goods as such in order to move on with more complicated claims, theorems and analyses.

3.37.b. We wish to emphasize, as noted in 3.37.a and also in 3.1 that our approach does not assume rationality; rather, it tests whether *C. elegans* choice behavior is consistent with a particular model of rational choice, namely utility maximization. As stated in the Discussion, from our perspective the most prominent aspects of our study are (1) *C. elegans* food choices obey the classic law of supply and demand; (2) *C. elegans* behaves as if maximizing utility; (3) *C. elegans* food-consumption decisions are well fit by the CES utility function; (4) A plausible mechanism for utility maximization has been identified. We have replaced “breaks new ground” with “establishes a new reference point.”

c. The authors are interpreting *C. elegans* feeding behavior using behavioral economics terms that sound extravagant when speaking of nematodes, but in reality, their findings are not extravagant and certainly do not need to be dressed as such in order to be significant. In this reviewer's view, the findings presented in the manuscript are interesting and they constitute a significant contribution. Attempting to attribute to them a dimension disproportionate to their real depth, dampens the initial enthusiasm in an unnecessary way.

3.37c. We wish to disagree, respectfully, with the concern that we are applying revealed preference theory inappropriately (“extravagantly”). On the contrary, we have taken revealed preference theory at its word, and applied it to a remarkably simple organism that nevertheless passes the test for utility maximization. This finding is significant not only for what reveals about the sophistication of *C. elegans* behavior, but also for our understanding of revealed preference theory itself. Economists will almost certainly be prompted to reexamine the perceived view that revealed preference decisions require awareness and cognition.

Line 979: What are the worms maximizing? This is a question that the authors admit comes up inevitably, but interestingly (and honestly, of course) at the same time they state that they do not have a plausible answer to it. Therefore, even the notion of utility (which is supposed to be maximized) remains obscure. In my understanding, this confirms the fundamental problem with the way this study is presented. The authors are not working toward testing a stated hypothesis (e.g., nematodes' behavior is such that maximizes x), but they rather quantify a nematode behavior in a way that fits the equations used in behavioral economics. What they don't take into account is that, sometimes, even if phenomenon A can be phenomenologically described by a set of equations that has been developed to describe (part of) phenomenon B, this does not mean that A is explained by the same physical or biological principles that steer phenomenon B. This is even more prominent if the organism involved in A is so very different regarding its brain faculties and societal construct than organism B. And even more so if the researchers fail to provide a satisfactory answer for the biological (in lack of maybe a psychological, societal or other) explanation for this behavior.

3.38. We wish to disagree, respectfully, with the reviewer in couple of respects. (i) “The authors are not working toward testing a stated hypothesis.” On the contrary, we have tested the hypothesis that *C. elegans* decision behavior satisfies the necessary and sufficient conditions for utility maximization. In other words, we tested the hypothesis that *C. elegans* choices maximize something. (ii) Although we have established that *C. elegans* choices satisfy the behavioral conditions for utility maximization we do not, on this basis, conclude anything about the biological mechanisms of this phenomenon. To do otherwise would be to misunderstand the limitations of revealed preference theory, which by design is not concerned with mechanism. Nevertheless, our finding of utility maximization in one of the most experimentally tractable organisms in contemporary neuroscience has potentially significant implications. It sets the stage for discovering how at least one nervous system implements this extraordinarily important behavior.

References

Avery, L. and Shtonda, B. B. (2003) ‘Food transport in the *C. elegans* pharynx’, *Journal of Experimental Biology*. doi: 10.1242/jeb.00433.

Camille, N. *et al.* (2011) ‘Ventromedial frontal lobe damage disrupts value maximization in humans’, *Journal of Neuroscience*. doi: 10.1523/JNEUROSCI.6527-10.2011.

Chung, H.-K., Tymula, A. and Glimcher, P. (2017) ‘The Reduction of Ventrolateral Prefrontal Cortex Gray Matter Volume Correlates with Loss of Economic Rationality in Aging’. doi: 10.1523/JNEUROSCI.1171-17.2017.

Gordus, A. *et al.* (2015) ‘Feedback from network states generates variability in a probabilistic olfactory circuit’, *Cell*, 161(2), pp. 215–227. doi: 10.1016/j.cell.2015.02.018.

Harbaugh, W. T., Krause, K. and Berry, T. R. (2001) ‘GARP for Kids’, *The American Economic Review*, 91(5), pp. 1539–1545.

Kacelnik, A. (2006) ‘Meanings of rationality’, in Hurley, S. L. and Nudds, M. (eds) *Rational Animals?* Oxford: Oxford University Press, pp. 87–106.

Larsch, J. *et al.* (2013) ‘High-throughput imaging of neuronal activity in *Caenorhabditis elegans*’, *Proceedings of the National Academy of Sciences of the United States of America*, 110(45). doi: 10.1073/pnas.1318325110.

Lee, K. S. *et al.* (2017) ‘Serotonin-dependent kinetics of feeding bursts underlie a graded response to food availability in *C. elegans*’, *Nature Communications*. Nature Publishing Group, 8, pp. 1–11. doi: 10.1038/ncomms14221.

Niacaris, T. and Avery, L. (2003) ‘Serotonin regulates repolarization of the *C. elegans* pharyngeal muscle’, *Journal of Experimental Biology*, 206(2), pp. 223–231. doi: 10.1242/jeb.00101.

Pastor-Bernier, A., Stasiak, A. and Schultz, W. (2019) ‘Orbitofrontal signals for two-component choice options comply with indifference curves of Revealed Preference Theory’, *Nature Communications*, 10(1). doi: 10.1038/s41467-019-12792-4.

Raizen, D. M. and Avery, L. (1994) ‘Electrical Activity and Behavior in the Pharynx of *Caenorhabditis elegans*’, *Neuron*, 12, pp. 483–495.

Shtonda, B. B. (2004) *ELECTROPHYSIOLOGICAL AND BEHAVIORAL MECHANISMS OF Caenorhabditis elegans FEEDING*. University of Texas Southwestern Medical Center.

Shtonda, B. B. (2006) ‘Dietary choice behavior in *Caenorhabditis elegans*’, *Journal of Experimental Biology*. doi: 10.1242/jeb.01955.

Song, B.-M. *et al.* (2013) ‘Recognition of familiar food activates feeding via an endocrine serotonin signal in *Caenorhabditis elegans*’, *eLife*, 2013(2). doi: 10.7554/*eLife*.00329.

Worthy, Soleil E. *et al.* (2018) ‘Identification of attractive odorants released by preferred bacterial food found in the natural habitats of *C. elegans*’, *PLoS ONE*, 13(7), pp. 1–14. doi: 10.1371/journal.pone.0201158.

Worthy, Soleil E *et al.* (2018) ‘Identification of Odor Blend Used by *Caenorhabditis elegans* for Pathogen Recognition’, *Chemical Senses*, 43(January), pp. 169–180. doi: 10.1093/chemse/bjy001.

Zaslaver, A. *et al.* (2015) ‘Hierarchical sparse coding in the sensory system of *Caenorhabditis elegans*’, *Proceedings of the National Academy of Sciences of the United States of America*, 112(4), pp. 1185–1189. doi: 10.1073/pnas.1423656112.

[Editors' note: further revisions were suggested prior to acceptance, as described below.]

Reviewer #3 (Recommendations for the authors):The authors have extensively revised large parts of the manuscript, especially in the Results and Discussion sections. The revised manuscript also includes updated figures and captions, new supplementary figures, etc. The manuscript is in much better shape, and much more information is provided, information that is critical to evaluate the results, their interpretation, and the final conclusions. The addition of new authors and expertise, as well as the change in the title are very welcome, too. Many of my concerns were addressed, and I appreciate the detailed explanations provided in some of the authors' responses. My understanding of their work is now much better. Nevertheless, the work continues to present fundamental problems. These are reflected mainly in comments #1, #5, and #11, which can be considered my major concerns. The rest of the comments can be considered minor or easier to answer.1. Comment 1 of first revision and response 3.1:The authors write in their rebuttal letter that they adopt the stance that if the agents' behavior adheres to the GARP, then they are rational (and not that rationality is a prerequisite). In their response, the authors list 3 papers to support the stance they claim to adopt, and all three of them have to do with humans, and the work reported has to do with reduced or altered observed rationality because of trauma, aging, etc. Displaying reduced rationality is fundamentally different than potentially lacking the very ability to be rational. I still feel that the issue of rationality needs to be addressed clearly in the manuscript. I explain:In the revised manuscript, apart from two references to published work on *C. elegans* observed occasional rationality with many exceptions (lines 106-110), the only important part that addresses rationality is the new text in lines 228-234. Here, the authors mention (line 228) that "tests of adherence to GARP have been utilized to assess the degree to which decision-making agents are rational in the economic sense", and later in the paper, they claim that nematodes adhere to GARP. The authors need to make clear whether they claim rationality for *C. elegans*, based on their findings, and if so, how is this rationality defined. And in addition, they need to make clear that whatever rationality is claimed for worms, it is not to be confused with what is broadly understood as rationality but is rather a term to describe a mathematically defined concept, the criteria for which are met by nematodes. If this is not absolutely clear, then the reader can be easily misled.

We have added a paragraph that makes clear that GARP establishes a limited type of rationality, which we refer to as *technical rationality*. We state that technical rationality neither presupposes nor establishes rationality in the psychological sense of awareness and reasoning (lines 222-230).

5. Line 301 (related also to response 3.18): " [The preference of naïve worms] could be the result of prior experience or developmental maturation": I do not understand this. What kind of prior experience? If I understand correctly, the worms have been grown in the lab for their entire life, and for a number of prior generations. Have they been fed anything else than *E. coli* during some part of their life? Have they been exposed to H or M food before? Have their ancestors been exposed to H or M food? If not, then the "prior experience" argument cannot possibly apply. If yes, then this should be mentioned, as it would dramatically change the perspective. As for developmental maturation: If the authors believe this is true, then assessing worms ranging from L3 to Day 1 is obscuring, and distinct age cohorts should be assayed separately (L3, L4, Day 1). Moreover, if this were true, it would mean that older or younger animals prefer, for example, H. Therefore, the ratio of younger/older animals in the tested population could skew the results this way or the other. How do the authors comment on that? As for the trained worms, acquiring a preference because of sampling the two foods during the 60 min training assay seems the most reasonable thing to assume. However, the authors claim that this scenario does not apply, because of the findings of the familiarity experiments in the Y chip, but it is not clear to me why this is the case. Besides, the conclusion in line 522 that the foods are qualitatively distinct, would explain the development of a preference during the 60 min of the assay.

We have deleted the sentence quoted above.

11. The new information provided in the revised manuscript on the budget constraint and the price of goods H and M, and how the budget constraint is imposed because of the Y-chip design, allowed me to get a new perspective of the Y-chip experiments. In consequence, several concerns arise. The process is as follows, roughly: A worm is placed in the chip, and two streams, one of H and one of M food, are provided. The preference is revealed by the worm bending its head toward the M or the H stream, and by performing pumps. The combination of pumps of H or M food is the preferred bundle. The whole process lasts for 12 mins. Therefore, it takes 12 min for the worms to behave in a way that reveals their preference. During these 12 mins, the preference is revealed through a sequential decision-making process, in which the worms sample the goods.i) The sequential nature of this process is fundamentally different than the one-time decision made by human subjects in analogous experiments. In those, humans select the bundle of preference in a single decision. Nematodes, however, make many decisions, sequentially, that define the preferred bundle. How do the authors believe that this affects the interpretation of the results, with respect to adhering to the GARP prerequisites? I posit that this difference can have many implications as someone is trying to determine whether nematodes meet the same prerequisites that humans meet but while under very different experimental conditions. Some additional methodological concerns are listed below.

The reviewer makes an important point here. Yes, the worm can be said to make a series of decisions during the 12 min. testing period, whereas GARP tests on humans involve one-shot decisions. This could be confusing to readers. However, the history of GARP and its relation to behavioral economics is illuminating here. As the reviewer will know, GARP is a refinement of the Strong Axiom of Revealed Preference (SARP) which, in turn is a refinement of Samuelson’s 1938^1^ Weak Axiom of Revealed Preference (WARP). In that work, which launched revealed preference theory, consumption is amount purchased per unit time:

“I assume in the beginning as known, i.e., empirically determinable under ideal conditions, the amounts of *n* economic goods which will be purchased per unit time by an individual faced with the prices of these goods and with a given total expenditure.”

Clearly, Samuelson had in mind the accumulation of goods over time and, therefore, a series of consumption decisions. Later, revealed preference theory was used to build models of ‘lifetime consumption,’ ^2–4^ including retirement spending. As this type of spending also occurs over time, it too implicitly involves a series of consumption decisions.

As our consumption metric integrates the rates of consumption of H and M foods over a fixed time, it is consistent with Samuelson’s original theory. Our definition of preference includes whatever micro-decisions are made during that time. This situation is akin to defining consumption in humans, as economists do, in terms of what is in the grocery cart as a consumer leaves the store. Placing items in the cart is a sequential process (one orange at time, so to speak), and initial biases and non-stationary preferences are irrelevant. Of course, by changing the definition of preference one could take those phenomena into account, but revealed preference theory does not require this.

It was not until the studies of Andreoni, Glimcher, Harbaugh, and Kariv that the classical theory of revealed preference was realized in terms of one-shot decisions. The one-shot decision framework was adopted purely for experimental purposes. Therefore, GARP can be investigated either way: in terms of sequential decisions over time, or in terms of one-shot decisions.

We have added to the manuscript a paragraph that explains why sequential decisions are consistent with revealed preference theory (lines 222-230).

ii) During the sequential sampling, the worm needs to bend its body toward this or the other side, in order to perform a pump. Consequently, switching between sides, therefore foods, has a cost for the worm (note that bending just the head does not allow for switching between the two flows, as evident in Video 1). How do we know that this cost does not affect its decision-making with respect to switching between foods? In addition, even though the authors say that the body bending while in the chip resembles the body bending during the sinusoidal locomotion on a GNM plate, in Video 1 we can clearly see the worm bending multiple times toward one side before switching to the other side. This is not what happens during regular locomotion. This can be related to the worm preferring to feed on one food, but also to the cost of the effort required in order to bend to the other side. Given the constraint along the mid-body, this effort seems not to be negligible.

We agree that, in theory, there could be energetic costs for switching sides. There could also be opportunity costs. And, during sequential decision making, these costs may contribute to relative preference for each food, in a process akin to effort discounting. But revealed preference theory by design is blind to underlying factors contributing to preferences. Therefore, costs, if they do occur, do not undermine our finding of utility maximization. One could argue that costs might be different in a differently shaped chip and, therefore, that relative preferences might be different as well. But we do not claim that our methodology measures the worm’s preferences independent of testing conditions. Nor would such a claim be a necessary condition for utility maximization.

iii) How do we know that the worm is not biased by the content of the first few pumps? Is there a correlation between the first few pumps and the final preferred bundle?

This is an interesting question. Our study does not have the statistical resolution to answer it, but we believe this question is not relevant to our finding of utility maximization. That’s because the food first eaten in the chip can be assimilated to the individualized history of the worm, its accumulated experience over its lifetime, including during testing. This history is part of what makes preferences subjective. And, as noted above, revealed preference theory is blind to underlying factors contributing to individual preferences.

It is also worth noting that the type of bias the reviewer imagines does not necessarily have an effect on population-level preference. Here’s the proof.

Let’s assume that worm *is* biased by the content of the first few pumps, such that it strongly prefers what it first ate. As stated in Materials and methods, worms are accommodated to the Y-chip in food-free buffer. After accommodation, foods are introduced. We can reasonably assume that the initial pumps will be left-right randomized, according to posture of the worm when food first reaches it. This situation can be modeled asfHH= nH+xH(nH+xH)+(nM−xH)fHM=nH−xM(nH−xM)+(nM+xM)

where fHH is the preference of worms the that first ate H food, and fHM is the preference of worms the first ate M food. The variables xH and xM represent, respectively, the extra pumps that occurred in H or M food. The expected preference across the population isFH=12(fHH+fHM)

Substituting for fHH and fHM and combining terms givesFH=nHnH+nM+xH−xM2(nH+nM)

In the simple case where the number of extra pumps on either side is equal (xH=xM), the right hand term in the above equation is zero, meaning there is no effect of food first eaten. In the extreme case of complete bias toward the food first eaten, xH=nM and xM=nH, and FH = 0.5. This scenario does not match our data, as preferences much greater or much less than 0.5 were observed (Figure 5B). For xH≠xM, FH can take a range of values greater or less than the baseline preference of FH=nH(nH+nM). Thus, the initial experience of each worm in the chip could be a factor that shapes preference. But, as noted above (lines 223-225) revealed preference theory is blind to underlying factors contributing to preferences.

iv) The initial body orientation of the worm is not controlled, as I would guess. How do we know that it does introduce a bias with respect to the initial conditions of the system?

This concern seems analogous to (iii). Initial posture can be assimilated to the worm’s subjective history, with no implications for the applicability of revealed preference theory.

iv) The fact that the worm develops a preference over the course of 12 min, as it is tasting (sampling) the two foods, cannot be overlooked. According to the literature, this can be enough time, for worm standards, to develop behavioral plasticity. How could this affect the "revealed preference" and the adherence of the worm to the GARP prerequisites? Especially since human subjects, who participate in carefully controlled experiments, do not undergo a similar (sequential) process.

As point of clarification, it is not a *fact* that preference “develops” during the 12 min. testing period. At this point, we don’t know whether or not this happens because, again, our data do not have the required statistical resolution. But, even if preference is modulated by experience in the chip, that experience can be assimilated to the worm’s subjective history, with no implications for the applicability of revealed preference theory.

v) The (iv) relates to the development of a preference during training in a plate with patches, for the accumulation assay, see comment #5. The authors are reluctant to admit that worms can develop a preference because of sampling the two foods during the 60min training assay. (1) They attempt to justify this reluctance based on the results of the familiarity experiments on the Y-chip. As mentioned in comment #5, I am not sure how the familiarity experiments lead to the conclusion that worms do not develop a preference during training. (2) Does this have to do with the authors not acknowledging that worms might be developing a preference during the Y-chip experiments?

1. It is conceivable that learning-through-sampling occurs during the 60 min food-preference assays. We now say this in the text (lines 302-303). But whether or not learning occurs during this assay has no bearing on the main conclusion of the paper – that the worm behaves as if maximizing utility – because that conclusion is based a different assay.

2. We agree that the logic behind the statement that familiarity experiments argue against preference acquisition during the Y-chip assay was unclear; it has been deleted.

All the above (i-iv) leave the key claim "Overall, this methodology results in decision-making paradigm that is formally analogous to GARP experiments in humans", line 1054, unsupported.To conclude: the revised manuscript is in much better shape than the original draft, and I recognize the authors' efforts to respond to the reviewers' many comments. The amount of work performed is impressive. However, this paper still presents fundamental problems. These are reflected mainly in comments #1, #5, and #11. Worm's adherence to GARP prerequisites is proposed to be the case, but the methodologies through which this conclusion is drawn and the interpretation of some of the findings are problematic. Trying to adjust human GARP behavioral experiments so that they can be run with nematodes does not seem to work, in this reviewer's opinion. Many of the parameters that experimenters are trying to control when studying humans, in order for them to be able to reliably claim adherence to GARP, are not controlled in the nematode experiments presented here (comment #11), something that can dramatically affect the interpretation of the results. Furthermore, important clarifications need to be made so that the reader is not misled (comment #1), and some explanations the authors provide for some of the observed behaviors are inadequately substantiated (comment #5).In light of the above, even if the authors addressed everything else satisfactorily, with concern #11 unresolved, I do not see how the main claim can be sufficiently supported and how this paper can be published in eLife. The claims that the authors are making are such that their substantiation should be above any ambiguity.Reviewer #3's additional comments from the consultation session:Sequential decision-making in the Y-chip is a fundamental concern. Here is why: The main claim the authors are making is that *C. elegans*' behavior abides by the GARP requirements. They claim so, based on their interpretation of experimental findings. By implementing a certain experimental design, they experimentally pose a question to the system, and the system (nematode consumers) responds by demonstrating a certain behavior, and this behavior is interpreted as GARP behavior because presumably, it is similar to the behavior displayed by the human counterparts. As a reference, when behavioral economists run experiments with human subjects, they apply a certain experimental design, they experimentally pose a question to the system (human consumers), and the system responds by demonstrating a certain behavior, and this behavior is defined as GARP behavior. Note that the experimental design followed by behavioral economists is very strictly controlled, to make sure that no undesired variables interfere with the results. However, the Y-chip experiments pose a different question to nematodes, than the question posed to human consumers: the Y-chip asks what the outcome is of a sequential decision-making process, whereas the GARP-defining human experiments ask what is the outcome of a one-time decision-making process. In other words, the problem is that Katzen et al. introduce a variable that is not present in the experimental design used in behavioral economics to decide whether the subject behaves as GARP: the variable of time. The worms display sequential decision-making, but GARP is supposed to explain a one-time decision-making behavior, not a sequential one. This discrepancy, ignoring the extra variable in the Katzen et al. experiment, is problematic and potentially very misleading.I am particularly concerned because of the potential impact of the claim that authors make: nematodes make economic decisions that can be described by the same set of principles that describe human economic decisions. For such a huge claim to stand, it has to be supported in a concrete way, without gaps in the thought process.Bottom line: I understand that the experimental design is what it is and that there is no way the Y-chip experiments are performed differently so that the worms could make a one-time decision. Therefore, I see two options: (A) In the current version of the manuscript, because of this discrepancy, the claim is weak and is not adequately supported, in my view. Therefore, I cannot suggest the manuscript be published in eLife. (B) The authors update the manuscript to admit the discrepancy, and they make it absolutely clear to the reader that the nematode system makes its decision according to a process very different than the one that human systems make their decision, in corresponding reference experiments that decide the subjects' GARP behavior. If the authors do that, then everything will be transparent, and the readers can decide for themselves whether the claim stands or not. In such a case, I am fine with publishing the work in eLife, if such is the suggestion made by everyone else.

We understand the distinction the reviewer has made and the confusion it could cause, and we are grateful to the reviewer for pointing this out to us. However, as described above (lines 173-207), quantification of consumption over time (sequential decisions), is consistent with classical revealed preference theory. Accordingly, relative to (A), the use of this method does not in any way weaken the claim of utility maximization in *C. elegans* and, regarding (B) there is no discrepancy to admit.

That said, we agree that the manuscript could be improved by clarifying the relationship between one-shot decisions and sequential decisions relative to classical revealed preference theory. We have added such a paragraph (lines 457-466).

References

1. Samuelson, P. A. A note on the pure theory of consumer’s behaviour. Economica 51, 61–71 (1938).

2. Ghez, G. R. & Becker, G. S. The Allocation of Time and Goods over the Life Cycle. (NBER, 1974).

3. Friedman, M. A Theory of the Consumption Function. (Princeton Universitiy Press, 1957).

4. Deaton, A. Understanding Consumption. (Clarendon Press).